# Characterizing, Evaluating, and Optimizing Complex Reasoning

**Haoran Zhang** [1 2]  **Yafu Li** [2 3]  **Zhi Wang** [4 2]  **Zhilin Wang** [2 5]  **Shunkai Zhang** [2 6]  **Xiaoye Qu** [2]  **Yu Cheng** [3 2]

## Abstract

Large Reasoning Models (LRMs) increasingly rely on reasoning traces with complex internal structures. However, existing work lacks a unified answer to three fundamental questions: (1) what defines high-quality reasoning, (2) how to reliably evaluate long, implicitly structured reasoning traces, and (3) how to use such evaluation signals for reasoning optimization. To address these challenges, we provide a unified perspective. (1) We introduce the *ME$^2$ principle* to characterize reasoning quality along macro- and micro-level concerning efficiency and effectiveness. (2) Built on this principle, we model reasoning traces as directed acyclic graphs (DAGs) and develop a DAG-based pairwise evaluation method, capturing complex reasoning structures. (3) Based on this method, we construct the TRM-Preference dataset and train a Thinking Reward Model (TRM) to evaluate reasoning quality at scale. Experiments show that thinking rewards serve as an effective optimization signal. At test time, selecting better reasoning leads to better outcomes (up to 19.3% gain), and during RL training, thinking rewards enhance reasoning and performance (up to 3.9% gain) across diverse tasks. Code and data are available at https://github.com/Simplified-Reasoning/TRM.

## 1. Introduction

As Large Reasoning Models (LRMs) advance, reasoning has emerged as a central mechanism underlying their per-

formance (Guo et al., 2025; Jaech et al., 2024). To solve challenging problems, LRMs increasingly rely on intermediate reasoning traces to perform multi-step deliberation before producing user-facing final responses (Shao et al., 2025). The final responses are typically concise and actionable, whereas the reasoning traces are longer, structurally more complex (Muennighoff et al., 2025; Guo et al., 2025), often exhibiting human-like cognitive behaviors such as exploration, reflection, and revision (Jiang et al., 2025). These traces are commonly generated automatically without rigorous quality control or standardized verification (Zou et al., 2025). As a result, they might contain redundant, inconsistent, or incorrect reasoning patterns, leading to increased inference cost, distorted conclusions, and unreliable evaluation (Lee & Hockenmaier, 2025; Kang et al., 2025; Zhan et al., 2025). Improving the quality of reasoning traces is therefore crucial for advancing the efficiency and effectiveness of large reasoning models.

Despite growing interest, existing research still lacks a unified answer to three fundamental questions: **(Q1)** What constitutes a high-quality reasoning trace? **(Q2)** How can we reliably evaluate long and implicitly structured reasoning traces? **(Q3)** How can we leverage such evaluation signals to optimize reasoning at scale? While prior work has explored individual aspects of reasoning, e.g., step-level correctness, local coherence (Jacovi et al., 2024), or verbosity (Hong et al., 2025), and has modeled reasoning with explicit structures (e.g., trees or graphs) (Jiang et al., 2025), these efforts do not yet offer a unified and scalable notion of reasoning quality that jointly accounts for both *structure* and *content*.

Recently, process reward models (PRMs) assign supervision to intermediate reasoning steps (Lightman et al., 2023; Xia et al., 2025) (e.g., ReasonFlux-PRM (Zou et al., 2025)), but typically rely on prompt-based absolute scoring (e.g., step-wise correctness labeling) and have limited capacity to capture long-range dependencies and complex, non-linear reasoning structures (Jiang et al., 2025). In contrast, outcome reward models (ORMs) trained with pairwise preferences offer more reliable and scalable supervision than absolute scoring (Levtsov & Ustalov, 2025; Stiennon et al., 2020), but they are designed to align response end outcome (e.g., helpful, honest, and harmless (Askell et al., 2021)) rather than to evaluate the quality of structured reasoning traces that reflect human-like cognitive processes.

Author email: Haoran Zhang <zzzhr97@gmail.com> [1]School of Artificial Intelligence, Shanghai Jiao Tong University, Shanghai, China [2]Shanghai Artificial Intelligence Laboratory, Shanghai, China [3]The Chinese University of Hong Kong, Hong Kong, China [4]Nanjing University, Suzhou, Jiangsu, China [5]University of Science and Technology of China, Hefei, Anhui, China [6]Peking University, Beijing, China. Correspondence to: Yafu Li <yafuly@gmail.com>, Yu Cheng <chengyu@cse.cuhk.edu.hk>.

*Proceedings of the 43$^{rd}$ International Conference on Machine Learning*, Seoul, South Korea. PMLR 306, 2026. Copyright 2026 by the author(s).

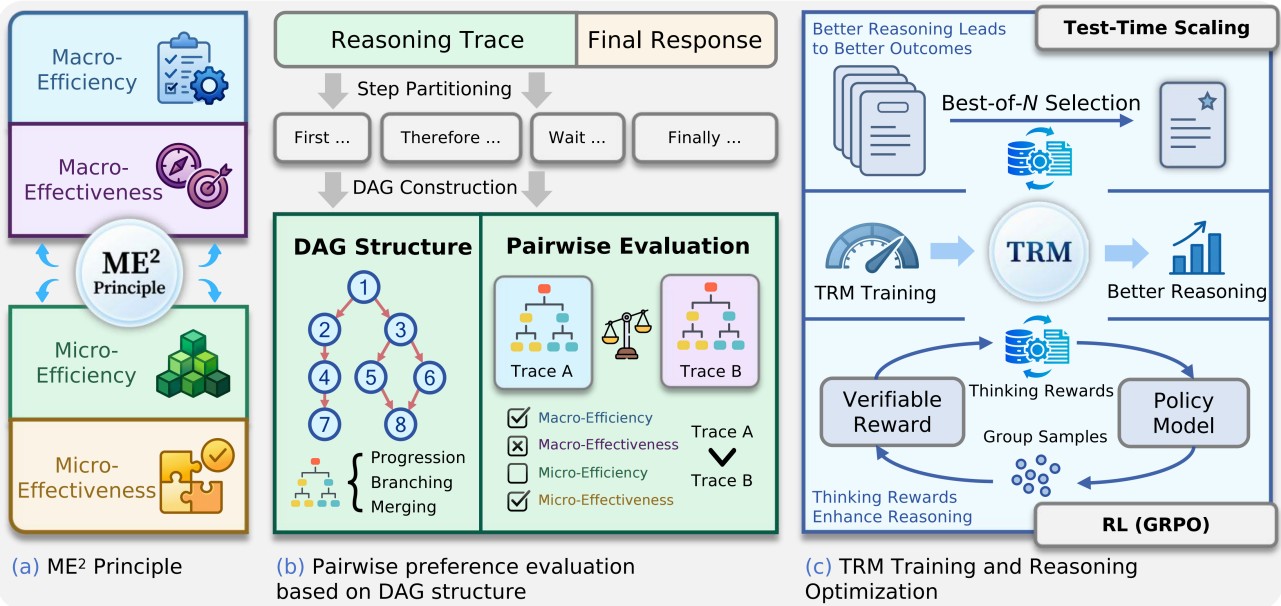

*Figure 1.* Overview of our framework. **(a)** *ME² principle* for characterizing reasoning quality (Sec. 3). **(b)** DAG-based reasoning abstraction and pairwise evaluation (Sec. 4). **(c)** TRM training, test-time scaling, and RL optimization (Sec. 5).

To address these challenges, we propose a unified framework that treats arbitrary reasoning traces as first-class optimization targets (Fig. 1). For **Q1**, we introduce ***ME² principle*** to characterize reasoning quality along two orthogonal axes: macro vs. micro (global structural organization vs. local step property) and effectiveness vs. efficiency. For **Q2**, we abstract *arbitrary* reasoning traces as directed acyclic graphs (DAGs), explicitly modeling progression, branching, and merging. For **Q3**, based on this abstraction, we construct the **TRM-Preference** dataset and train a lightweight **Thinking Reward Model (TRM)** with a Bradley–Terry objective. Crucially, TRM is trained exclusively on *verified-correct* reasoning preference pairs, decoupling reasoning quality from answer correctness and remaining orthogonal to answer-based verifiable signals provided by rule-based verifiers (Wen et al., 2025; Ma et al., 2025; Zeng et al., 2025). Compared with PRMs that often entangle step supervision with answer correctness, TRM targets scalable evaluation and optimization of reasoning quality.

We conduct comprehensive experiments to evaluate TRM from both test-time and training-time perspectives across diverse tasks. At test time, TRM enables effective test-time scaling by selecting high-quality reasoning traces, yielding gains of up to 19.3% and demonstrating that better reasoning quality leads to better outcomes. During training, incorporating thinking rewards into RL enhances reasoning and delivers gains of up to 3.9% across multiple models and tasks. These results show that reasoning quality can reliably guide both test-time selection and training-time optimization, offering an effective pathway for improving reasoning performance in large reasoning models. Our contributions:

- We provide a unified and systematic framework for characterizing, evaluating, and optimizing complex reasoning.
- We introduce ***ME² principle*** as a principled characterization of reasoning quality, and instantiate it through a DAG-based pairwise evaluation method, the **TRM-Preference** dataset, and a scalable **Thinking Reward Model (TRM)** for reasoning trace evaluation.
- We demonstrate that reasoning quality is an effective optimization signal: at test time, selecting **better reasoning leads to better outcomes** with gains of up to 19.3%; during training, **thinking rewards enhance reasoning** with gains of up to 3.9% across diverse tasks.

## 2. Related Work

**Characterizing High-quality Reasoning.** Prior work has attempted to define "good" reasoning traces, yet existing criteria are fragmented and lack a unified, high-level definition. Some work treat reasoning as a linear chain and characterize quality mainly through limited local properties (e.g., step correctness or short-range coherence) (Vacareanu et al., 2024; Jacovi et al., 2024; Zou et al., 2025), or focus primarily on efficiency (e.g., length or verbosity) (Hong et al., 2025; Xu et al., 2025). More recent studies recognize that complex reasoning exhibits rich structural signals and model traces as trees or graphs (Jiang et al., 2025; Minegishi et al., 2025; Xiong et al., 2025; Lee et al., 2025). However, these structure-based views focus on structural signals rather than explicitly defining what constitutes high-quality reasoning. They typically do not separate reasoning quality from outcome correctness, without considering step-level

local properties. In contrast, our *ME² principle* offers a unified perspective that decouples reasoning quality from outcome correctness, while jointly considering both local step properties and global structural organization.

**Reasoning Structure Modeling.** Recent work (Minegishi et al., 2025; Jiang et al., 2025) models reasoning traces as structured objects (e.g., trees or graphs) to capture non-linear behaviors. Trees are easy to construct, but cannot express complex structures such as merging, where a node has multiple predecessors (Jiang et al., 2025). Fully general graphs are more expressive, yet difficult to model and do not naturally preserve the causal order implied by step-by-step generation (Xiong et al., 2025; Li et al., 2025a). DAGs offer a practical compromise, capturing rich structure while supporting topological ordering aligned with the emitted step sequence (Lee et al., 2025; Zhang et al., 2025a). However, existing methods either rely on predefined output formats or markers, or require complex and costly pipelines to recover structure from arbitrary traces, limiting their scalability (Zhang et al., 2025a; Sultan et al., 2024; Jiang et al., 2025). To address these gaps, we propose a scalable pathway for modeling arbitrary free-form traces as DAGs that are directly usable for downstream tasks.

**Reward Modeling.** Prior work on reward modeling for language models mainly follows two paradigms. Process reward models (PRMs) assign supervision to intermediate reasoning steps (Lightman et al., 2023; Zou et al., 2025; Xia et al., 2025), but typically rely on prompt-based absolute scoring and struggle to capture complex, non-linear reasoning structures (Jiang et al., 2025; She et al., 2025). Outcome reward models (ORMs), trained with pairwise preferences, offer more stable and scalable supervision (Stiennon et al., 2020; Levtsov & Ustalov, 2025), yet are primarily designed to align final response outcomes rather than internal reasoning quality. In contrast, our TRM treats reasoning traces as first-class optimization targets, using pairwise preferences over traces to provide a reliable supervision signal that captures both structural and qualitative properties of reasoning while remaining disentangled from answer correctness.

## 3. Characterizing Reasoning Trace Quality

In this section, we introduce *ME² principle*, a concise principle for characterizing reasoning trace quality that goes beyond linear, step-wise level and accounts for both local step validity and global structural organization.

### 3.1. Overview

Modeling reasoning quality requires considerations along multiple axes. In terms of granularity, global structural organization governs how branches are introduced, explored,

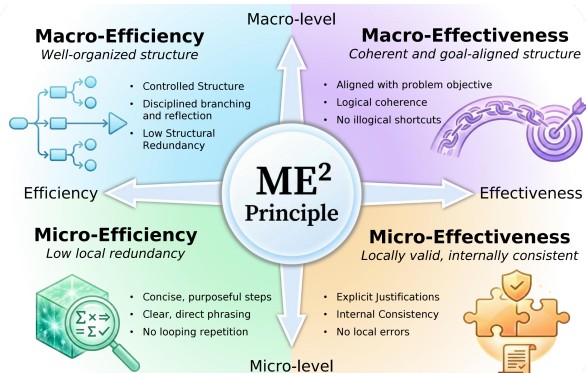

*Figure 2.* The *ME² principle*, characterizing reasoning trace quality along macro/micro granularity and efficiency/effectiveness.

and resolved (*macro-level*), while fine-grained step content determines the validity of local inferences (*micro-level*). In terms of objectives, traces often exhibit redundancy that increases inference cost and exacerbate long-context issues (*efficiency*) (Muennighoff et al., 2025; Kang et al., 2025). At the same time, a trace should make genuine progress toward a well-justified conclusion, maintaining logical coherence and staying aligned with the problem (*effectiveness*) (Huang et al., 2025; Barez et al., 2025). Motivated by these considerations, we introduce *ME² principle* as a unified characterization of reasoning trace quality that disentangles *where* quality manifests (macro vs. micro) and *what* quality entails (efficiency vs. effectiveness), as illustrated in Fig. 2.

### 3.2. *ME² principle*

**Macro-Efficiency.** *Whether the reasoning structure is well organized, avoiding unnecessary branching and reflection.* A macro-efficient trace maintains a controlled reasoning structure, introduces or closes branches only when needed, and applies reflection or verification selectively to guide subsequent steps. Macro-efficiency degrades when traces repeatedly reopen completed branches or perform redundant restarts and rechecks that fail to advance the main line of reasoning (Zhao et al., 2025; Jiang et al., 2025).

**Macro-Effectiveness.** *Whether the reasoning structure remains logically coherent and aligned with the problem objective.* A macro-effective trace remains on target with respect to the problem and exhibits a smooth, well-articulated line of reasoning across structurally adjacent steps. In contrast, macro-effectiveness degrades when branch objectives are underspecified, topics shift abruptly, or the trace makes illogical shortcuts (Barez et al., 2025). Such failures are especially common in hard problems, where superficially plausible yet invalid arguments may emerge (e.g., pseudo-proofs in challenging mathematical tasks) (Kalai et al., 2025).

**Micro-Efficiency.** *Whether individual steps avoid local redundancy.* A micro-efficient trace consists of concise, purposeful steps that perform concrete operations, such as explicit calculations or decisive validity checks, expressed in clear and direct terms. In contrast, micro-efficiency degrades when steps contain redundant statements or wording, unnecessary hedging or modifiers, or looping patterns that repeat prior content without refinement (Li et al., 2025b).

**Micro-Effectiveness.** *Whether individual steps are locally valid and internally consistent.* A micro-effective trace consists of steps that provide explicit justifications, perform coherent and well-founded computations, and use consistent notation and assumptions. In contrast, micro-effectiveness degrades when steps contain local errors, e.g., contradictions, hallucinations, unsupported claims, or incorrect arithmetic (Lightman et al., 2023; Lee & Hockenmaier, 2025).

## 4. Evaluating Reasoning Traces

In this section, based on the *ME$^2$ principle*, we propose an automated evaluation framework that supports reliable pairwise comparison and underpins the reward model training.

### 4.1. Step Partitioning

Given a raw reasoning trace $s$, we partition it into atomic reasoning steps. We start with a coarse segmentation by splitting $s$ on "\n\n", producing multiple base blocks. To obtain more stable step boundaries across heterogeneous traces, we refine this partition using step-prefix patterns. Specifically, we extract the leading tokens (e.g., the first word) of each base block and compute their frequencies over a large corpus of traces. High-frequency prefixes are then used as refined delimiters to re-partition $s$ into steps $(s_1, \ldots, s_n)$. This procedure yields consistent and semantically meaningful step boundaries across models (Jiang et al., 2025) (See App. D for details and prefix statistics).

### 4.2. Reasoning Structuring

As noted in Sec. 2, directed acyclic graphs (DAGs) offer a natural abstraction for reasoning traces. Given partitioned steps $(s_1, \ldots, s_n)$, we model the underlying reasoning structure as a DAG $G = (V, E)$, where each node $v_i \in V$ corresponds to the step $s_i$, and each directed edge $e \in E$ encodes a semantic dependency between steps. The central goal is to construct the edge set $E$, thereby transforming unstructured steps into a structured DAG. Such representation naturally expresses three canonical reasoning patterns: progression, branching, and merging, as illustrated in Fig. 3.

Since reasoning steps are generated sequentially, the observed order $(v_1, \ldots, v_n)$ can be viewed as a topological ordering of $G$ (Lee et al., 2025). We therefore construct $G$

incrementally by traversing the steps in order and, for each node $v_i$, selecting its parent nodes from earlier steps to form directed edges leading into $v_i$. Concretely, we define an attachment pool $\mathcal{P}_i \subseteq \{v_1, \ldots, v_{i-1}\}$ as the candidate parent set for $v_i$, and prompt an LLM to select $\mathrm{Parent}(v_i) \subseteq \mathcal{P}_i$ based on semantic continuity and transition cues. To avoid exposing the LLM to excessively long contexts, $\mathcal{P}_i$ includes a subset of nodes rather than the full set $\{v_1, \ldots, v_{i-1}\}$, consisting of (i) the nodes along the current main branch from the root to $v_{i-1}$ and (ii) a small set of representative branch endpoints (see App. E for construction details and App. J.1 for the prompt design). This design is sufficient in practice for recovering accurate parent relations.

Finally, to obtain a compact structure, we compress the DAG by collapsing consecutive linear chains of nodes into *super-nodes*, each representing a semantically continuous reasoning span (e.g., nodes 2 and 4 in Fig. 3 can form a super-node). The node set $V$ is replaced by these super-nodes, while the structure presented by edge set $E$ remains unchanged. This procedure is summarized in Alg. 1, with representative DAG examples shown in App. K.1 Fig. 18.

---

**Algorithm 1** DAG Construction

---

**Input:** partitioned reasoning steps $(s_1, \ldots, s_n)$
**Output:** reasoning DAG $G = (V, E)$
Initialize $V \leftarrow \{v_1\}$, $E \leftarrow \emptyset$
**for** $i = 2$ **to** $n$ **do**
    Create node $v_i$ for step $s_i$ and add it to $V$
    Construct attachment pool $\mathcal{P}_i \subseteq \{v_1, \ldots, v_{i-1}\}$
    Select parent nodes $\mathrm{Parent}(v_i) \subseteq \mathcal{P}_i$ using an LLM
    **for each** $v_j \in \mathrm{Parent}(v_i)$ **do**
        $E \leftarrow E \cup \{(v_j, v_i)\}$
    **end for**
**end for**
Collapse consecutive linear chains in $G$ into super-nodes
Replace $V$ with the resulting super-nodes

---

### 4.3. Pairwise Evaluation

Given two reasoning traces $s^{(a)}$ and $s^{(b)}$, we first extract their corresponding DAGs $G^{(a)}$ and $G^{(b)}$ as described above, and then evaluate their relative quality via pairwise comparison under the *ME$^2$ principle*. To make the comparison robust and tractable, we transform each trace into complementary abstractions that separately capture macro-level structure and micro-level step quality.

**Macro-level Abstraction.** For the DAG $G$, we construct a concise structural narrative reflecting its global organization. Let $V = \{v_1, \ldots, v_t\}$ denote the set of super-nodes. For each $v_k \in V$, we prompt an LLM to generate a short summary $a_k$ describing its intent, key intermediate conclusions, and structural role. These summaries are then linearized into

*Figure 3.* Macro- and micro-level abstractions over a reasoning DAG with three canonical structures (progression, branching, and merging). Edges are directed from top to bottom, and node indices indicate step order. *Progression*: linear continuation. *Branching*: a node expands into multiple child nodes. *Merging*: multiple parent nodes converge into a single child node.

a compact textual sequence $\mathcal{A}_{\text{macro}}(G) = (a_1, \ldots, a_t)$ using a simple traversal scheme with explicit structural annotations (details in App. J.2). This abstraction makes the global structure of the reasoning explicit, enabling macro-level assessment without being distracted by low-level verbosity.

**Micro-level Abstraction.** To evaluate local reasoning quality, we focus on the dominant path $\pi(G)$, defined as the main trajectory from the root node to the node corresponding to the final answer (details in App. J.3). Although auxiliary branches may explore alternatives or verification steps, $\pi(G)$ concentrates the sequence of steps that most directly support the final conclusion. We therefore use $\mathcal{A}_{\text{micro}}(G) = \pi(G)$ as the primary evidence for micro-level evaluation.

**Hierarchical Evaluation.** Given two traces, following *ME² principle*, we compare them at the macro abstraction $\mathcal{A}_{\text{macro}}(G)$ and the micro abstraction $\mathcal{A}_{\text{micro}}(G)$, evaluating efficiency and effectiveness at each level and producing four dimension-wise judgments $\{(y_d, r_d)\}_{d=1}^{4}$, where $y_d \in \{a \succ b, b \succ a, \text{tie}\}$ and $r_d$ is a brief natural-language rationale. Each judgment is produced by an LLM using its corresponding dimension-specific prompt (App. J.4), and the resulting judgments and rationales are aggregated by the LLM into an overall preference $(y, r)$ for downstream reward modeling, where $y \in \{a \succ b, b \succ a, \text{tie}\}$. Since pairwise judgments can be sensitive to presentation order (i.e., position bias (Shi et al., 2024)), we evaluate each pair under both original and reversed orderings to enable **robust evaluation**, repeating the comparison multiple times and retaining only consistent, non-tied labels.

# 5. Optimizing Reasoning with Thinking Rewards

In this section, we show how the proposed *ME² principle* and the resulting Thinking Reward Model (TRM) can be used to optimize reasoning in practice. We first describe how we construct pairwise preference data and train TRM, and then demonstrate how TRM supports both test-time scaling and reinforcement learning, enabling models to internalize

higher-quality reasoning behaviors.

## 5.1. Reward Modeling

**Dataset Construction.** We construct the **TRM-Preference** dataset from WebInstruct-verified (Ma et al., 2025), which contains verifiable STEM and math problems requiring non-trivial multi-step reasoning. We sample 64K prompts and generate candidate reasoning traces[1] using multiple open-source reasoning model families, including Qwen3 (Yang et al., 2025), DeepSeek-Distill (Guo et al., 2025), and GPT-OSS (Agarwal et al., 2025). To decouple reasoning quality from answer correctness, *we retain only traces whose final answers are verified as correct*, as correctness can be reliably assessed by rule-based verifiers, allowing the learned reward signal to focus on reasoning quality rather than duplicating answer-based supervision.

For each prompt, we generate multiple candidate reasoning traces and apply the evaluation pipeline described in Sec. 4 to construct reasoning structures and perform pairwise preference judgments under the *ME² principle*, using DeepSeek-V3.2 (Liu et al., 2025) as an automated evaluator. To avoid over-representation of individual prompts, we sample at most four trace pairs per prompt. This process yields 103K training pairs and a held-out validation set of 1.5K pairs, forming the final **TRM-Preference** dataset. Additional details are provided in App. F, with illustrative case studies of the constructed DAGs shown in App. K.1 Fig. 18. We further report preference-label reliability validation and offline construction cost in App. G.4 and App. G.5.

**TRM Training.** We initialize TRM from Llama-3.1-8B-Instruct (Grattafiori et al., 2024) by replacing the language modeling head with a scalar value head, and train it on the TRM-Preference dataset using the Bradley–Terry loss (Bradley & Terry, 1952), following standard preference modeling practice (Stiennon et al., 2020; Kirk et al., 2023). Training details and dynamics are provided in App. G.1, and the Bradley–Terry formulation is summarized in App. I.

---

[1] A reasoning trace refers to the generated reasoning content before the reasoning-termination marker (e.g., `</think>`).

*Table 1.* Pairwise preference evaluation on the validation set. $\sqrt{}$, $\sim$, and $\times$ denote correct, tied, and incorrect judgments, respectively. Accuracy is computed by counting ties as incorrect.

| METHOD | $\sqrt{}$ | $\sim$ | $\times$ | ACCURACY |
|---|---|---|---|---|
| Qwen2.5-Math-PRM-7B | 693 | 0 | 804 | 46.3% |
| ReasonFlux-PRM-7B | 935 | 0 | 562 | 62.5% |
| PromptOnly | 1176 | 232 | 89 | 78.6% |
| TRM (ours) | 1326 | 0 | 171 | 88.6% |

### 5.2. Validation Set Analysis

**Evaluation Protocol and Results.** We evaluate TRM on the validation set against three baselines: **ReasonFlux-PRM-7B** (Zou et al., 2025), **Qwen2.5-Math-PRM-7B** (Zhang et al., 2025b), and **PromptOnly**. PromptOnly applies our pairwise evaluation prompts (App. J.4) with DeepSeek-V3.2 (Liu et al., 2025) to raw, unstructured traces, using the same robust protocol in Sec. 4.3 to reduce position bias. As shown in Tab. 1, TRM achieves the highest accuracy, as expected. ReasonFlux-PRM-7B captures part of the preference signal (62.5%), likely due to its focus on local step-level effectiveness (see App. C.1 for further discussion), while Qwen2.5-Math-PRM-7B performs close to random, consistent with its limited modeling of long-range and structural reasoning observed in Zou et al. (2025).

**Limitations of Direct Prompt-Based Evaluation.** The prompt-based DeepSeek-V3.2 evaluator ranks second but produces a large number of ties (232); when ties are excluded, its accuracy reaches 93%, indicating that our constructed preference data is reliable and that non-tied pairs typically exhibit clear quality differences. However, the high tie rate exposes a fundamental limitation of direct prompt-based evaluation: many reasoning trace pairs differ primarily in how reasoning steps are organized and consolidated, rather than in local step content. Such differences (e.g., redundant branching or illogical shortcuts, as characterized by $ME^2$ *principle*) are weakly expressed in flat, unstructured traces and are therefore difficult to distinguish via direct prompt-based comparison. By explicitly modeling reasoning structure, our DAG-based abstractions (Sec. 4) surface these signals and enable more fine-grained and discriminative assessment aligned with the $ME^2$ *principle*.

### 5.3. TRM-Guided Test-Time Scaling

**Settings.** We evaluate TRM in test-time scaling (Muennighoff et al., 2025) with Best-of-$N$ selection, where rewards are used to select the highest-scoring output from generated traces. Under this setup, TRM favors traces that best aligns with the $ME^2$ *principle*. We compare TRM against representative PRMs, including ReasonFlux-PRM-7B (Zou et al., 2025) and Qwen2.5-Math-PRM-7B (Zhang et al., 2025b), using GPT-OSS-20B (Agarwal et al., 2025)

and Qwen3-8B (Yang et al., 2025) as response models. Experiments are conducted on AIME24 and AIME25, with each setting repeated five times for robustness. Additional details and results on larger models are provided in App. G.2 and App. G.6.

**From Better Reasoning to Better Outcomes.** Although TRM models only reasoning trace quality without considering user-facing responses, and is trained exclusively on verified-correct traces without any explicit correctness supervision, we still observe a clear correspondence between better reasoning and better final outcomes. As shown in Fig. 4, selecting traces that are more aligned with the $ME^2$ *principle* consistently improves Best-of-$N$ accuracy as $N$ increases. In contrast, existing PRMs such as ReasonFlux-PRM-7B and Qwen2.5-Math-7B-PRM explicitly incorporate correctness-oriented supervision during training (Zou et al., 2025; Zhang et al., 2025b). Despite this difference, TRM achieves comparable or even superior performance, with gains of up to 19.3% (e.g., from 44.7% at $N = 1$ to 64.0% at $N = 16$ on AIME24 with Qwen3-8B), demonstrating that *better reasoning, as measured by closer alignment with the $ME^2$ principle, leads to better final outcomes*.

### 5.4. TRM-Guided RL Optimization

**Thinking Reward Signals.** To explore how TRM improves reasoning quality during training, we adopt reinforcement learning with verifiable rewards (RLVR) (Guo et al., 2025; Zeng et al., 2025) and integrate TRM as an auxiliary reward signal within Group Relative Policy Optimization (GRPO) (Shao et al., 2024). GRPO optimizes a binary verifiable reward $r_v \in \{0, 1\}$ that enforces outcome correctness, treating all reasoning traces that yield the same answer as equivalent. In contrast, TRM provides a thinking reward $r_t$ that evaluates the quality of the underlying reasoning trace. Since it is trained exclusively on verified-correct traces, $r_t$ is used to rank and shape alternative correct reasoning paths. We combine these two signals via gated reward shaping:

$$r = r_v \cdot (1 - \alpha + \alpha \cdot \text{Sigmoid}(r_t)), \quad (1)$$

where $\alpha$ balances outcome supervision and reasoning-quality shaping, and the sigmoid maps the unbounded thinking reward to $(0, 1)$ for stability. For a group of $G$ sampled responses $\{o_i\}_{i=1}^{G}$, we compute the group advantage $\hat{A}_i = \frac{r_i - \text{mean}(\{r_i\}_{i=1}^{G})}{\text{std}(\{r_i\}_{i=1}^{G})}$. Then, the GRPO objective is

$$\mathcal{J}_{\text{GRPO}}(\theta) = \mathbb{E}_{q \sim \mathcal{D}, \{o_i\}_{i=1}^{G} \sim \pi_{\theta_{\text{old}}}(\cdot|q)}$$
$$\left[ \frac{1}{G} \sum_{i=1}^{G} \frac{1}{|o_i|} \sum_{t=1}^{|o_i|} \left( f_{i,t}(\theta) - \beta D_{\text{KL}}(\pi_\theta \| \pi_{\text{ref}}) \right) \right]. \quad (2)$$

where $f_{i,t}(\theta) = \min(\hat{r}_{i,t}(\theta)\hat{A}_i, \text{clip}(\hat{r}_{i,t}(\theta), 1 - \epsilon, 1 + \epsilon)\hat{A}_i)$, and $\hat{r}_{i,t}(\theta) = \frac{\pi_\theta(o_{i,t}|q, o_{i,<t})}{\pi_{\theta_{\text{old}}}(o_{i,t}|q, o_{i,<t})}$. Note that thinking rewards

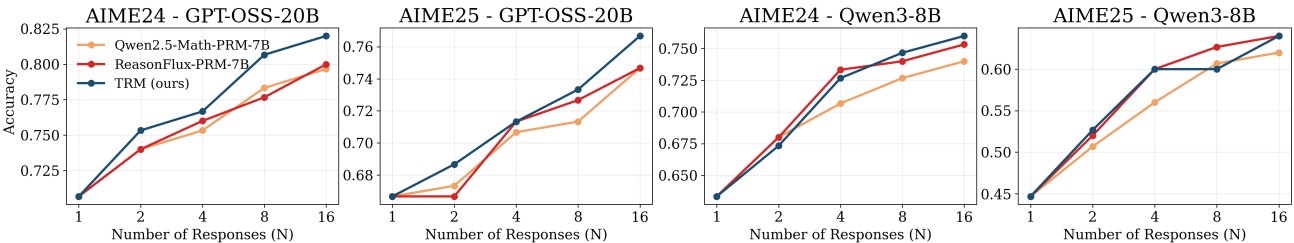

*Figure 4.* Best-of-$N$ test-time scaling results on AIME24 and AIME25, comparing Qwen2.5-Math-PRM-7B, ReasonFlux-PRM-7B, and our TRM. The first two and last two panels use GPT-OSS-20B and Qwen3-8B as the response models, respectively.

*Table 2.* Main results across multiple benchmarks. BaseModel denotes the policy model before training. Verifier denotes training with only the verifiable reward $r_v$. ReasonFlux and TRM denote training with ReasonFlux-PRM-7B and our TRM, respectively.

| Models | STEM Performance (%) | | | | | | Math Performance (%) | | | | | |
|---|---|---|---|---|---|---|---|---|---|---|---|---|
| | GPQA | SuperGPQA | MMLU-Pro | BBEH | FS | Avg. | AMC | AIME24 | AIME25 | MATH500 | Olympiad | Avg. |
| **Qwen2.5-Math-1.5B** | | | | | | | | | | | | |
| BaseModel | 10.1 | 13.2 | 19.9 | 1.3 | 7.7 | 10.4 | 22.2 | 2.7 | 0.6 | 32.0 | 15.4 | 14.6 |
| Verifier | 21.7 | 17.0 | 31.4 | 6.2 | 10.4 | 17.3$_{+6.9}$ | 43.8 | 7.9 | 4.3 | 69.0 | **37.5** | 32.5$_{+17.9}$ |
| ReasonFlux | 23.7 | **17.6** | 32.0 | 5.2 | 9.1 | 17.5$_{+7.1}$ | 45.0 | 10.0 | 4.9 | 71.4 | 35.0 | 33.3$_{+18.7}$ |
| TRM (ours) | **27.3** | 17.3 | **33.0** | 6.8 | **12.6** | **19.4**$_{+9.0}$ | **48.8** | **11.4** | **7.0** | **74.2** | 37.3 | **35.7**$_{+21.1}$ |
| **Qwen2.5-Math-7B** | | | | | | | | | | | | |
| BaseModel | 22.7 | 19.8 | 38.6 | 6.1 | 14.4 | 20.3 | 50.8 | 12.8 | 7.7 | 69.0 | 32.4 | 34.5 |
| Verifier | 38.9 | 25.4 | 47.5 | 8.6 | 19.2 | 27.9$_{+7.6}$ | 62.0 | 16.8 | 13.7 | 81.6 | 46.1 | 44.0$_{+9.5}$ |
| ReasonFlux | 41.9 | 25.6 | 49.4 | **9.5** | 23.9 | 30.1$_{+9.8}$ | 61.9 | 17.1 | 13.2 | **83.6** | 46.5 | 44.5$_{+10.0}$ |
| TRM (ours) | **42.4** | **26.7** | **50.5** | 9.2 | **26.5** | **31.1**$_{+10.8}$ | **63.4** | **19.1** | **17.1** | 83.0 | **47.4** | **46.0**$_{+11.5}$ |
| **Llama-3.1-8B-Instruct** | | | | | | | | | | | | |
| BaseModel | 16.2 | 22.1 | 43.6 | 8.6 | 20.3 | 22.2 | 19.4 | 3.9 | 0.2 | 47.6 | 10.4 | 16.3 |
| Verifier | 29.3 | 24.9 | 47.5 | 12.8 | 25.2 | 27.9$_{+5.7}$ | 24.7 | 5.1 | 1.2 | 52.6 | 18.4 | 20.4$_{+4.1}$ |
| ReasonFlux | 31.8 | **27.3** | 50.9 | 13.2 | 26.0 | 29.8$_{+7.6}$ | 29.3 | **8.8** | 1.4 | **55.0** | 21.5 | 23.2$_{+6.9}$ |
| TRM (ours) | **36.4** | 26.7 | **52.6** | **14.0** | **29.1** | **31.8**$_{+9.6}$ | **32.8** | 8.1 | **2.4** | 53.8 | **23.9** | **24.2**$_{+7.9}$ |

can be readily integrated into other online policy optimization algorithms (Schulman et al., 2017; Hu, 2025).

**Theoretical Guarantees.** We provide two complementary theoretical analysis to justify TRM-guided reward shaping.

**Theorem 5.1** (Optimal Policy Invariance of Eq. 1). *Consider the RLVR objective with a binary verifiable reward $r_v \in \{0, 1\}$ and the gated reward $r$ defined in Eq. 1. Optimizing Eq. 1 preserves the set of optimal policies of the original RLVR objective defined solely by $r_v$.*

A formal statement and proof are provided in App. A, Theorem 1. This shows that the gated reward reshapes the learning signal without changing the set of optimal policies.

**Theorem 5.2** (Lower Bound on Policy Improvement). *Consider one natural policy gradient update with step size $\beta \ll 1$. Define $V^\pi(s)$ as state-value function under policy $\pi$, $D_{TRM}(\pi_t) = \mathbb{E}_{s \sim \rho} \mathbb{V}_{a \sim \pi_t} [\alpha A^{TRM}(s, a)]$ and $A_{TRM}(\pi_t) = \mathbb{E}_{s \sim \rho} \mathbb{E}_{a \sim \pi_t} [\alpha A^{TRM}(s, a) A^{\pi_t}(s, a)]$. There exists a constant $C > 0$ such that the expected policy improvement satisfies*

$$\mathbb{E}_{s \sim \rho} [V^{\pi_{t+1}}(s) - V^{\pi_t}(s)] \geq C\beta \cdot (D_{TRM}(\pi_t) + A_{TRM}(\pi_t)).$$

A formal statement, proof, and analysis are provided in

App. B. $D_{TRM}(\pi_t)$ shows that policy improvement is driven by reward variance, where verifier rewards fail to distinguish answer-correct solutions and TRM provides subtle signals to differentiate verified-correct traces (e.g., efficient vs. redundant), mitigating the signal plateau induced by binary rewards. $A_{TRM}(\pi_t)$ captures the alignment between TRM and policy advantages, ensuring that learning remains consistent with the primary objective of accuracy. This clarifies the role of TRM in providing richer optimization signals that bias learning toward $ME^2$ *principle* among correct traces.

**Settings.** Based on the WebInstruct-Verified dataset (Ma et al., 2025), we train three models with varying reasoning capabilities: Qwen2.5-Math-1.5B, Qwen2.5-Math-7B (Yang et al., 2024), and Llama-3.1-8B-Instruct (Grattafiori et al., 2024). As baselines, we consider two reward signals: (1) **Verifier**, which relies solely on the verifiable reward $r_v$ provided by a verifier, and (2) **ReasonFlux-PRM-7B**, a strong PRM that evaluates reasoning trace quality at the step level using the same reward formulation in Eq. 1. The weighting parameter $\alpha$ is fixed to 0.2. Additional experimental details and training settings are provided in App. G.3.

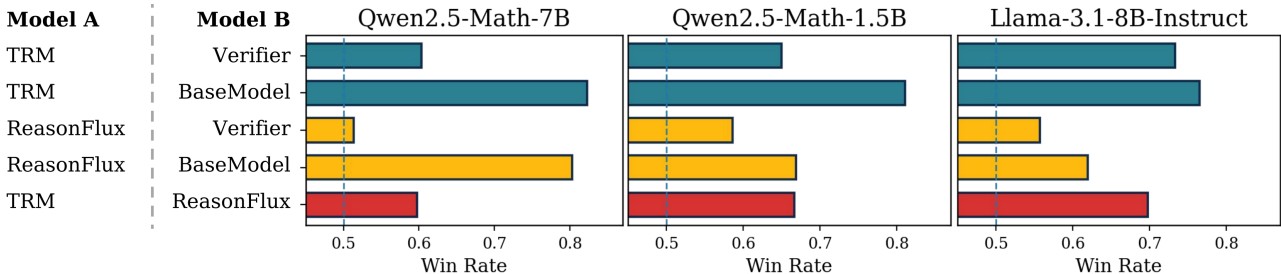

*Figure 5.* Pairwise win rates (ties excluded) across different **policy models**. TRM, Verifier, and ReasonFlux denote policies trained with our TRM, the verifiable reward $r_v$, and ReasonFlux-PRM-7B, respectively. Each bar reports the win rate of Model A against Model B.

**Benchmarks.** We use 10 representative and challenging benchmarks covering diverse domains and difficulty levels. For general reasoning, we use GPQA-Diamond (Rein et al., 2023), SuperGPQA (Du et al., 2025), MMLU-Pro (Wang et al., 2024), BBEH (Kazemi et al., 2025), and FrontierScience (FS) (OpenAI, 2025). For mathematical reasoning, we use AMC, AIME24, AIME25, MATH500 (Hendrycks et al., 2021), and Olympiad (He et al., 2024), using AVG@32 on AIME24 and AIME25 for robustness.

**From Thinking Rewards to Training Gains.** As reported in Tab. 2, TRM outperforms both the Verifier and ReasonFlux-PRM-7B. Compared to the Verifier baseline, TRM improves performance by approximately 2%–4%, with the largest gains on Llama-3.1-8B-Instruct (3.9% on STEM and 3.8% on Math). TRM also surpasses ReasonFlux-PRM-7B, with particularly strong improvements on challenging benchmarks such as GPQA, AIME24, and AIME25, indicating that thinking rewards provide an effective training signal beyond verifiable reward. The ablation study in App. H further supports the choice of $\alpha = 0.2$.

## 6. Analysis

### 6.1. TRM Improves Reasoning Quality

In this section, we provide empirical comparisons and case studies to investigate how TRM improves reasoning quality.

**Empirical Comparison.** We evaluate reasoning quality by applying the checkpoints trained in Sec. 5.4 to AIME24, AIME25, and GPQA-Diamond, and conduct pairwise evaluation following Sec. 4 with DeepSeek-V3.2 as the evaluator across the four dimensions of $ME^2$ principle. As shown in Fig. 5, TRM-trained policies consistently outperform the BaseModel, Verifier-trained, and ReasonFlux-trained policies in reasoning quality. These improvements indicate that distilling $ME^2$ principle into TRM and using it as a supervision signal during training effectively enhances the quality of reasoning produced by the policy model.

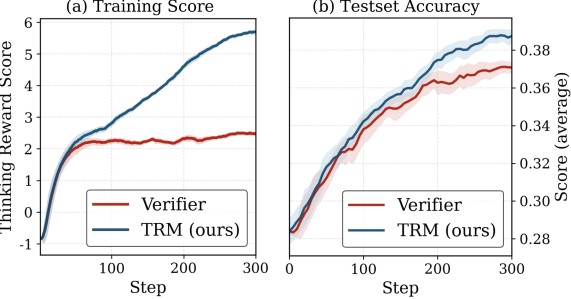

*Figure 6.* Training dynamics of Qwen2.5-Math-7B: (a) TRM scores and (b) test accuracy under verifier-guided and TRM-guided RL optimization. Test accuracy is evaluated on tasks with rule-based verifiers (i.e., excluding FrontierScience (OpenAI, 2025)).

*Table 3.* Computational cost in GPU hours (Qwen2.5-Math-7B).

| Method | Verifier | ReasonFlux | TRM (ours) |
|---|---|---|---|
| GPU Hours | $44 \times 4$ | $52 \times 4$ | $49 \times 4$ |

**Case Study.** We further provide qualitative case studies to explore how thinking rewards relate to reasoning quality in practice. In App. K.2 and App. K.3, we compare reasoning traces generated under identical prompts by a TRM-trained policy and a baseline, and observe that the TRM-trained policy follows a more disciplined reasoning trajectory that is more closely aligned with $ME^2$ principle. In App. K.4 and App. K.5, we analyze two reasoning traces scored by our TRM and find that the higher-scoring trace exhibits clearer structure and greater efficiency under the proposed $ME^2$ principle. Please refer to App. K.2–K.5 for details.

### 6.2. Training Dynamics and Computational Cost

Fig. 6 illustrates the training dynamics of Qwen2.5-Math-7B under verifier- and TRM-guided reinforcement learning. In Fig. 6(a), TRM-guided RL yields a steadily increasing thinking reward throughout training. In contrast, verifier-guided RL, despite showing an early improvement in reasoning quality, quickly plateaus, with reasoning behaviors converging in the mid-to-late stages. In Fig. 6(b), TRM-guided

RL converges faster and achieves higher test accuracy than verifier-guided RL. Besides, Tab. 3 reports the corresponding training cost in GPU hours. The additional overhead introduced by TRM is modest compared to verifier-guided RL (49 vs. 44) and remains lower than ReasonFlux, which requires two forward passes per step to compute full reward signals (Zou et al., 2025).

## 7. Conclusion

In this work, we present a unified framework for characterizing, evaluating, and optimizing complex reasoning. We introduce *ME²* *principle* as a principled characterization of reasoning quality, and instantiate it through a scalable DAG-based abstraction and evaluation pipeline. Building on this foundation, we construct the TRM-Preference dataset and train a Thinking Reward Model for reasoning evaluation. Extensive experiments demonstrate that thinking rewards are an effective optimization signal: at test time, selecting better reasoning leads to better outcomes; during training, thinking rewards enhance reasoning with improvements across diverse models and tasks. These results establish reasoning quality as a first-class optimization target and offer a practical pathway for improving the efficiency and effectiveness of large reasoning models.

## Acknowledgements

We extend our gratitude to all the reviewers for their valuable feedback and suggestions. This work is supported by the School of Artificial Intelligence, Shanghai Jiao Tong University and the Shanghai Artificial Intelligence Laboratory.

## Impact Statement

This paper presents work whose goal is to advance the field of Machine Learning. There are many potential societal consequences of our work, none which we feel must be specifically highlighted here.

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

# Appendix Contents

This appendix provides supplementary theoretical analyses, methodological details, experimental settings, and qualitative case studies that support and extend the main paper.

## A. Optimal Policy Invariance of TRM-Guided Reward Shaping

Ensuring the optimal policy invariance requires a potential-based reward shaping scheme (Ng et al., 1999; Wang et al., 2023; Müller & Kudenko, 2025). The original reward shaping scheme is:

$$r = r_v \cdot (1 - \alpha + \alpha \cdot \text{Sigmoid}(r_t)). \tag{3}$$

Since $r_v \in \{0, 1\}$, the shaping scheme in Eq. 3 can be converted to another form:

$$r = (1 - \alpha) \cdot r_v + \alpha \cdot \mathbb{I}(r_v = 1) \cdot \text{Sigmoid}(r_t). \tag{4}$$

Let $\Phi(\cdot) : S \to \mathbb{R}$ denote a real-valued potential function over the state space $s \in S$, i.e., $\Phi(s)$ is the potential of state $s$. We formulate the shaping reward $R_s$ as the difference between the potentials of adjacent states as

$$R_s(s_t, a_t, s_{t+1}) = \gamma \Phi(s_{t+1}) - \Phi(s_t). \tag{5}$$

In the LLM reasoning setting, the states and action within a given query-response pair are defined as $s_t := [q, o_{i, \leq t}]$, $a_t := o_{i, t+1}$, and $s_{t+1} := [q, o_{i, \leq t+1}]$. We can take the potential function to be the thinking bonus $\text{TRM}(\cdot)$ for any intermediate state $s_t$ of the complete reasoning chain:

$$\Phi(s_t) := \mathbb{I}(r_v = 1) \cdot \text{TRM}(s_t). \tag{6}$$

The shaping reward becomes:

$$R_s\left([q, o_{i, \leq t}], o_{i, t+1}, [q, o_{i, \leq t+1}]\right) = \mathbb{I}(r_v = 1) \cdot \left(\gamma \text{TRM}([q, o_{i, \leq t+1}]) - \text{TRM}([q, o_{i, \leq t}])\right). \tag{7}$$

In GRPO settings, policy gradients are derived at the sequence level. Then, the shaping reward for a complete query-response pair is calculated as

$$\begin{aligned}
R_s\left([q, o_i]\right) &= \sum_{t=0}^{T-1} \gamma^t R_s\left([q, o_{i, \leq t}], o_{i, t+1}, [q, o_{i, \leq t+1}]\right) \\
&= \sum_{t=0}^{T-1} \gamma^t \mathbb{I}(r_v = 1)[\gamma \text{TRM}([q, o_{i, \leq t+1}]) - \text{TRM}([q, o_{i, \leq t}])] \\
&= \gamma^T \cdot \mathbb{I}(r_v = 1) \cdot \text{TRM}([q, o_{i, \leq T}]) - \mathbb{I}(r_v = 1) \cdot \text{TRM}(q) \\
&= \gamma^T \cdot \mathbb{I}(r_v = 1) \cdot \text{TRM}([q, o_i]),
\end{aligned} \tag{8}$$

where the think reward bonus is zero for the query $q$, i.e., $\text{TRM}(q) = 0$ That is, the shaping reward for a complete query-response pair is derived as the thinking reward bonus of the final response. This formulation avoids the need to calculate the thinking reward bonus of any intermediate sentences.

Finally, we use the new reward function as

$$\begin{aligned}
r &= (1 - \alpha) \cdot r_v + \alpha \cdot R_s([q, o_i]) \\
&= (1 - \alpha) \cdot r_v + \alpha \cdot \gamma^T \cdot \mathbb{I}(r_v = 1) \cdot \text{TRM}([q, o_i]) \\
&= (1 - \alpha) \cdot r_v + \alpha \cdot \gamma^T \cdot \mathbb{I}(r_v = 1) \cdot \text{Sigmoid}(r_t).
\end{aligned} \tag{9}$$

That is the reason why we change the original reward in Eq. 3 to a new form as in Eq. 4.

**Theorem 1** (Optimal Policy Invariance). *Let $M = (S, A, T, R, \gamma)$ denote the MDP for the LLM reasoning task. $\text{TRM}(\cdot):$ $S \mapsto \mathbb{R}$ is a real-valued function that computes the thinking reward bonus $\text{TRM}(s)$ of the state $s$ within a group of responses. We formulate $R_s(\cdot): S \times A \times S \mapsto \mathbb{R}$ as a shaping reward function that is the difference between thinking reward bonuses of two adjacent states, such that for all $s \in S, a \in A, s' \in S, R_s(s, a, s') = \gamma \text{TRM}(s') - \text{TRM}(s)$. Then, with any constant balancing ratio $\alpha$, every optimal policy in the transformed MDP $M' = (S, A, T, R + \alpha R_s, \gamma)$ will also be an optimal policy in $M$, and vice versa.*

*Proof.* For the original MDP $M$, we know that its optimal Q-function $Q_M^*$ satisfies the Bellman optimality equation (Sutton & Barto, 2018):

$$Q_M^*(s,a) = \mathbb{E}_{s'}\left[R(s,a,s') + \gamma\max_{a'\in A}Q_M^*(s',a')\right]. \tag{10}$$

With some simple algebraic manipulation, we can get:

$$Q_M^*(s,a) - \alpha\text{TRM}(s) = \mathbb{E}_{s'}\left[R(s,a,s') + \alpha\Big(\gamma\text{TRM}(s') - \text{TRM}(s)\Big) + \gamma\max_{a'\in A}\Big(Q_M^*(s',a') - \alpha\text{TRM}(s')\Big)\right]. \tag{11}$$

If we now define $\hat{Q}_{M'}(s,a) \triangleq Q_M^*(s,a) - \alpha\text{TRM}(s)$ and substitute that and $R_s(s,a,s') = \gamma\text{TRM}(s') - \text{TRM}(s)$ into the previous equation, we can get:

$$\begin{aligned}
\hat{Q}_{M'}(s,a) &= \mathbb{E}_{s'}\left[R(s,a,s') + \alpha R_s(s,a,s') + \gamma\max_{a'\in A}\hat{Q}_{M'}(s',a')\right] \\
&= \mathbb{E}_{s'}\left[R'(s,a,s') + \gamma\max_{a'\in A}\hat{Q}_{M'}(s',a')\right],
\end{aligned} \tag{12}$$

which is exactly the Bellman optimality equation for the transformed MDP $M'$, where $R' = R + \alpha R_s$ is the reward function for $M'$. Thus, $Q_{M'}^*(s,a) = \hat{Q}_{M'}(s,a) = Q_M^*(s,a) - \alpha\text{TRM}(s)$, and the optimal policy for $M'$ therefore satisfies:

$$\begin{aligned}
\pi_{M'}^*(s) &= \arg\max_{a\in A}Q_{M'}^*(s,a) \\
&= \arg\max_{a\in A}\Big[Q_M^*(s,a) - \alpha\text{TRM}(s)\Big] \\
&= \arg\max_{a\in A}Q_M^*(s,a),
\end{aligned} \tag{13}$$

and is therefore also optimal in $M$. To show every optimal policy in $M$ is also optimal in $M'$, simply apply the same proof with the roles of $M$ and $M'$ interchanged (and using $-R_s$ as the shaping reward). This completes the proof. $\square$

## B. Lower Bound on Policy Improvement

**Theorem 2** (Lower Bound on Policy Improvement). *For the base policy iterate $\pi_t$, after one update step with learning rate $\beta \ll 1$, there exists a constant $C > 0$, such that over state distribution $\rho$ the following holds:*

$$\mathbb{E}_{s\sim\rho}[V^{\pi_{t+1}}(s) - V^{\pi_t}(s)] \geq C\beta\cdot\underbrace{\mathbb{E}_{s\sim\rho}\mathbb{V}_{a\sim\pi_t}[\alpha A^{TRM}(s,a)]}_{\text{distinguishability from TRM}} + C\beta\cdot\underbrace{\mathbb{E}_{s\sim\rho}\mathbb{E}_{a\sim\pi_t}[\alpha A^{TRM}(s,a)A^{\pi_t}(s,a)]}_{\text{alignment between }\pi_t\text{ and TRM}} \tag{14}$$

The first term in the lower bound formalizes that policy improvement is directly proportional to the variance of the reward signal. If the reward signal cannot distinguish between different actions or trajectories (low variance), the policy update becomes stagnant or inefficient. In standard RL for reasoning (like RLVR), all correct traces receive a reward of $1$, and all incorrect ones receive $0$. This creates a *signal plateau* for correct answers, that is, the model cannot distinguish a high-quality, efficient proof from a redundant, *hallucinatory* but ultimately correct one.

In our method, we include a think reward $r_t$ that can introduce variance among successful trajectories. The theorem suggests that this increased distinguishability allows the model to more effectively push the model toward the most efficient and effective reasoning patterns under the proposed *ME² principle*, rather than any path that hits the right answer.

The second term in the lower bound indicates that our TRM should not be too misaligned with the base policy $\pi$; otherwise, $\mathbb{E}[\langle\alpha A^{TRM}, A^{\pi_t}\rangle]$ would be too negative. Since TRM rewards are only added to correct solutions, our method naturally follows the theoretical requirement of not being misaligned with the primary goal of accuracy. By distinguishing valid pathways without incentivizing incorrect solutions, the TRM $r_t$ functions as an auxiliary advantage $A^{TRM}$ that is congruent with the primary advantage $A^{\pi_t}$. This alignment drives the policy toward more advantageous paths among those leading to the truth, guided by the ME² principles.

## B.1. Notations

A (finite) Markov Decision Process (MDP) $M = (\mathcal{S}, \mathcal{A}, P, r, \gamma, \rho)$ is specified by: a finite state space $\mathcal{S}$; a finite action space $\mathcal{A}$; a transition model $P$ where $P(s' \mid s, a)$ is the probability of transitioning into state $s'$ upon taking action $a$ in state $s$; a reward function $r : \mathcal{S} \times \mathcal{A} \to [0, 1]$ where $r(s, a)$ is the immediate reward associated with taking action $a$ in state $s$; a discount factor $\gamma \in [0, 1)$; a starting state distribution $\rho$ over $\mathcal{S}$. The agent chooses actions according to a stochastic policy $\pi : \mathcal{S} \to \Delta(\mathcal{A})$ (where $\Delta(\mathcal{A})$ is the probability simplex over $\mathcal{A}$), and, overloading notation, we write $a_t \sim \pi(\cdot \mid s_t)$.

A policy induces a distribution over trajectories $\tau = (s_t, a_t, r_t)_{t=0}^{\infty}$, where $s_0$ is drawn from the starting state distribution $\rho$, and, for all subsequent timesteps $t$, $a_t \sim \pi(\cdot \mid s_t)$ and $s_{t+1} \sim P(\cdot \mid s_t, a_t)$. The value function $V^\pi : \mathcal{S} \to \mathbb{R}$ is defined as the discounted sum of future rewards starting at state $s$ and executing $\pi$, i.e.

$$V^\pi(s) := \mathbb{E}_{\tau \sim \pi} \left[ \sum_{t=0}^{\infty} \gamma^t r(s_t, a_t) \mid \pi, s_0 = s \right], \tag{15}$$

where the expectation is with respect to the randomness of the trajectory $\tau$ induced by $\pi$ in $M$. The action-value (or Q-value) function $Q^\pi : \mathcal{S} \times \mathcal{A} \to \mathbb{R}$ and the *advantage function* $A^\pi : \mathcal{S} \times \mathcal{A} \to \mathbb{R}$ are defined as:

$$Q^\pi(s, a) = \mathbb{E} \left[ \sum_{t=0}^{\infty} \gamma^t r(s_t, a_t) \mid \pi, s_0 = s, a_0 = a \right], \quad A^\pi(s, a) := Q^\pi(s, a) - V^\pi(s). \tag{16}$$

It is useful to define the discounted state visitation distribution $d_{s_0}^\pi$ of a policy $\pi$ as:

$$d_{s_0}^\pi(s) := (1 - \gamma) \sum_{t=0}^{\infty} \gamma^t \Pr^\pi(s_t = s \mid s_0), \tag{17}$$

where $\Pr^\pi(s_t = s \mid s_0)$ is the state visitation probability that $s_t = s$, after we execute $\pi$ starting at state $s_0$. Again, we overload notation and write:

$$d_\rho^\pi(s) = \mathbb{E}_{s_0 \sim \rho} \left[ d_{s_0}^\pi(s) \right], \tag{18}$$

where $d_\rho^\pi$ is the discounted state visitation distribution under initial state distribution $\rho$.

## B.2. Problem Formulation and Preliminary Lemmas

We present the policy improvement bound where the base policy is softmax parameterized and is updated with natural policy gradient (NPG) (Kakade, 2001). The language policy is implemented as

$$\pi_\theta(a|s) = \frac{\exp(\theta_{s,a})}{\sum_{a' \in \mathcal{A}} \exp(\theta_{s,a'})}, \tag{19}$$

where $\theta \in \mathbb{R}^{|\mathcal{S}| \times |\mathcal{A}|}$ is a class of parameters that controls the probability of taking action $a$ at state $s$. By incorporating the think reward bonus $\text{TRM}(\cdot)$, the RL objective of our method is

$$J(\pi_\theta) = \frac{1}{1 - \gamma} \mathbb{E}_{s \sim d_{s_0}^{\pi_\theta}} \mathbb{E}_{a \sim \pi_\theta(\cdot|s)} \left[ A^{\pi_\theta}(s, a) + \alpha A^{\text{TRM}}(s, a) \right] \tag{20}$$

Following policy gradient derivations (Sutton et al., 1999), we get:

$$\nabla_\theta J(\pi_\theta) = \frac{1}{1 - \gamma} \mathbb{E}_{s \sim d_{s_0}^{\pi_\theta}} \mathbb{E}_{a \sim \pi_\theta(\cdot|s)} \left[ \nabla_\theta \log \pi_\theta(a|s) \cdot \left( A^{\pi_\theta}(s, a) + \alpha A^{\text{TRM}}(s, a) \right) \right]. \tag{21}$$

The NPG algorithm defines a Fisher information matrix (FIM) induced by the gradient of the policy $\pi$:

$$F_\rho(\pi_\theta) = \mathbb{E}_{s \sim d_\rho^{\pi_\theta}} \mathbb{E}_{a \sim \pi_\theta(\cdot|s)} \left[ \nabla_\theta \log \pi_\theta(a|s) \cdot (\nabla_\theta \log \pi_\theta(a|s))^\top \right]. \tag{22}$$

The policy is updated according to the NPG rule w.r.t. the objective in Eq. 20:

$$
\begin{aligned}
\pi_{\theta_{t+1}} &= \pi_{\theta_t} + \beta F_\rho(\pi_{\theta_t})^\dagger \cdot \left( \nabla_\theta J(\pi_\theta)|_{\pi=\pi_{\theta_t}} \right), \\
&= \pi_{\theta_t} + \beta F_\rho(\pi_{\theta_t})^\dagger \cdot \left( \nabla_\theta \left( A^{\pi_{\theta_t}} + \alpha A^{\text{TRM}} \right) |_{\pi=\pi_{\theta_t}} \right),
\end{aligned}
\tag{23}
$$

where $\beta$ is the learning rate, and $F^\dagger$ denotes the Moore-Penrose pseudoinverse of the matrix $F$.

**Lemma 3** (Policy Difference). *For all policies $\tilde{\pi}, \pi$ and states $s_0$, the policy difference from updating $\pi$ to $\tilde{\pi}$ is:*

$$
V^{\tilde{\pi}}(s_0) - V^\pi(s_0) = \frac{1}{1-\gamma} \mathbb{E}_{s \sim d^{\tilde{\pi}}_{s_0}} \mathbb{E}_{a \sim \tilde{\pi}(\cdot|s)} \left[ A^\pi(s, a) \right].
\tag{24}
$$

*Proof.* Let $\Pr^{\tilde{\pi}}(\tau \mid s_0 = s)$ denote the probability of observing a trajectory $\tau$ when starting in state $s$ and following the policy $\tilde{\pi}$. Using a telescoping argument, we have:

$$
\begin{aligned}
V^{\tilde{\pi}}(s) - V^\pi(s) &= \mathbb{E}_{\tau \sim \Pr^{\tilde{\pi}}(\tau|s_0=s)} \left[ \sum_{t=0}^\infty \gamma^t r(s_t, a_t) \right] - V^\pi(s) \\
&= \mathbb{E}_{\tau \sim \Pr^{\tilde{\pi}}(\tau|s_0=s)} \left[ \sum_{t=0}^\infty \gamma^t \left( r(s_t, a_t) + V^\pi(s_t) - V^\pi(s_t) \right) \right] - V^\pi(s) \\
&\overset{(a)}{=} \mathbb{E}_{\tau \sim \Pr^{\tilde{\pi}}(\tau|s_0=s)} \left[ \sum_{t=0}^\infty \gamma^t \left( r(s_t, a_t) + \gamma V^\pi(s_{t+1}) - V^\pi(s_t) \right) \right] \\
&\overset{(b)}{=} \mathbb{E}_{\tau \sim \Pr^{\tilde{\pi}}(\tau|s_0=s)} \left[ \sum_{t=0}^\infty \gamma^t \left( r(s_t, a_t) + \gamma \mathbb{E}\left[ V^\pi(s_{t+1}) \mid s_t, a_t \right] - V^\pi(s_t) \right) \right] \\
&= \mathbb{E}_{\tau \sim \Pr^{\tilde{\pi}}(\tau|s_0=s)} \left[ \sum_{t=0}^\infty \gamma^t A^\pi(s_t, a_t) \right] \\
&= \frac{1}{1-\gamma} \mathbb{E}_{s' \sim d^{\tilde{\pi}}_s} \mathbb{E}_{a \sim \tilde{\pi}(\cdot|s)} [A^\pi(s', a)],
\end{aligned}
\tag{25}
$$

where (a) rearranges terms in the summation and cancels the $V^\pi(s_0)$ term with the $-V^\pi(s)$ outside the summation, (b) uses the tower property of conditional expectations, and the final equality follows from the definition of $d^\pi_s$ in Eq. 17. □

**Lemma 4** (Natural policy gradient update). *With the softmax parameterization in Eq. 19, the natural policy gradient update in Eq. 23 takes the form:*

$$
\pi_{\theta_{t+1}}(a|s) = \pi_{\theta_t}(a|s) \cdot \frac{\exp\left( \beta \left( A^{\pi_{\theta_t}}(s, a) + \alpha A^{\text{TRM}}(s, a) \right) / (1-\gamma) \right)}{Z^t(s)},
\tag{26}
$$

*where $Z^t(s) = \sum_{a \in \mathcal{A}} \pi_{\theta_t}(a|s) \cdot \exp\left( \beta \left( A^{\pi_{\theta_t}}(s, a) + \alpha A^{\text{TRM}}(s, a) \right) / (1-\gamma) \right)$ is the partition function.*

*Proof.* We follow the proof in (Agarwal et al., 2021), with the key difference of separately accounting for the TRM-related term $A^{\text{TRM}}$. Following the definition of compatible function approximation in (Sutton et al., 1999), which was also invoked in (Kakade, 2001), for a vector $w \in \mathbb{R}^{|\mathcal{S}| \times |\mathcal{A}|}$, we define the error function:

$$
L^\theta(w) = \mathbb{E}_{s \sim d^{\pi_\theta}_\rho, a \sim \pi_\theta(\cdot|s)} \left( w^\top \nabla_\theta \log \pi_\theta(\cdot|s) - (A^{\pi_\theta}(s, a) + \alpha A^{\text{TRM}}(s, a)) \right)^2.
\tag{27}
$$

Let $w^*_\theta$ be the minimizer of $L^\theta(w)$ with the smallest $\ell_2$ norm. Then, by the definition of Moore-Penrose pseudoinverse, it can be derived as

$$
\begin{aligned}
w^\star_\theta &= F_\rho(\theta)^\dagger \mathbb{E}_{s \sim d^{\pi_\theta}_\rho, a \sim \pi_\theta(\cdot|s)} \left[ \nabla_\theta \log \pi_\theta(a|s) \cdot \left( A^{\pi_\theta}(s, a) + \alpha A^{\text{TRM}}(s, a) \right) \right] \\
&= (1-\gamma) F_\rho(\theta)^\dagger \nabla_\theta J(\pi_\theta).
\end{aligned}
\tag{28}
$$

In other words, $w_\theta^*$ is precisely proportional to the NPG update direction. Note further that for the Softmax policy parameterization in Eq. 19, we have

$$\frac{\partial \log \pi_\theta(a|s)}{\partial \theta_{s',a'}} = \mathbb{I}\left[s = s'\right] \cdot \left(\mathbb{I}\left[a = a'\right] - \pi_\theta\left(a'|s\right)\right), \tag{29}$$

where $\mathbb{I}[\mathcal{E}]$ is the indicator of $\mathcal{E}$ being true. Further, we have:

$$w^\top \nabla_\theta \log \pi_\theta(a|s) = w_{s,a} - \sum_{a' \in \mathcal{A}} w_{s,a'} \cdot \pi_\theta\left(a'|s\right). \tag{30}$$

Since $\sum_{a \in \mathcal{A}} \pi(a|s) A^\pi(s, a) = 0$, this immediately yields that $L^\theta(A^{\pi_\theta} + \alpha A^{\text{TRM}}) = 0$. However, this might not be the unique minimizer of $L^\theta$, which is problematic since $w^*(\theta)$ as defined in terms of the Moore-Penrose pseudoinverse is formally the smallest norm solution to the least-squares problem, which $A^{\pi_\theta} + \alpha A^{\text{TRM}}$ may not be. However, given any vector $v \in \mathbb{R}^{|\mathcal{S}| \times |\mathcal{A}|}$, let us consider solutions of the form $A^{\pi_\theta} + \alpha A^{\text{TRM}} + v$. Due to the form of the derivatives of the policy for the softmax parameterization (recall Eq. 29), we have for any state $s, a$ such that $s$ is reachable under $\rho$:

$$v^\top \nabla_\theta \log \pi_\theta(a|s) = \sum_{a' \in \mathcal{A}} \left(v_{s,a'} \mathbb{I}\left[a = a'\right] - v_{s,a'} \cdot \pi_\theta\left(a'|s\right)\right) = v_{s,a} - \sum_{a' \in \mathcal{A}} v_{s,a'} \cdot \pi_\theta\left(a'|s\right). \tag{31}$$

Note that here we have used that $\pi_\theta$ is a stochastic policy with $\pi_\theta(a|s) > 0$ for all actions $a$ in each state $s$, so that if a state is reachable under $\rho$, it will also be reachable using $\pi$, and hence the zero derivative conditions apply at each reachable state. For $A^{\pi_\theta} + \alpha A^{\text{TRM}} + v$ to minimize $L^\theta$, we would like $v^\top \nabla_\theta \log \pi_\theta(a|s) = 0$ for all $s, a$ so that $v_{s,a}$ is independent of the action and can be written as a constant $c_s$ for each $s$ by the above equality. Hence, the minimizer of $L^\theta(w)$ is determined up to a state-dependent offset, and

$$F_\rho(\theta)^\dagger \nabla_\theta J(\pi_\theta) = \frac{A^{\pi_\theta} + \alpha A^{\text{TRM}}}{1 - \gamma} + v, \tag{32}$$

where $v_{s,a} = c_s$ for some $c_s \in \mathbb{R}$ for each state $s$ and action $a$. Finally, we observe that this yields the update:

$$\theta_{t+1} = \theta_t + \frac{\beta}{1 - \gamma}\left(A^{\pi_\theta} + \alpha A^{\text{TRM}}\right) + \beta v,$$
$$\pi_{\theta_{t+1}}(a|s) = \pi_{\theta_t}(a|s) \cdot \frac{\exp\left(\beta\left(A^{\pi_{\theta_t}}(s, a) + \alpha A^{\text{TRM}}(s, a)\right) / (1 - \gamma) + \beta c_s\right)}{Z_t(s)}, \tag{33}$$

Owing to the normalization factor $Z_t(s)$, the state-dependent offset $c_s$ cancels in the updates for $\pi$, so that the resulting policy is invariant to the specific choice of $c_s$. Hence, we pick $cs \equiv 0$, which yields the statement of Lemma 4. $\qquad \square$

### B.3. Proof of Theorem 2

*Proof.* We follow the proof in (Setlur et al., 2025). From the policy difference in Lemma 3, we have

$$\mathbb{E}_{s \sim \rho}[V^{\pi_{t+1}}(s)] - \mathbb{E}_{s \sim \rho}[V^{\pi_t}(s)] = \frac{1}{1 - \gamma}\mathbb{E}_{s \sim d_\rho^{\pi_{t+1}}}\mathbb{E}_{a \sim \pi_{t+1}(\cdot|s)}[A^{\pi_t}(s, a)]. \tag{34}$$

From the natural policy gradient update in Lemma 4, we can re-write $A^{\pi_t}(s, a)$ as

$$A^{\pi_t}(s, a) = \frac{1 - \gamma}{\beta} \cdot \log\left(\frac{\pi_{t+1}(a|s) \cdot Z_t(s)}{\pi_t(a|s)}\right) - \alpha A^{\text{TRM}}(s, a), \tag{35}$$

Substituting Eq. 35 in Eq. 34, we can get:

$$\mathbb{E}_{s \sim \rho}[V^{\pi_{t+1}}(s)] - \mathbb{E}_{s \sim \rho}[V^{\pi_t}(s)] =$$
$$\frac{1 - \gamma}{\beta}\mathbb{E}_{s \sim d_\rho^{\pi_{t+1}}}\left[\text{KL}\left(\pi_{t+1}(\cdot \mid s)\|\pi_t(\cdot \mid s)\right)\right] + \frac{1 - \gamma}{\beta}\log Z_t(s) - \mathbb{E}_{s \sim d_\rho^{\pi_{t+1}}}\mathbb{E}_{a \sim \pi_{t+1}(\cdot|s)}[\alpha A^{\text{TRM}}(s, a)]. \tag{36}$$

The partition function $Z_t(s)$ can be re-arranged as

$$\log Z_t(s) = \log \mathbb{E}_{a \sim \pi_t(\cdot|s)}\left[\exp\left(\beta\left(A^{\pi_t}(s, a) + \alpha A^{\text{TRM}}(s, a)\right) / (1 - \gamma)\right)\right]. \tag{37}$$

Applying Jensen's inequality ($\log \mathbb{E}[\exp(X)] \geq \mathbb{E}[X]$) and $\mathbb{E}_{\pi_t}[A^{\pi_t}(s, a)] = 0$, we get:

$$
\begin{aligned}
\log Z_t(s) &\geq \frac{\beta}{1-\gamma} \cdot \mathbb{E}_{a \sim \pi_t(\cdot|s)} \left[ A^{\pi_t}(s, a) + \alpha A^{\text{TRM}}(s, a) \right] \\
&= \frac{\beta}{1-\gamma} \cdot \mathbb{E}_{a \sim \pi_t(\cdot|s)} \left[ \alpha A^{\text{TRM}}(s, a) \right].
\end{aligned}
\tag{38}
$$

Note that the KL term in Eq. 36 is always non-negative, and we can lower bound our policy improvement as

$$
\mathbb{E}_{s \sim \rho}[V^{\pi_{t+1}}(s)] - \mathbb{E}_{s \sim \rho}[V^{\pi_t}(s)] \geq \mathbb{E}_{s \sim d_\rho^{\pi_{t+1}}} \left[ \langle \pi_{t+1}(\cdot \mid s) - \pi_t(\cdot \mid s), \alpha A^{\text{TRM}}(s, \cdot) \rangle \right],
\tag{39}
$$

where the inner product is the standard Euclidean product, as the action space $\mathcal{A}$ is discrete.

In the following, we will treat the distribution $\pi_{t+1}(\cdot|s)$ as a vector denoted by $\pi$. Re-arranging the natural policy gradient update in Eq. 23 we get:

$$
\pi_{t+1}(a|s) - \pi_t(a|s) = \pi_t(a|s) \left( \frac{\exp \left( \beta \left( A^{\pi_t}(s, a) + \alpha A^{\text{TRM}}(s, a) \right) / (1 - \gamma) \right)}{Z_t(s)} - 1 \right).
\tag{40}
$$

We note that for $\beta \ll 1$, $\exp \left( \beta (A^{\pi_t}(s, a) + \alpha A^\mu(s, a)) / (1 - \gamma) \right) = \Theta(1 + \beta(A^{\pi_t}(s, a) + \alpha A^{\text{TRM}}(s, a)) / (1 - \gamma))$, where the terms that grow as $\omega(\beta/(1 - \gamma))$ are ignored. Based on this, for $\beta \ll 1$, there exist constants $0 < C_1 < C_2$ such that:

$$
\exp \left( \frac{\beta}{1-\gamma} (A^{\pi_t}(s, a) + \alpha A^{\text{TRM}}(s, a)) \right) - 1 \in \left[ \frac{C_1 \beta}{1-\gamma} (A^{\pi_t}(s, a) + \alpha A^{\text{TRM}}(s, a)), \frac{C_2 \beta}{1-\gamma} (A^{\pi_t}(s, a) + \alpha A^{\text{TRM}}(s, a)) \right].
\tag{41}
$$

Then, we have:

$$
\begin{aligned}
\pi_{t+1}(a \mid s) - \pi_t(a|s) &\geq \pi_t(a|s) \left( \frac{1 + \frac{C_1 \beta}{1-\gamma}(A^t(s, a) + \alpha A^{\text{TRM}}(s, a))}{1 + \frac{C_2 \beta}{1-\gamma} \mathbb{E}_{a|\pi_t(\cdot|s)}[A^{\pi_t}(s, a) + \alpha A^{\text{TRM}}(s, a)]} - 1 \right) \\
&\geq \frac{C_3 \beta}{1-\gamma} \cdot \frac{(\pi_t(a|s)(A^{\pi_t}(s, a) + \alpha A^{\text{TRM}}(s, a)) - \pi_t(a|s) \mathbb{E}_{a \sim \pi_t(\cdot|s)}[A^t(s, a) + \alpha A^{\text{TRM}}(s, a)])}{1 + \frac{C_2 \beta}{1-\gamma} \mathbb{E}_{a \sim \pi_t(\cdot|s)}[A^{\pi_t}(s, a) + \alpha A^{\text{TRM}}(s, a)]} \\
&= \frac{C_3 \beta}{1-\gamma} \cdot \frac{(\pi_t(a|s)(A^{\pi_t}(s, a) + \alpha A^{\text{TRM}}(s, a)) - \pi_t(a|s) \mathbb{E}_{a \sim \pi_t(\cdot|s)}[\alpha A^{\text{TRM}}(s, a)])}{1 + \frac{C_2 \beta}{1-\gamma} \mathbb{E}_{a \sim \pi_t(\cdot|s)}[A^{\pi_t}(s, a) + \alpha A^{\text{TRM}}(s, a)]},
\end{aligned}
\tag{42}
$$

where we reused $\mathbb{E}_{\pi_t}[A^{\pi_t}(s, a)] = 0$, and $C_3 > 0$ is a constant.

We now plug in the above lower bound into Eq. 39 to get the final lower bound on the policy improvement. For this, we will once again use the assumption that the learning rate $\beta \ll 1$, which allows us to use $1 + \frac{\beta}{1-\gamma} \mathbb{E}_{a \sim \pi_t(\cdot|s)}[A^{\pi_t}(s, a) + \alpha A^{\text{TRM}}(s, a)] \geq C_4$ for some constant $C_4 > 0$. This is because, in our setting, the range of the advantages is $[-1, 1]$. Since, advantages are bounded in $[-1, 1]$, we know that $1 + \frac{C_2 \beta}{1-\gamma} \mathbb{E}_{a \sim \pi_t(a|s)}[A^t(s, a) + \alpha A^{\text{TRM}}(s, a)] \leq 1 - \beta$. Thus,

$$
\begin{aligned}
\mathbb{E}_{s \sim \rho} \left[ V^{\pi_{t+1}}(s) - V^{\pi_t}(s) \right] \geq \frac{C_3}{1-\beta} \Bigg( &\beta \mathbb{E}_{s \sim d_\rho^{\pi_{t+1}}} \left[ \mathbb{E}_{a \sim \pi_t(\cdot|s)} \left[ \alpha A^{\text{TRM}}(s, a) A^{\pi_t}(s, a) \right] \right] \\
&+ \beta \mathbb{E}_{s \sim d_\rho^{\pi_{t+1}}} \left[ \mathbb{E}_{a \sim \pi_t(\cdot|s)} \left[ \left( \alpha A^{\text{TRM}}(s, a) \right)^2 \right] \right] \\
&- \beta \mathbb{E}_{s \sim d_\rho^{\pi_{t+1}}} \left[ \left( \mathbb{E}_{a \sim \pi_t(\cdot|s)} \left[ \alpha A^{\text{TRM}}(s, a) \right] \right)^2 \right] \Bigg).
\end{aligned}
\tag{43}
$$

Setting $C$ from Theorem 2 as $C_3/(1 - \beta)$, we have

$$
\begin{aligned}
\mathbb{E}_{s \sim \rho} \left[ V^{t+1}(s) - V^t(s) \right] \geq &C \beta \mathbb{E}_{s \sim d_\rho^{\pi_{t+1}}} \left[ \mathbb{E}_{a \sim \pi_t(\cdot|s)} \left[ \alpha A^{\text{TRM}}(s, a) A^{\pi_t}(s, a) \right] \right] \\
&+ C \beta \mathbb{E}_{s \sim d_\rho^{\pi_{t+1}}} \left[ \mathbb{E}_{a \sim \pi_t(\cdot|s)} \left[ \left( \alpha A^{\text{TRM}}(s, a) \right)^2 \right] \right] \\
&- C \beta \mathbb{E}_{s \sim d_\rho^{\pi_{t+1}}} \left[ \left( \mathbb{E}_{a \sim \pi_t(\cdot|s)} \left[ \alpha A^{\text{TRM}}(s, a) \right] \right)^2 \right].
\end{aligned}
\tag{44}
$$

Then, we get a tighter bound as

$$\mathbb{E}_{s\sim\rho}\left[V^{\pi_{t+1}}(s) - V^{\pi_t}(s)\right] \geq C\beta\mathbb{E}_{s\sim d_\rho^{\pi_{t+1}}}\left[\mathbb{V}_{a\sim\pi_t(\cdot|s)}\left[\alpha A^{\mathrm{TRM}}(s,a)\right] + \mathbb{E}_{a\sim\pi_t(\cdot|s)}\left[\alpha A^{\mathrm{TRM}}(s,a)A^t(s,a)\right]\right]. \quad (45)$$

Now, for the last step we note that $d_\rho^{\pi_{t+1}}$ is component wise larger than the initial state distribution $\rho$, and this gives us the final result:

$$\mathbb{E}_{s\sim\rho}\left[V^{\pi_{t+1}}(s) - V^{\pi_t}(s)\right] \geq C\beta \cdot \mathbb{E}_{s\sim\rho}\left[\mathbb{V}_{a\sim\pi_t(\cdot|s)}\left[\alpha A^{\mathrm{TRM}}(s,a)\right] + \mathbb{E}_{a\sim\pi_t(\cdot|s)}\left[\alpha A^{\mathrm{TRM}}(s,a)A^{\pi_t}(s,a)\right]\right]. \quad (46)$$

This completes the proof.

$\square$

## C. Discussion

### C.1. Discussion on ReasonFlux-PRM and our TRM

In this section, we contrast ReasonFlux-PRM and our TRM along three key design dimensions.

**Separation of Reasoning Quality and Correctness.**  ReasonFlux-PRM incorporates explicit correctness-oriented signals into its reward formulation by conditioning step-level rewards on the final response and applying response-aligned supervision (Zou et al., 2025). As a result, reasoning quality and outcome correctness are coupled in its training objective. In contrast, our TRM is trained exclusively on verified-correct reasoning traces and does not observe incorrect answers during training, thereby isolating reasoning quality from correctness. This design makes TRM orthogonal to verifiable rewards in the RLVR setting, where correctness is enforced by a verifier and TRM focuses on shaping the quality of reasoning among correct solutions.

**Explicit Modeling of Reasoning Quality by $ME^2$ principle.**  ReasonFlux-PRM primarily evaluates local properties of reasoning traces, such as step-level soundness and short-range semantic coherence between adjacent steps or between steps and the final response. In contrast, our TRM is built on an explicit characterization of reasoning quality ($ME^2$ principle) that decomposes evaluation into macro- and micro-level dimensions. Starting from this multi-level perspective, we model reasoning traces as DAGs to support structured evaluation across different granularities. This formulation enables TRM to assess global organization, including long-range dependencies, branching, and merging behaviors, while remaining grounded in well-defined dimensions of reasoning quality, rather than relying solely on local similarity signals.

**Pairwise Preference Supervision vs. Absolute Scoring.**  Another key difference lies in the form of supervision used to train the reward model. ReasonFlux-PRM relies on absolute or quasi-absolute scoring signals at the step level, which are known to be difficult to calibrate consistently and often require substantial human or model-side effort (Lightman et al., 2023; Zhang et al., 2025b). In contrast, our TRM is trained using pairwise preference supervision over reasoning traces. Such pairwise judgments are generally more reliable than absolute scoring and scale more naturally to large datasets, enabling robust learning of reasoning quality without requiring finely calibrated scores (Levtsov & Ustalov, 2025; Liusie et al., 2024; Raina et al., 2024).

### C.2. Discussion on Self-Refine

Self-Refine is a representative test-time scaling method that iteratively improves model outputs through self-feedback and refinement (Madaan et al., 2023; Muennighoff et al., 2025). Under our DAG formulation, each refinement round can be viewed as a verification and revision step that causally depends on the previous draft. As a result, the refinement process is naturally represented as a progression structure, where later steps build on and revise earlier ones. Although the abstract algorithmic view of Self-Refine resembles a loop, the actual reasoning trace produced by the model corresponds to a sequential unrolling of refinement iterations, which remains acyclic with respect to dependency and is therefore compatible with a DAG representation aligned with generation order (Lee et al., 2025).

Explicitly modeling cyclic structures would require recovering fully general graphs from arbitrary free-form reasoning traces. This is difficult and brittle in practice, as it demands reliably identifying revisited latent states and long-range dependencies across the trace, a challenge widely noted in prior work on structuring complex reasoning (Xiong et al., 2025; Minegishi

et al., 2025; Jiang et al., 2025). While predefined markers or constrained output formats can be used to enforce parseable cyclic structures, such approaches rely on strict prompting conventions and do not generalize well to arbitrary traces. In contrast, our DAG abstraction provides a practical and scalable compromise. It captures the essential causal structure of Self-Refine as iterative verification and revision, while remaining applicable to unconstrained, real-world reasoning traces without requiring specialized formats.

### C.3. Discussion on Robust Evaluation

Although DAG construction relies on an LLM to infer semantic dependencies between reasoning steps, the resulting evaluation is relatively robust. Our evaluation is based on relative, pairwise comparison rather than absolute scoring, which reduces sensitivity to individual errors. In addition, we apply a robust evaluation protocol (Sec. 4.3) that includes forward and reversed orderings and repeated comparisons, and use a strong and stable evaluator (DeepSeek-V3.2) throughout. Empirically, this design yields reliable and consistent preference judgments, which is consistent with the downstream performance of our TRM.

## D. Details of Step Partitioning

We further elaborate on the step partitioning procedure used in Sec. 4. As an initial preprocessing step, we split a reasoning trace $s$ using "\n\n", producing a sequence of raw segments $(\hat{s}_0, \hat{s}_1, \ldots, \hat{s}_m)$. This splitting is convenient and model-agnostic, but it often yields partitions that are overly fine-grained. In practice, models frequently use "\n\n" for formatting purposes, such as breaking lines within a single formula or organizing a continuous chain of thought, rather than to denote genuine step boundaries. This procedure can introduce an excessive number of raw steps, unnecessarily increasing the complexity of subsequent DAG construction.

To obtain more coherent step boundaries, we analyze the prefix words of each raw segment $\hat{s}_i$. We find that certain high-frequency prefix words serve as reliable indicators of step transitions, as they often signal progression, contrast, or shifts in reasoning focus. By identifying and leveraging these prefix keywords, we merge adjacent raw segments into more meaningful steps, where each step typically contains a locally continuous span of reasoning rather than an arbitrary fragment.

In the dataset construction described in Sec. 5.1, we use three open-source reasoning model families: Qwen3, DeepSeek-R1-Distill, and GPT-OSS. For each family, we collect statistics over prefix words appearing in the raw segments and manually select a small set of frequent and semantically meaningful keywords that reliably indicate the start of a new reasoning step. The resulting distributions are shown in Fig. 7. Across model families, we observe both commonalities and stylistic differences. Keywords such as "so" and "but" consistently rank among the most frequent, reflecting shared discourse patterns for progression and contrast. At the same time, family-specific preferences are evident. For example, "therefore" appears frequently in the DeepSeek-R1-Distill and Qwen3 families but is less common in GPT-OSS traces. Conversely, GPT-OSS often uses keywords such as "need" to explicitly mark subgoals and advance reasoning (e.g., *Need to compute* $\frac{\alpha}{2}$ *from earlier.*").

Overall, reasoning traces exhibit a mixture of shared discourse cues and model-specific stylistic patterns. By analyzing these prefix keywords and using them to refine step boundaries, we obtain more coherent and stable step partitions. This refinement substantially reduces unnecessary fragmentation and provides a cleaner and more reliable foundation for downstream DAG construction.

## E. Details of DAG Construction

This section provides supplementary details for the DAG construction procedure described in Sec. 4.2. As introduced in the main text, we construct the reasoning DAG incrementally by traversing the partitioned reasoning steps in their generation order. For each newly processed step, an LLM is prompted to select its parent nodes from a maintained attachment pool, thereby inducing directed edges that capture semantic dependencies between steps. This incremental formulation allows the resulting DAG to express progression, branching, and merging patterns observed in LLM-generated reasoning traces.

**Full Attachment Pool Design.** A straightforward design choice is to include *all* previously generated steps as candidates in the attachment pool when determining the parents of a new node. While this approach preserves complete historical information and, in principle, allows the LLM to recover any valid dependency structure, it is impractical in large-scale or long-horizon settings. As the number of reasoning steps grows, such a design leads to two major issues: (i) excessive

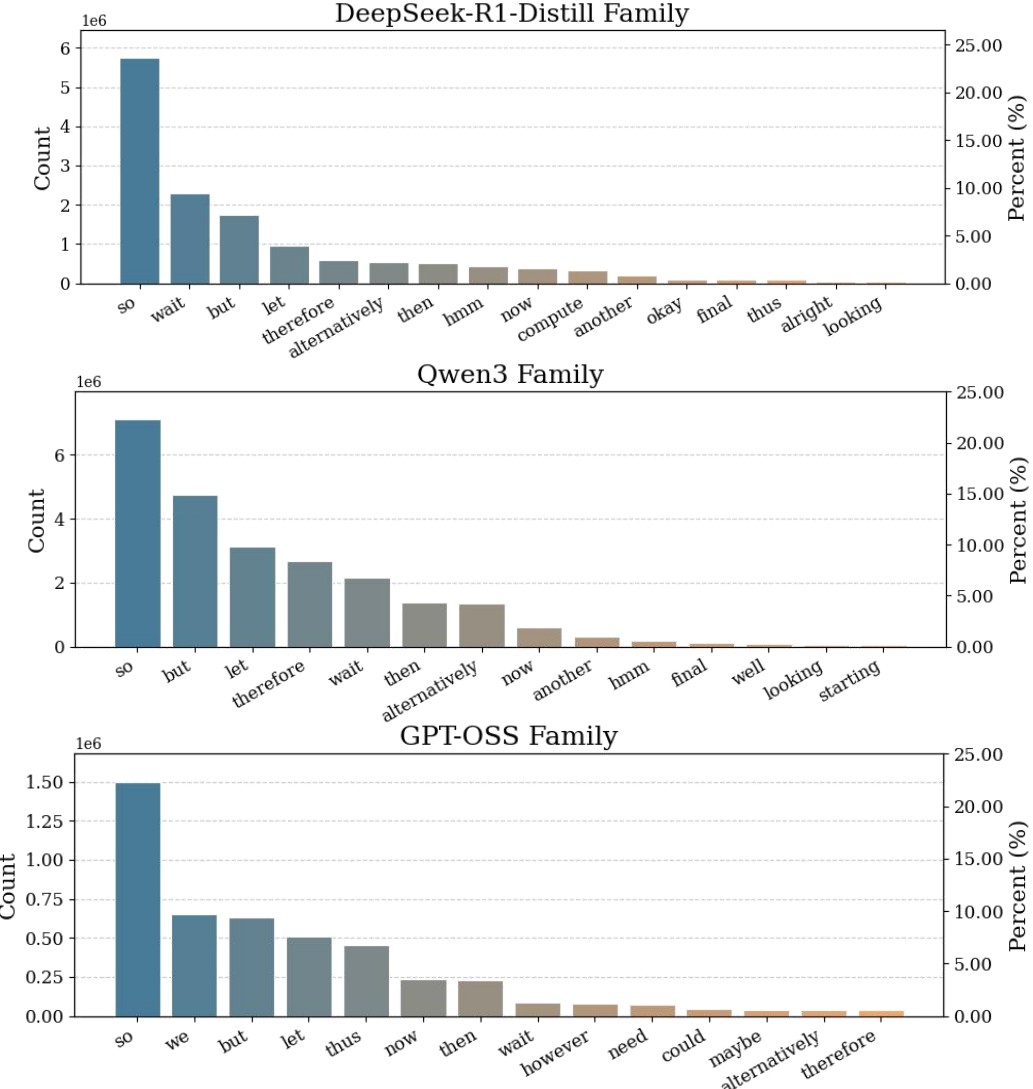

*Figure 7.* Distribution of selected high-frequency prefix keywords across raw steps obtained by splitting reasoning traces with "\n\n". The left $y$-axis shows the total count of each keyword, while the right $y$-axis indicates its proportion among all raw steps. Results are shown for three reasoning model families.

prompt length and inference cost when presenting long contexts to the LLM, and (ii) degraded parent-selection accuracy due to long-context interference, where the model struggles to identify the most semantically relevant predecessors among many candidates.

**Simplified Attachment Pool Design.** To balance structural fidelity and computational efficiency, we adopt a simplified attachment pool design, illustrated in Fig. 8. At each step $i$, the attachment pool $\mathcal{P}_i$ is composed of two parts: (i) the *current main branch*, defined as the path from the root node to the most recently processed node, and (ii) a small set of *representative branch endpoints*, namely leaf nodes from other branches that are not on the main branch. This construction is consistent with the pool definition described in Sec. 4.2. The main branch captures the reasoning path up to the current step and therefore provides the most likely parents for semantic continuation. Meanwhile, the branch endpoints summarize the latest states of alternative reasoning paths. Including these nodes enables the model to reconnect to earlier exploratory branches, which is crucial for accurately modeling merging behaviors, such as reflection, verification, or consolidation of parallel lines of reasoning.

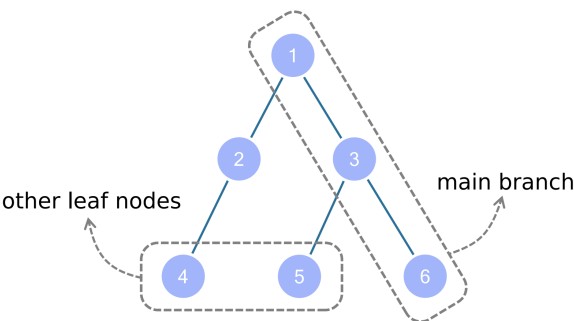

*Figure 8.* Illustration of the simplified attachment pool. All edges are directed from top to bottom, and node indices indicate the step order. The gray dashed region denotes the attachment pool for a given new step, consisting of nodes along the main branch and leaf nodes from other branches.

**Rationale and Complexity Analysis.** Compared to using all prior nodes, the simplified attachment pool reduces the candidate set size from $O(n)$ to $O(d + b)$, where $n$ is the total number of steps processed so far, $d$ is the depth of the current main branch, and $b$ is the number of retained branch endpoints. In practice, both $d$ and $b$ are smaller than $n$, leading to notable reductions in prompt length and inference cost. While this design necessarily omits some older and inactive nodes, our empirical analysis on reasoning traces from multiple model families suggests that most parent-child relations arise from either (i) local continuation along the current main branch, or (ii) reconnection to the most recent nodes of alternative branches. Under these conditions, the simplified attachment pool captures the majority of semantically meaningful dependencies observed in practice.

As a result, although it does not preserve the full historical context of the reasoning trace, this design provides a reasonable approximation to the complete DAG context in our experimental settings. We find that itis sufficient in practice for recovering accurate reasoning structures and serves as a key component enabling scalable and reliable modeling of reasoning DAGs.

## F. Details of Preference Dataset Construction

In this section, we describe the detailed procedure for constructing the preference dataset used in our experiments. The judgment steps involved in verification, structural annotation, and pairwise evaluation are performed using DeepSeek-V3.2 (Liu et al., 2025) as the judge model.

**Response Generation and Verification.** We sample 64K prompts from the WebInstruct-verified dataset (Ma et al., 2025) and generate candidate reasoning traces using multiple open-source reasoning models from different model families. Specifically, for the Qwen3 family (Yang et al., 2025), we use Qwen3-1.7B, Qwen3-8B, and Qwen3-32B; for the DeepSeek-Distill family (Guo et al., 2025), we use DeepSeek-Distill-Llama-8B and DeepSeek-Distill-Llama-70B; and for the GPT-OSS family (Agarwal et al., 2025), we use GPT-OSS-20B and GPT-OSS-120B. For all models, we set the maximum generation length to 8192 tokens, with temperature 0.7 and top-$p$ 0.95. For each prompt–model pair, we sample $n = 4$ reasoning traces.

To ensure answer correctness, we verify all generated responses using the same verifier prompt as in Ma et al. (2025), and retain only verified-correct reasoning traces for subsequent pairwise evaluation. This filtering step is introduced to decouple reasoning quality from final answer correctness, allowing the preference judgments to focus on the properties emphasized by *ME²* *principle* rather than being dominated by outcome-level errors.

**Motivation for Correctness Filtering.** In preliminary analysis, we observe that when incorrect answers are included, the judge model may face ambiguous trade-offs, for example between a reasoning trace that largely follows *ME²* *principle* but contains a minor error leading to an incorrect final answer, and another trace that violates *ME²* *principle* by relying on illogical shortcuts or spurious mathematical justifications yet happens to arrive at the correct answer. In such cases, correctness can overshadow structural and procedural reasoning quality, introducing noise and inconsistency into pairwise judgments. Restricting evaluation to verified-correct traces mitigates this effect and yields a cleaner supervision signal for comparing reasoning quality under a shared correctness constraint.

**Step Partitioning.** As shown in App. D, we analyze the empirical distribution of step-prefix patterns across reasoning traces from different model families and identify a small set of frequent and semantically indicative keywords that tend to mark the start of a new reasoning step. Using these keywords, we segment each reasoning trace into a sequence of atomic steps, applying family-specific keyword sets to account for stylistic differences across model outputs.

**DAG Construction.** Based on the resulting step sequences, we construct reasoning DAGs according to the procedure outlined in Sec. 4.2. Parent–child relations between steps are assigned incrementally using LLM-based selection over the attachment pool. The constructed DAGs provide the structured representations used in subsequent macro- and micro-level pairwise evaluation.

**Pairwise Evaluation.** We evaluate the processed reasoning traces using the procedure described in Sec. 4.3, operating on their corresponding DAG representations. All pairwise comparisons follow the hierarchical evaluation protocol and are conducted multiple times under both original and reversed presentation orders (three evaluations per order) to ensure robustness.

## G. Experimental Details

### G.1. Thinking Reward Model Training

We train the Thinking Reward Model (TRM) using the `trl RewardTrainer`[2]. Training is performed for one epoch with 12 gradient accumulation steps, resulting in a total batch size of 192. We use a learning rate of $1 \times 10^{-6}$ with a cosine scheduler and a warmup ratio of 0.03. To stabilize training, we set the reward centering coefficient to 0.001, which constrains the mean reward during optimization and improves numerical stability. The total training time is approximately 20 hours, and the resulting training dynamics are shown in Fig. 9.

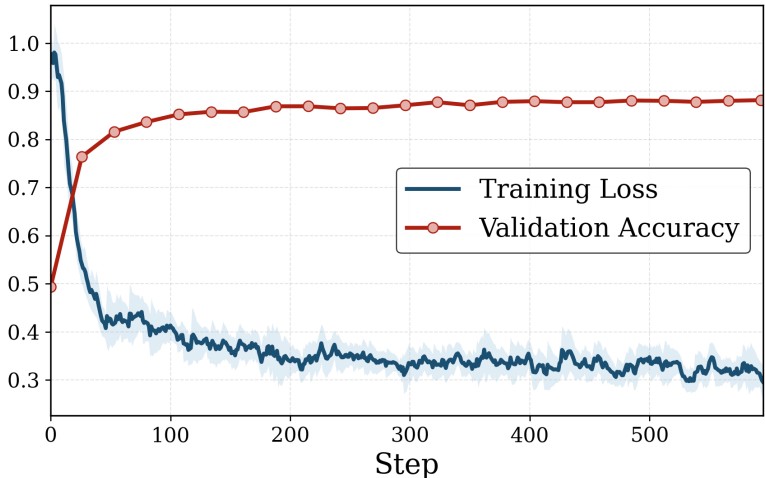

*Figure 9.* Training loss and validation accuracy during TRM training. The training loss converges to around 0.30, and the validation accuracy plateaus at 88.6%, suggesting stable optimization and strong generalization performance.

### G.2. Test-Time Scaling

We use Qwen3-8B and GPT-OSS-20B as generator models. For each prompt $x$, the generator samples $N$ candidate outputs at temperature $T = 0.6$ and top-p $= 0.95$, where $N \in \{1, 2, 4, 8, 16\}$. Each candidate consists of a reasoning trace and its final response. TRM scores candidates using only the reasoning trace, producing a scalar thinking reward. In contrast, PRMs (Qwen2.5-Math-7B-PRM and ReasonFlux-PRM-7B) score each candidate using both the reasoning trace and the final response. For each method, we select the candidate with the highest reward and report accuracy of the selected outputs.

[2]https://huggingface.co/docs/trl/main/en/reward_trainer

*Table 4.* Reliability validation of preference labels on 400 sampled pairs. Accuracy is computed over non-tie judgments.

| EVALUATOR | ACCURACY |
|---|---|
| Gemini 3.1 Pro | 86.3% |
| GPT-5.4 | 93.0% |
| Claude Opus 4.6 | 89.8% |
| Human experts | 90.0% |

## G.3. RL Optimization

We implement GRPO training using `verl`[3] and train three policy models. We use a learning rate of $5 \times 10^{-7}$, PPO minibatch size 128, training batch size 512, and PPO clip ratio 0.3. For each prompt, we sample 8 responses at temperature 1.0 and train for 300 steps. We use General-Verifier (Ma et al., 2025) with math verification as the verifier. For the reward signal of **ReasonFlux-PRM-7B** (Zou et al., 2025), we follow the paper's aggregation, which combines the step-level reward and trajectory-level reward. Following Yan et al. (2025), we use the following system prompt for Qwen2.5-Math-1.5B and Qwen2.5-Math-7B:

> **Qwen2.5-Math-7B System Prompt**
>
> Your task is to follow a systematic, thorough reasoning process before providing the final solution. This involves analyzing, summarizing, exploring, reassessing, and refining your thought process through multiple iterations. Structure your response into two sections: Thought and Solution. In the Thought section, present your reasoning using the format: "`<think>`\n thoughts `</think>`\n". Each thought should include detailed analysis, brainstorming, verification, and refinement of ideas. After "`</think>`\n" in the Solution section, provide the final, logical, and accurate answer, clearly derived from the exploration in the Thought section. If applicable, include the answer in `\boxed{}` for closed-form results like multiple choices or mathematical solutions.
> `{problem}`

For the comparatively weaker Llama-3.1-8B-Instruct, we use a simplified system prompt:

> **System Prompt**
>
> `{problem}`
> Please reason step by step, and put your final answer within `\boxed{}`.

## G.4. Reliability Validation

To validate that the constructed preference labels reflect reasoning quality rather than the stylistic bias of the judge model, we sample 400 preference pairs and ask three human experts and three strong LLMs (Gemini 3.1 Pro, GPT-5.4, and Claude Opus 4.6) to independently annotate each pair with win/tie/loss labels. We follow the same robust protocol as in Sec. 4.3 and report non-tie accuracy, excluding tied comparisons from the denominator. As shown in Tab. 4, the preference labels show high agreement with both human and LLM judgments, suggesting that the labels are largely aligned with the *ME²* *principle* rather than being artifacts of DeepSeek-V3.2.

## G.5. Offline Construction Cost

The construction cost of TRM-Preference is incurred once offline. For DAG construction, the simplified attachment pool reduces the candidate set size from $O(n)$ to $O(d + b)$, where $n$ is the total number of processed steps, $d$ is the depth of the current main branch, and $b$ is the number of retained branch endpoints. In our implementation, we process approximately 180K traces with roughly 30 LLM calls per trace and about 1K tokens per call, resulting in about 5.4B tokens for DAG construction. Pairwise comparison uses approximately 130K comparisons with four dimensions and four orderings, plus aggregation, totaling about 3.6B tokens. Including the remaining steps, the full pipeline uses less than 12B DeepSeek-V3.2 tokens, corresponding to roughly $5K in total or $0.05 per retained pair over the 103K retained pairs. This cost is amortized because the resulting TRM is reused for both RL optimization and test-time scaling.

---

[3]https://github.com/volcengine/verl

*Table 5.* Best-of-$N$ test-time scaling results on AIME25 with larger generators.

| MODEL | METHOD | $N = 1$ | $N = 2$ | $N = 4$ | $N = 8$ | $N = 16$ |
|---|---|---|---|---|---|---|
| GPT-OSS-120B | Qwen-PRM | 80.0 | 80.7 | 82.7 | 83.3 | 83.3 |
| GPT-OSS-120B | ReasonFlux-PRM | 80.0 | 82.0 | 82.0 | 84.0 | 84.7 |
| GPT-OSS-120B | TRM (ours) | 80.0 | 81.3 | 83.3 | 84.7 | 86.7 |
| Qwen3-235B-A22B | Qwen-PRM | 71.3 | 72.0 | 74.0 | 75.3 | 76.7 |
| Qwen3-235B-A22B | ReasonFlux-PRM | 71.3 | 72.7 | 74.0 | 76.0 | 76.0 |
| Qwen3-235B-A22B | TRM (ours) | 71.3 | 72.7 | 74.7 | 76.0 | 78.7 |

*Table 6.* Ablation study of the weighting coefficient $\alpha$ on Qwen2.5-Math-7B.

| $\alpha$ | STEM Performance (%) | | | | | | Math Performance (%) | | | | | |
|---|---|---|---|---|---|---|---|---|---|---|---|---|
| | GPQA | SuperGPQA | MMLU-Pro | BBEH | FS | Avg. | AMC | AIME24 | AIME25 | MATH500 | Olympiad | Avg. |
| 0.8 | 37.9 | 25.2 | 48.5 | 9.3 | 19.6 | 28.1 | 61.9 | 13.2 | 15.1 | 82.0 | 46.8 | 43.8 |
| 0.5 | 39.9 | 26.0 | 49.9 | **10.4** | 23.9 | 30.0 | **65.6** | 18.3 | 14.9 | 81.4 | **48.3** | 45.7 |
| 0.2 (ours) | **42.4** | **26.7** | **50.5** | 9.5 | **26.5** | **31.1** | 63.4 | **19.1** | **17.1** | **83.6** | 47.4 | **46.1** |

## G.6. Test-Time Scaling on Larger Models

We further evaluate test-time scaling on larger generators, GPT-OSS-120B and Qwen3-235B-A22B, using the same protocol as in Sec. 5.3. Tab. 5 reports Best-of-$N$ accuracy on AIME25. TRM consistently outperforms Qwen-PRM and ReasonFlux-PRM on both generators. The gains are smaller than those on smaller generators, which is expected given the stronger base performance, but the results show that TRM remains effective at larger model scales.

## H. Ablation Study

Tab. 6 reports an ablation on the weighting coefficient $\alpha$ in Eq. 1 conducted on Qwen2.5-Math-7B. Among the three tested values $\{0.2, 0.5, 0.8\}$, $\alpha = 0.2$ yields the best average performance on both STEM and math, while larger $\alpha$ values lead to lower averages. This suggests that a relatively small $\alpha$ can provide a reasonable balance in practice, keeping the verifier signal as the primary driver for outcome correctness while allowing TRM to shape and rank alternative verified-correct reasoning traces.

## I. Brief Introduction of Bradley–Terry Loss

The Bradley–Terry (BT) model is a standard probabilistic model for pairwise comparisons (Bradley & Terry, 1952). Given two candidates $a$ and $b$ (e.g., two reasoning traces under the same prompt $x$), a reward model $r_\theta(x, \cdot)$ assigns each candidate a scalar score, and the BT model defines the preference probability as

$$P_\theta(a \succ b \mid x) = \frac{\exp(r_\theta(x, a))}{\exp(r_\theta(x, a)) + \exp(r_\theta(x, b))} = \sigma(r_\theta(x, a) - r_\theta(x, b)), \tag{47}$$

where $\sigma(\cdot)$ is the sigmoid function. This formulation has become the default objective for reward modeling with pairwise preference data in RLHF (Stiennon et al., 2020; Ouyang et al., 2022). Given a preference dataset $\mathcal{D} = \{(x, a_w, a_l)\}$ with winner $a_w$ and loser $a_l$, the BT loss is the negative log-likelihood:

$$\mathcal{L}_{\text{BT}}(\theta) = -\mathbb{E}_{(x, a_w, a_l) \sim \mathcal{D}} \left[ \log \sigma(r_\theta(x, a_w) - r_\theta(x, a_l)) \right], \tag{48}$$

which encourages the model to assign higher scores to preferred candidates while avoiding unreliable prompt-based absolute scoring.

# J. Prompt Design

## J.1. DAG Construction Prompt

Fig. 16 and Fig. 17 show the prompt used for DAG construction. We construct the DAG in a step-by-step manner: for each newly processed step $v_i$, we provide the judge model with an attachment pool of previously seen steps and ask it to decide the connection type and the parent step IDs.

## J.2. DAG Linearization for Macro-level Evaluation

For macro-level evaluation, we convert each super-node DAG into a linearized textual abstraction. Each super-node corresponds to a merged span of reasoning steps that is semantically coherent, and is summarized into a single sentence. The super-nodes are arranged according to their original emission order, which aligns with the stepwise generation process and yields a consistent topological ordering. For each super-node, we attach a concise structural header indicating its role in the DAG, including whether it functions as a root, branching, merging, or terminal node, together with its depth and parent–child relations.

This abstraction exposes the global organization of the reasoning trace in a compact manner. Structural patterns such as the introduction of alternative solution paths, their expansion through branching, and their eventual consolidation or termination become directly observable at the text level. As illustrated in Fig. 10, branching nodes mark points where the reasoning diverges, while merge and leaf nodes reflect how intermediate conclusions are reconciled or finalized. The resulting abstraction removes low-level step content while preserving the structural signals required for macro-level comparison, and is used as input to the macro-level pairwise evaluation prompts.

## J.3. Dominant Path Extraction for Micro-level Evaluation

For micro-level evaluation, we extract a dominant-path abstraction from the DAG that corresponds to the main trajectory leading to the final answer. Concretely, we identify the leaf node or nodes that contain the final-answer statement and collect these nodes together with all of their ancestors in the DAG. The resulting set of nodes forms an induced subgraph that captures the primary line of reasoning, excluding auxiliary branches that reflect side explorations or verification attempts.

The nodes in this dominant subgraph are then serialized according to the original step emission order and concatenated into a single continuous text. This dominant-path abstraction preserves the local progression of reasoning steps along the main path while omitting structurally peripheral content, and is used directly as input to the micro-level evaluation prompts.

## J.4. Evaluation Prompt

We design four pairwise evaluation prompts, each corresponding to one dimension of the $ME^2$ *principle*. The prompts differ in the abstraction they take as input, reflecting the intended scope of evaluation.

The **Macro-Efficiency** prompt (Fig. 11) and the **Macro-Effectiveness** prompt (Fig. 12) operate on the macro-level abstraction produced by DAG linearization, which exposes the global organization of reasoning traces.

The **Micro-Efficiency** prompt (Fig. 13) and the **Micro-Effectiveness** prompt (Fig. 14) operate on the micro-level abstraction obtained from dominant-path extraction, which captures the main reasoning trajectory in a linearized form.

Finally, the aggregation prompt (Fig. 15) integrates the four dimension-specific judgements to produce a single overall pairwise preference. It treats the prior decisions and rationales as structured auxiliary inputs, allowing the evaluator to resolve potential conflicts across dimensions and issue a holistic judgement aligned with the $ME^2$ *principle*. This aggregation step provides the final supervision signal used for constructing pairwise preference data.

---

**Example – Macro-level Abstraction via DAG Linearization**

**Prompt:** I'm trying to invert the following function: $f(x) = \frac{1}{2}x^2 \ln x - \frac{1}{4}x^2$ for all $x > 1$. How do I proceed?
Please reason step by step, and put your final answer within \boxed{}.

**Linearized macro text:**
`<Root & Branch id=0 depth=0 parents=[] children=[1, 2, 9]>`
Although the function is invertible due to monotonicity, the inverse cannot be expressed in closed form; after factoring as $y = x^2\left(\frac{1}{2}\ln x - \frac{1}{4}\right)$, no algebraic manipulation yields an explicit solution for $x$ in terms of $y$, necessitating implicit or numerical treatment.

`<Leaf id=1 depth=1 parents=[0] children=[]>`
Recognizing the equation as transcendental prevents algebraic solution; substitution or approximation is suggested as a potential path forward.

`<Branch id=2 depth=1 parents=[0] children=[3, 4]>`
Shifts focus to monotonicity via derivative; strict monotonicity implies invertibility.

`<Leaf id=3 depth=2 parents=[2] children=[]>`
After simplification the derivative becomes $x \ln x$, which is positive for $x > 1$, so $f(x)$ is strictly increasing and therefore invertible on that interval.

`<Branch id=4 depth=2 parents=[2] children=[5, 6]>`
After recognizing the equation's transcendental nature, a substitution aligns it with the Lambert W function form: starting from $y = \frac{1}{2}x^2 \ln x - \frac{1}{4}x^2$, the substitution $u = \ln x$ leads to $2y = e^{2u}(u - \frac{1}{2})$, and with $v = u - \frac{1}{2}$, this becomes $2y = e^{2v+1}v$, preparing for solution via special functions.

`<Leaf id=5 depth=3 parents=[4] children=[]>`
After rewriting $2y = e^{2v+1}v$ as $2y = e \cdot e^{2v}v$, the expression is restated in its original form, confirming equivalent representations without further simplification.

`<Branch id=6 depth=3 parents=[4] children=[7, 8]>`
After reformulating the equation into Lambert W form via substitution, the inverse is derived and verified numerically, with attention shifting to selecting the correct branch of the Lambert W function based on domain considerations; the final expression is $x = \sqrt{e} \cdot e^{\frac{1}{2}W\left(\frac{4y}{e}\right)}$, pending branch validation.

`<Leaf id=7 depth=4 parents=[6] children=[]>`
After realizing an earlier assumption was flawed, the calculation of $e^{0.568}$ was refined using approximation, but the resulting product failed to match the target value, ruling out the candidate solution.

`<Leaf id=8 depth=4 parents=[6] children=[]>`
After narrowing $z$ between 0.38 and 0.39, linear interpolation gives $z \approx 0.3874$, satisfying $ze^z = 0.569$; back-substitution confirms $x = 2$, verifying the inverse function.

`<Leaf id=9 depth=1 parents=[0] children=[]>`
The key transformation occurs when the equation is rewritten in the form $ze^z = 4y/e$, enabling use of the Lambert W function; subsequent back-substitutions yield $x = \sqrt{e} \cdot e^{\frac{1}{2}W(4y/e)}$, which expresses the inverse in terms of a special function due to the non-algebraic solvability of the original relation.

*Figure 10.* A simple example of a macro-level abstraction obtained by linearizing a DAG-structured reasoning trace, where each block is prefixed with a structural tag (e.g., `<Branch id=6 depth=3 parents=[4] children=[7,8]>` indicates a branching super-node at depth 3 that expands the reasoning from node 4 into two subsequent paths).

---

**Prompt – Macro-Efficiency**

```
### Task
Please evaluate the given two reasoning traces to determine which trace is
overall stronger in terms of **Macro-Efficiency** (Whether the reasoning
structure is well organized, avoiding unnecessary branching and reflection),
or whether they remain tied.  Both traces already reach the correct final
answer.

### Question
${question}
<End of Question>

### Reasoning Traces
Trace 1:
${trace_one}
<End of Trace 1>

Trace 2:
${trace_two}
<End of Trace 2>

### Evaluation Focus
- Both traces are already validated as correct.
- Read the summaries and their header tags (e.g., '<Root & Branch id=0
depth=0 parents=[...]  children=[...]>') to see when branches open, how long
they persist, and how decisively they merge back.
- Reward structures that stay lean, retire dead ends quickly, and use
reflections selectively to consolidate progress or eliminate alternatives
in a way that guides subsequent steps.  Focus on whether major branches hand
off conclusions cleanly and keep the agenda tight.
- Penalize structural sprawl such as repeated branch reopenings, broad
detours that lead nowhere, or recurring summaries that restate the same idea
without advancing the main line of reasoning.  When verification is entirely
absent where complexity would reasonably call for it, treat that as poor
process discipline.
- Note when the reasoning dedicates effort to decisive paths versus
dispersing across loosely managed threads.  Efficiency includes allowing
necessary checks, but only when they shorten or stabilise the plan.
- In your reasoning paragraph, explicitly contrast Trace 1 and Trace 2 by
naming at least one concrete Macro-level Efficiency strength or weakness for
each before arriving at the decision.

### Output Format
One reasoning paragraph (1-8 sentences) analyzing the DAG headers, reasoning
texts and how structural patterns impact efficiency.
<|decision|> followed immediately by exactly one decision:  Trace 1, Trace
2, or Tie.
```

*Figure 11.* Macro-Efficiency evaluation prompt.

**Prompt – Macro-Effectiveness**

```
### Task
Please evaluate the given two reasoning traces to determine which trace
is overall stronger in terms of **Macro-Effectiveness** (Whether the
reasoning structure remains logically coherent and aligned with the problem
objective), or whether they remain tied.  Both traces already reach the
correct final answer.

### Question
${question}
<End of Question>

### Reasoning Traces
Trace 1:
${trace_one}
<End of Trace 1>

Trace 2:
${trace_two}
<End of Trace 2>

### Evaluation Focus
- Both traces are already validated as correct.
- Read the structural headers (e.g., '<Merge id=4 depth=2 parents=[1, 3]
children=[5]>') to see how objectives are introduced, how branches justify
their existence, and how conclusions are reintegrated.
- Reward traces where every major branch summary reinforces the problem
objective, keeps assumptions aligned, and explains why the path shifts focus
before merging back.
- Penalize vague or conflicting summaries that leave a branch's purpose
unclear, macro-level structural jumps that skip causal links, or conclusions
that do not obviously descend from the earlier context.
- Highlight whether the overall reasoning trace keeps a consistent framing
of the problem, acknowledges corrections, and closes every branch with a
clear semantic handoff.
- In your reasoning paragraph, explicitly contrast Trace 1 and Trace 2 by
naming at least one concrete Macro-level Effectiveness strength or weakness
for each before arriving at the decision.

### Output Format
One reasoning paragraph (1-8 sentences) explaining how the DAG headers,
structural patterns and reasoning texts demonstrate (or fail to demonstrate)
coherent global flow.
<|decision|> followed immediately by exactly one decision:  Trace 1, Trace
2, or Tie.
```

*Figure 12.* Macro-Effectiveness evaluation prompt.

---

**Prompt – Micro-Efficiency**

```
### Task
Please evaluate the given two reasoning traces to determine which trace is
overall stronger in terms of **Micro-Efficiency** (Whether individual steps
avoid local redundancy), or whether they remain tied.  Both traces already
reach the correct final answer.

### Question
${question}
<End of Question>

### Reasoning Traces
Trace 1:
${trace_one}
<End of Trace 1>

Trace 2:
${trace_two}
<End of Trace 2>

### Evaluation Focus
- Both traces are already validated as correct.
- Favor tight progressions where reflections, checks, or restatements
 perform concrete operations (e.g., calculations, variable updates, explicit
 decisions) that advance the main reasoning path.  Insightful reflections
 that catch errors or consolidate prior results should be rewarded, as they
 reduce future redundant reasoning.
- Penalize bursts of short, low-content steps that introduce no new
 information or operations (e.g., ``Let me think'', ``Retrying'') that
 appear without substantive reasoning, as well as filler phrases, hedging,
 or emotional commentary that add no evidence.
- Penalize repeated paraphrases of the same equation or conclusion,
 digressions into unrelated facts, or speculative branches that never
 resolve back into the main solution.  Long streaks of shallow reflections
 or self-talk without introducing new reasoning steps should count against
 efficiency.
- Penalize excessive trial-and-error loops, conflicting numerical attempts,
 or recalculations that simply restate previous values without refinement.
- Reward traces that reuse earlier results effectively, justify each local
 transition with explicit math or logic, and keep verification steps focused
 on issues that matter.
- In your reasoning paragraph, explicitly contrast Trace 1 and Trace 2 by
 naming at least one concrete Micro-level Efficiency strength or weakness for
 each before arriving at the decision.

### Output Format
One reasoning paragraph (1-8 sentences) justifying the efficiency judgment
with concrete evidence from the traces.
<|decision|> followed immediately by exactly one decision:  Trace 1, Trace
2, or Tie.
```

*Figure 13.* Micro-Efficiency evaluation prompt.

---

**Prompt – Micro-Effectiveness**

```
### Task
Please evaluate the given two reasoning traces to determine which trace is
overall stronger in terms of **Micro-Effectiveness** (Whether individual
steps are locally valid and internally consistent), or whether they remain
tied.  Both traces already reach the correct final answer.

### Question
${question}
<End of Question>

### Reasoning Traces
Trace 1:
${trace_one}
<End of Trace 1>

Trace 2:
${trace_two}
<End of Trace 2>

### Evaluation Focus
- Both traces are already validated as correct.
- Examine the detailed reasoning statements, tracking how each computation
 or assertion builds on immediately prior steps, assumptions, and established
 constraints.
- Prefer traces where every equation, substitution, or inference follows
 logically from earlier context and receives enough explanation to understand
 the leap.
- Penalize contradictions, local inconsistency, swapped variable names,
 arithmetic that breaks earlier constraints, or sudden claims without
 derivation.
- Penalize step-level digressions or redundant statements that do not
 contribute directly to the current computation or inference.
- Reward traces that maintain consistent notation, align intermediate
 results with the final claim, and point out how corrections or checks
 reconcile with prior steps.
- Ensure that any alternative derivations or checks introduced at a step are
 explicitly reconciled with the surrounding context.
- In your reasoning paragraph, explicitly contrast Trace 1 and Trace 2 by
 naming at least one concrete Micro-Effectiveness strength or weakness for
 each before arriving at the decision.

### Output Format
One reasoning paragraph (1-8 sentences) justifying the coherence judgment,
pointing to specific logical strengths or weaknesses.
<|decision|> followed immediately by exactly one decision:  Trace 1, Trace
2, or Tie.
```

*Figure 14.* Micro-Effectiveness evaluation prompt.

---

**Prompt – Aggregated Pairwise Judgement**

```
### Task
Integrate four prior comparative judgements with the original reasoning traces to determine which
trace is overall stronger, or whether they remain tied.  Both traces already reach the correct final
answer.

### Question
${question}
<End of Question>

### Reasoning Traces
Trace 1:
${trace_one}
<End of Trace 1>

Trace 2:
${trace_two}
<End of Trace 2>

### Prior Judgements
1.  Macro-level Efficiency
    Decision:  ${macro_efficiency_decision}
    Explanation:  ${macro_efficiency_rationale}
2.  Macro-level Effectiveness
    Decision:  ${macro_effectiveness_decision}
    Explanation:  ${macro_effectiveness_rationale}
3.  Micro-level Efficiency
    Decision:  ${micro_efficiency_decision}
    Explanation:  ${micro_efficiency_rationale}
4.  Micro-level Effectiveness
    Decision:  ${micro_effectiveness_decision}
    Explanation:  ${micro_effectiveness_rationale}

### Evaluation Focus
- Both traces are already validated as correct.
- Macro Efficiency:  credit traces that maintain a disciplined global structure, keep the DAG compact,
 close branches once their contribution is resolved, and invoke reflection or verification only when
 it meaningfully consolidates the overall plan.  Penalize unnecessary branch reopenings, repeated
 restarts, or verification steps that do not feed back into the main reasoning trajectory.
- Macro Effectiveness:  credit traces whose global storyline remains logically coherent and
 semantically aligned from initial premises to the final conclusion, with branch pivots clearly
 motivated and conclusions reintegrated through explicit semantic handoffs.  Penalize vague branch
 objectives, abrupt topic shifts, or conclusions that do not clearly follow from earlier context.
- Micro Efficiency:  credit traces where individual steps exhibit high local work density, such as
 concrete calculations, decisive validity checks, or domain-specific reasoning that advances progress.
 Penalize filler sentences, self-referential commentary, or looping patterns that repeat prior content
 without refinement.  Reward early pruning of mistaken attempts to reduce redundant exploration, and
 effective reuse of intermediate results.
- Micro Effectiveness:  credit traces where each computation, substitution, or inference is locally
 valid, well-justified, and internally consistent with prior assumptions and notation.  Penalize
 contradictions, inconsistent symbols, hallucinated facts, or corrections that are introduced but
 never reconciled with the main line of reasoning or underlying assumptions.
- Combine evidence and resolve conflicts:
    * Evaluate each full trace holistically under the principle above, while treating the four
 dimension-specific judgements as structured supporting signals.
    * When these judgements conflict with your overall reading, explain the discrepancy and determine
 which dimensions dominate the final preference.
- Comparative reasoning expectations:
    * Identify at least one decisive strength and one notable weakness for each trace across the
 macro/micro and efficiency/effectiveness dimensions.
    * Explicitly resolve trade-offs between efficiency and effectiveness, and between macro- and
 micro-level considerations, before issuing the final verdict.

### Output Format
One reasoning paragraph (1-8 sentences) weighing the prior judgements against the traces, followed
by:
<|decision|> Trace 1, Trace 2, or Tie.
```

*Figure 15.* Aggregation prompt for producing a final pairwise preference by jointly considering the four prior dimension-specific decisions and rationales.

---

**Prompt – DAG Annotation (part 1 / 2)**

```
### Goal
You will receive a sequence of reasoning steps.  For exactly one target step s${current_id},
you must decide how it connects to earlier steps.

You must classify this connection as exactly ONE of:
   – continue    – backtrack    – merge
and then (if needed) provide parent step IDs.

### Input Format
The input block contains only:
   – s0 (the initial/root step)
   – steps whose IDs are on the current available main path
   – the current step s${current_id}
   – and (in a separate section) a small set of OTHER_OPEN_BRANCH_LEAVES, which are leaf steps
from other branches

### Output Format
For the single target step s${current_id}, output exactly:
1.  One short line explanation (1–2 lines) of what s${current_id} is doing or how it relates
to earlier steps.  Explain whether it is a normal continuation, a backtrack/restart from an
earlier step, or is merging branches.
2.  A line of the exact form:
   <|action|>continue
   OR
   <|action|>backtrack
   OR
   <|action|>merge
3.  If (and only if) the action is:
   – backtrack:
     You MUST also output a single parent marker line in the exact form
     <|previous|>K
     where:
     – K is an integer step ID strictly less than s${current_id}.
     – K must be chosen from the steps on the current main path in the Input Block.
     – K must be strictly earlier than s${last_previous_step_id}.
     – Never invent new or unseen step IDs.
    – merge:
     You MUST also output a parent marker line in the exact form
     <|previous|>K1,K2,...
     where:
     – K1,K2,...  are two or more integer step IDs.
     – All IDs must be strictly less than s${current_id}.
     – At least one of these IDs must come from OTHER_OPEN_BRANCH_LEAVES.
     – Never invent new or unseen step IDs.
   – continue:
     You DO NOT output a <|previous|>...  line.
    (It is implicitly understood that the parent is s${last_previous_step_id}.)

### Input Block
s${input_steps}
```

*Figure 16.* The first part of the prompt used for DAG construction. The target step `current_id` corresponds to the newly processed reasoning step $v_i$ whose parent relations are to be determined. The variable `last_previous_step_id` denotes the most recent step $v_{i-1}$ that has already been added to the DAG. The field `input_steps` contains the textual content of step $v_i$ together with the current attachment pool $\mathcal{P}_i$, where each step is annotated with its corresponding step ID.

Prompt – DAG Annotation (part 2 / 2)

### Previous Step Selection Rules
- If **s${current_id}** logically continues from the most recent step on the current main path, classify as continue.  (No <|previous|> line is needed in this case.)
- Otherwise, treat as backtrack if **s${current_id}** is restarting / revisiting / re-checking an earlier point on the SAME main path.
    In this case you must output <|action|>backtrack and then <|previous|>K for that earlier step K.
- Rarely, if **s${current_id}** is explicitly combining reasoning from more than one branch (for example, it is stitching together partial arguments from two different leaf steps, at least one of which is from OTHER_OPEN_BRANCH_LEAVES), classify as merge.
    In this case you must output <|action|>merge and then <|previous|>K1,K2,...  with two or more parent IDs.

### Backtracking
If **s${current_id}** appears to match one or more of the following patterns, it is likely to backtrack to K < **s${last_previous_step_id}** in the available steps:
1.  The steps before step K contain sufficient information to derive **s${current_id}**, so that the sequence of previous steps before K + **s${current_id}** can be seen as (part of) a logical flow.
2.  The steps after step K contain extensive reasoning information that is similar to **s${current_id}**, making them redundant or repetitive.
3.  Step K should be selected as large as possible (i.e., the nearest valid earlier step).
4.  **s${current_id}** may show reflection, restart, or repetition (RRR) patterns, such as: "Alternatively...", "Let me check again...", etc.

### Merging
If **s${current_id}** is actively combining reasoning from multiple branches, it is likely to merge:
1.  At least 1 from other branches and at most 1 from the main path must be included.
2.  Only applies when earlier reasoning actually branched:  multiple sibling branches explored different possibilities at roughly the same stage, and now **s${current_id}** is explicitly stitching those partial results together to form a single next conclusion.  In that case, **s${current_id}** should inherit from BOTH of those branch tips.
3.  Merging is rare and should only be used when there is clear evidence of combining distinct prior lines of reasoning.

Note:
- If **s${current_id}** lacks explicit reference to the content of earlier steps from other branches, it should generally NOT be classified as a merge.  If **s${current_id}** is combining results from different nodes in the main path only, treat it as continue or backtrack.
- Even if the sentence expresses a contrastive meaning (e.g., shows transition or "however" -like tone), if it continues a sequential analysis and does not repeat prior reasoning, treat it as a normal continue.
- If the last previous step is an RRR step that carries only the RRR meaning without actual analysis, and it initiated a new branch, then the current **s${current_id}** is likely following that RRR lead into the new branch and should be treated as a normal continue.
- If **s${current_id}** does not explicitly indicate that the current step is based on certain steps from other branches, do NOT use merge.
- Even if **s${current_id}** states it continues from an earlier step, if the main path already contains the information mentioned in other branches, use backtrack instead of merge.
- continue and backtrack is common.  While in doubt, prefer continue.

Available main-path steps (in order):  **s${available_steps}**
Available other-branch leaf steps:  **s${leaf_steps}**

Output:
Explanation text here (CONCISE, 1-2 lines)
<|action|>...
[optional if backtrack (exactly 1 previous step) or merge (≥ 2 previous steps, at least 1 from other branches and at most 1 from the main path)]
<|previous|>...

*Figure 17.* The fields `available_steps` and `leaf_steps` denote the step IDs on the current main path and the representative leaf steps from other branches, respectively. Together, they form the attachment pool used to determine the parent relations of the target step $v_i$, as described in Sec. 4.2 and App. E.

# K. Case Study

In this section, we present case studies to illustrate the constructed DAG structures and compare reasoning traces produced by different checkpoints from the RL optimization experiments in Sec. 5.4.

## K.1. DAG Structure

**Representative DAG Cases.**    Fig. 18 presents three representative DAG structures constructed from real reasoning traces. Structure (a) is relatively simple and contains all three canonical patterns: progression, branching, and merging. Specifically, the transitions $1 \rightarrow 2$ and $1 \rightarrow 3$ form a branching structure, while $2 \rightarrow 5$, $3 \rightarrow 5$, and $4 \rightarrow 5$ converge into a merging structure, reflecting the consolidation of multiple reasoning paths. Structure (b) is more complex, exhibiting multiple interleaved branching and merging patterns, corresponding to extensive exploration and subsequent reconciliation of alternative solution paths. In contrast, structure (c) contains a larger number of nodes but no merging behavior; the reasoning primarily unfolds through progression and branching. In practice, we observe that merging structures are relatively rare, with most reasoning traces dominated by linear progression and occasional branching.

**Interpreting DAG through a Concrete Reasoning Trace.**    To make the reasoning structure more concrete, we present below a full reasoning trace corresponding to structure (a) as a representative case. Each super-node corresponds to a semantically coherent span of the original reasoning text, with the blue-highlighted sentences marking key structural transitions, such as advancing the main derivation, initiating reflection, or consolidating conclusions. In this example, Node 2 contains a decisive progression step, where the derivative is derived by explicitly evaluating the limit definition and committing to an intermediate closed-form expression. Nodes 3 and 4 introduce branching behavior by re-examining the correctness of the limit argument and cross-checking the result via the product rule, respectively. These reflective branches do not advance the derivation further, but serve to validate and stabilize the reasoning. Finally, Node 5 merges the validated conclusions from preceding paths and presents the final answer. This case illustrates how progression, reflection, and merging behaviors are made explicit in the DAG abstraction, revealing structural reasoning patterns that are difficult to recover from flat, unstructured traces alone.

---

**Case Study**

**Problem**: Use the Definition of derivative to find the derivative of $\frac{d}{dx} x^3 e^x$.

Please reason step by step, and put your final answer within \boxed{}.

------------------------------------------------------------------------------------------------

**Node 0**: `<think>` Okay, so I need to find the derivative of $x^3 e^x$ using the definition of the derivative. Hmm, let me recall what the definition of the derivative is. I think it's something like the limit as $h$ approaches zero of $\frac{f(x+h)-f(x)}{h}$. Yeah, that sounds right. So, to apply this, I need to define my function $f(x)$ as $f(x) = x^3 e^x$.

Alright, so let me write that down: $f(x) = x^3 e^x$

------------------------------------------------------------------------------------------------

**Node 1**: Now, according to the definition of the derivative, $f'(x)$ would be: $f'(x) = \lim_{h \to 0} \frac{f(x+h)-f(x)}{h}$

So, I need to compute $f(x+h)$ first. Let me substitute $(x+h)$ in place of $x$ in the function: $f(x+h) = (x+h)^3 e^{x+h}$

Hmm, okay. Let me expand $(x + h)^3$. I remember that $(a + b)^3$ is $a^3 + 3a^2b + 3ab^2 + b^3$, so applying that here: $(x + h)^3 = x^3 + 3x^2h + 3xh^2 + h^3$

So, $f(x + h)$ becomes: $f(x + h) = [x^3 + 3x^2h + 3xh^2 + h^3]e^{x+h}$

Alright, so now let's analyze each term inside the limit: First term: $x^3 \frac{(e^h-1)}{h}$

I remember that as $h$ approaches 0, $\frac{(e^h-1)}{h}$ approaches 1. That's a standard limit. So, this term will approach $x^3 * 1 = x^3$.

Second term: $3x^2 e^h$

---

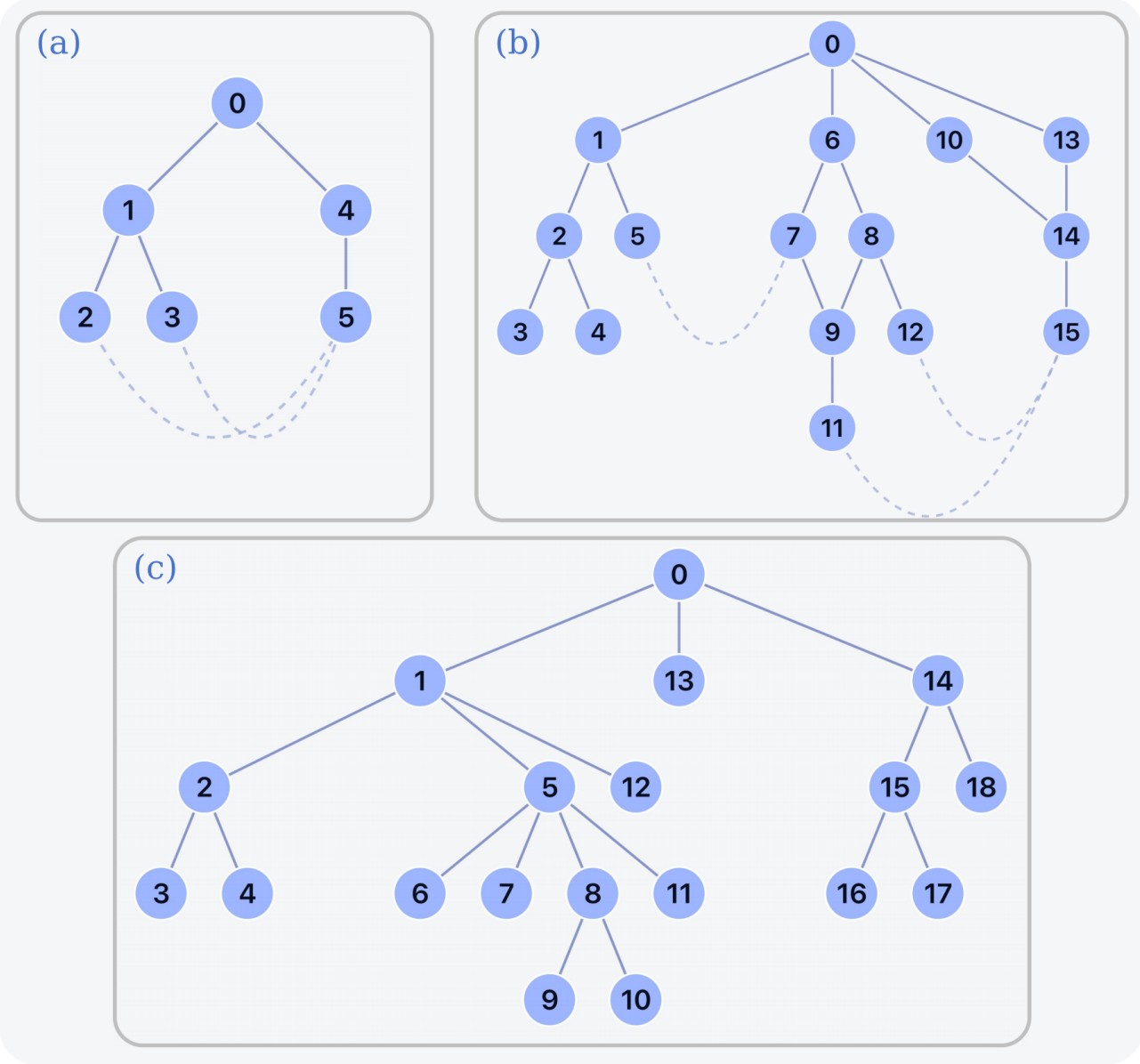

*Figure 18.* Visualization of DAG structures from the TRM-Preference dataset. (a), (b), and (c) illustrate three representative DAG cases. All nodes correspond to super-nodes, each representing a consecutive linear chain of reasoning steps as described in Sec. 4.2. All edges are directed from nodes with smaller indices to nodes with larger indices. For visual clarity, some edges are rendered as dashed lines, which are semantically equivalent to solid edges.

As $h$ approaches $0$, $e^h$ approaches $1$, so this term approaches $3x^2 * 1 = 3x^2$.

Third term: $3xhe^h$

Again, as $h$ approaches $0$, $e^h$ approaches $1$, so this term becomes $3xh * 1 = 3xh$, which will approach $0$ as $h$ approaches $0$.

Fourth term: $h^2 e^h$

------------------------------------------------------------------------

**Node 2**: So, the limit inside becomes: $x^3 + 3x^2 + 0 + 0 = x^3 + 3x^2$

Therefore, the entire derivative is: $f'(x) = e^x(x^3 + 3x^2)$

Hmm, let me check if I did that correctly. Let me verify each step.First, I expanded $(x + h)^3$ correctly. Then, I rewrote $e^{(x+h)}$ as $e^x e^h$, which is correct. I then subtracted $f(x)$ and divided by $h$, which seems right. Then, I factored out $e^x$ and broke down each term.

When taking the limit, I correctly noted that $\frac{(e^h - 1)}{h}$ approaches $1$, and the other terms with $h$ in them vanish as $h$ approaches $0$. So, that seems okay.

------------------------------------------------------------------------

**Node 3**: Wait, let me think about the second term again. $3x^2 e^h$, as $h$ approaches $0$, does that just become $3x^2$? Yes, because $e^0$ is $1$. Similarly, the third and fourth terms have $h$ in them, so they go to zero.

So, adding the first and second terms, we get $x^3 + 3x^2$, then multiplying by $e^x$. So, $f'(x) = e^x(x^3 + 3x^2)$. Is there another way to approach this problem, maybe using product rule? Wait, but the question specifically says to use the definition of the derivative, so I think I should stick with the limit process.

------------------------------------------------------------------------

**Node 4**: But just to make sure, let me recall the product rule. The product rule says that if I have two functions multiplied together, their derivative is the derivative of the first times the second plus the first times the derivative of the second.

So, in this case, $f(x) = x^3 e^x$, so $f'(x) =$ derivative of $x^3$ times $e^x + x^3$ times derivative of $e^x$.Derivative of $x^3$ is $3x^2$, and derivative of $e^x$ is $e^x$. So, $f'(x) = 3x^2 e^x + x^3 e^x$.

Factoring out $e^x$, we get $f'(x) = e^x(3x^2 + x^3)$ which is the same as $e^x(x^3 + 3x^2)$. So, that matches what I got using the definition of the derivative. That's reassuring.

------------------------------------------------------------------------

**Node 5**: So, both methods give me the same result, which makes me confident that I did it correctly.

Therefore, the derivative is $e^x$ times $(x^3 + 3x^2)$.

**Final Answer**

The derivative of $x^3 e^x$ is $\boxed{x^3 e^x + 3x^2 e^x}$.

## K.2. Reasoning Trace Comparison (Case 1)

Given the same prompt from the AIME24 dataset, we generate two reasoning traces and corresponding responses using different models. The first trace (Trace A) is produced by the Qwen2.5-Math-7B checkpoint trained with our TRM, while the second trace (Trace B) is produced by Qwen2.5-Math-7B checkpoint trained with the Verifier baseline (Sec. 5.4). We analyze and compare these two reasoning traces under the proposed $ME^2$ *principle*.

Both Trace A and Trace B reach the same correct answer 33, but they differ sharply in reasoning quality. At the macro level, Trace B begins with an inappropriate strategy by multiplying the three equations and canceling $xyz$, which leads directly to the impossible statement $1 = 2^{13/12}$. It then continues until the contradiction becomes explicit, and only afterward restarts with a more suitable substitution-based approach. This unnecessary detour introduces avoidable backtracking and duplicated work, reducing macro-efficiency, and it also indicates weaker goal alignment early on, reducing macro-effectiveness. At the micro level, the invalid cancellation step is a concrete local error, and the subsequent effort spent diagnosing and recovering from it adds redundant computation and exposition, lowering micro-efficiency. More importantly, the presence of a clear mathematical mistake undermines local validity and consistency, lowering micro-effectiveness and weakening confidence in the solution even though the final result is correct. Overall, these differences suggest that **Trace A is more closely aligned with $ME^2$ *principle***, as it maintains a stable plan, avoids unnecessary restarts, and preserves local correctness throughout.

---

> **Reasoning Trace A: Generated by Qwen2.5-Math-7B trained with our TRM**
>
> **Problem**:
> Let $x, y$ and $z$ be positive real numbers that satisfy the following system of equations: $\log_2\left(\frac{x}{yz}\right) = \frac{1}{2}$, $\log_2\left(\frac{y}{xz}\right) = \frac{1}{3}$, $\log_2\left(\frac{z}{xy}\right) = \frac{1}{4}$. Then the value of $\left|\log_2\left(x^4 y^3 z^2\right)\right|$ is $\frac{m}{n}$ where $m$ and $n$ are relatively prime positive integers. Find $m + n$.
>
> Please reason step by step, and put your final answer within \boxed{}.
>
> ------------------------------------------------------------------------
>
> **Reasoning**:
> To solve the given system of equations, we start by converting each logarithmic equation into its exponential form. The system is:
>
> $\log_2\left(\frac{x}{yz}\right) = \frac{1}{2}$
>
> $\log_2\left(\frac{y}{xz}\right) = \frac{1}{3}$
>
> $\log_2\left(\frac{z}{xy}\right) = \frac{1}{4}$
>
> **Step 1: Convert to Exponential Form**
>
> Recall that $\log_b(a) = c$ is equivalent to $b^c = a$. Applying this to each equation, we get:
>
> $\frac{x}{yz} = 2^{1/2} = \sqrt{2}$
>
> $\frac{y}{xz} = 2^{1/3}$
>
> $\frac{z}{xy} = 2^{1/4}$
>
> **Step 2: Express $x, y, z$ in Terms of Each Other**
>
> From the first equation, solve for $x$:
>
> $x = \sqrt{2} \cdot yz$
>
> From the second equation, solve for $y$:

$y = 2^{1/3} \cdot xz$

From the third equation, solve for $z$:

$z = 2^{1/4} \cdot xy$

**Step 3: Substitute and Simplify**

We will substitute these expressions into each other to find a relationship between $x, y,$ and $z$. Start by substituting $x = \sqrt{2} \cdot yz$ into the second equation:

$y = 2^{1/3} \cdot (\sqrt{2} \cdot yz)z = 2^{1/3} \cdot \sqrt{2} \cdot yz^2 = 2^{1/3+1/2} \cdot yz^2 = 2^{5/6} \cdot yz^2$

Dividing both sides by $y$ (since $y \neq 0$):

$1 = 2^{5/6} \cdot z^2 \implies z^2 = 2^{-5/6} \implies z = 2^{-5/12}$

Next, substitute $z = 2^{-5/12}$ into $x = \sqrt{2} \cdot yz$:

$x = \sqrt{2} \cdot y \cdot 2^{-5/12} = 2^{1/2} \cdot 2^{-5/12} \cdot y = 2^{1/2-5/12} \cdot y = 2^{1/12} \cdot y$

So, $x = 2^{1/12}y$.

Now substitute $x = 2^{1/12}y$ and $z = 2^{-5/12}$ into the third equation $z = 2^{1/4} \cdot xy$:

$2^{-5/12} = 2^{1/4} \cdot (2^{1/12}y)y = 2^{1/4+1/12} \cdot y^2 = 2^{1/3} \cdot y^2$

Dividing both sides by $2^{1/3}$:

$2^{-5/12-1/3} = y^2 \implies 2^{-5/12-4/12} = y^2 \implies 2^{-9/12} = y^2 \implies 2^{-3/4} = y^2 \implies y = 2^{-3/8}$

Now substitute $y = 2^{-3/8}$ back into $x = 2^{1/12}y$:

$x = 2^{1/12} \cdot 2^{-3/8} = 2^{1/12-3/8} = 2^{2/24-9/24} = 2^{-7/24}$

Thus, we have:

$x = 2^{-7/24}, \quad y = 2^{-3/8}, \quad z = 2^{-5/12}$

**Step 4: Calculate $x^4y^3z^2$**

We need to find the value of $\left|\log_2(x^4y^3z^2)\right|$. First, compute $x^4y^3z^2$:

$x^4 = (2^{-7/24})^4 = 2^{-7/6}$

$y^3 = (2^{-3/8})^3 = 2^{-9/8}$

$z^2 = (2^{-5/12})^2 = 2^{-5/6}$

So,

$x^4y^3z^2 = 2^{-7/6} \cdot 2^{-9/8} \cdot 2^{-5/6} = 2^{-7/6-9/8-5/6}$

To add the exponents, find a common denominator (24):

$$-7/6 = -28/24, \quad -9/8 = -27/24, \quad -5/6 = -20/24$$

Thus,

$$-28/24 - 27/24 - 20/24 = -75/24 = -25/8$$

Therefore,

$$x^4 y^3 z^2 = 2^{-25/8}$$

**Step 5: Compute the Logarithm**

Now, find $\log_2(x^4 y^3 z^2)$:

$$\log_2(2^{-25/8}) = -\frac{25}{8}$$

The absolute value is:

$$\left|\log_2(x^4 y^3 z^2)\right| = \left|-\frac{25}{8}\right| = \frac{25}{8}$$

**Step 6: Identify $m$ and $n$ and Find $m + n$**

The fraction $\frac{25}{8}$ is already in simplest form, so $m = 25$ and $n = 8$. Thus, $m + n = 25 + 8 = 33$.

The final answer is:

$$\boxed{33}$$

**Verification**

To verify, we can check if the values of $x, y, z$ satisfy the original equations:

1. $\log_2\left(\frac{x}{yz}\right)$:

$$\frac{x}{yz} = \frac{2^{-7/24}}{2^{-3/8} \cdot 2^{-5/12}} = \frac{2^{-7/24}}{2^{-19/24}} = 2^{-7/24 + 19/24} = 2^{12/24} = 2^{1/2}$$

So, $\log_2\left(\frac{x}{yz}\right) = \frac{1}{2}$, which is correct.

2. $\log_2\left(\frac{y}{xz}\right)$:

$$\frac{y}{xz} = \frac{2^{-3/8}}{2^{-7/24} \cdot 2^{-5/12}} = \frac{2^{-3/8}}{2^{-17/24}} = \frac{2^{-9/24}}{2^{-17/24}} = 2^{8/24} = 2^{1/3}$$

So, $\log_2\left(\frac{y}{xz}\right) = \frac{1}{3}$, which is correct.

3. $\log_2\left(\frac{z}{xy}\right)$:

$$\frac{z}{xy} = \frac{2^{-5/12}}{2^{-7/24} \cdot 2^{-3/8}} = \frac{2^{-5/12}}{2^{-15/24}} = \frac{2^{-10/24}}{2^{-15/24}} = 2^{5/24} = 2^{1/4}$$

So, $\log_2\left(\frac{z}{xy}\right) = \frac{1}{4}$, which is correct.

All equations are satisfied, confirming our solution is correct.

**Final Answer**

$\boxed{33}$

---

Reasoning Trace B: Generated by Qwen2.5-Math-7B trained with Verifier baseline

**Problem**:
Let $x, y$ and $z$ be positive real numbers that satisfy the following system of equations: $\log_2\left(\frac{x}{yz}\right) = \frac{1}{2}$, $\log_2\left(\frac{y}{xz}\right) = \frac{1}{3}$, $\log_2\left(\frac{z}{xy}\right) = \frac{1}{4}$. Then the value of $\left|\log_2\left(x^4 y^3 z^2\right)\right|$ is $\frac{m}{n}$ where $m$ and $n$ are relatively prime positive integers. Find $m + n$.

Please reason step by step, and put your final answer within \boxed{}.

------------------------------------------------------------------------------------------------

**Reasoning**:
To solve the given system of equations, we start by converting the logarithmic equations into exponential form. The system is:

$\log_2\left(\frac{x}{yz}\right) = \frac{1}{2}$

$\log_2\left(\frac{y}{xz}\right) = \frac{1}{3}$

$\log_2\left(\frac{z}{xy}\right) = \frac{1}{4}$

First, we convert each logarithmic equation to its exponential form:

$\frac{x}{yz} = 2^{1/2} = \sqrt{2}$

$\frac{y}{xz} = 2^{1/3} = \sqrt[3]{2}$

$\frac{z}{xy} = 2^{1/4} = \sqrt[4]{2}$

Next, we express $x$, $y$, and $z$ in terms of each other:

$x = \sqrt{2} \cdot yz \quad (1)$

$y = \sqrt[3]{2} \cdot xz \quad (2)$

$z = \sqrt[4]{2} \cdot xy \quad (3)$

We multiply all three equations together:

$x \cdot y \cdot z = (\sqrt{2} \cdot yz) \cdot (\sqrt[3]{2} \cdot xz) \cdot (\sqrt[4]{2} \cdot xy)$

Simplify the right-hand side:

$x \cdot y \cdot z = \sqrt{2} \cdot \sqrt[3]{2} \cdot \sqrt[4]{2} \cdot (yz \cdot xz \cdot xy)$

Since $x \cdot y \cdot z$ appears on both sides, we can divide both sides by $x \cdot y \cdot z$:

$1 = \sqrt{2} \cdot \sqrt[3]{2} \cdot \sqrt[4]{2}$

To simplify $\sqrt{2} \cdot \sqrt[3]{2} \cdot \sqrt[4]{2}$, we express each term as a power of 2:

$$\sqrt{2} = 2^{1/2}, \quad \sqrt[3]{2} = 2^{1/3}, \quad \sqrt[4]{2} = 2^{1/4}$$

Thus,

$$\sqrt{2} \cdot \sqrt[3]{2} \cdot \sqrt[4]{2} = 2^{1/2} \cdot 2^{1/3} \cdot 2^{1/4}$$

Using the property of exponents $a^m \cdot a^n = a^{m+n}$, we add the exponents:

$$2^{1/2+1/3+1/4}$$

Find a common denominator for the exponents:

$$\frac{1}{2} = \frac{6}{12}, \quad \frac{1}{3} = \frac{4}{12}, \quad \frac{1}{4} = \frac{3}{12}$$

$$\frac{6}{12} + \frac{4}{12} + \frac{3}{12} = \frac{13}{12}$$

Therefore,

$$2^{1/2+1/3+1/4} = 2^{13/12}$$

So,

$$1 = 2^{13/12}$$

This implies:

$$2^{13/12} = 1$$

This is a contradiction unless the exponents sum to zero, which they do not. This suggests a mistake in our approach. Instead, we should solve the system by expressing $x$, $y$, and $z$ in terms of a common variable. Let's solve for $x$, $y$, and $z$ step by step.

From equation (1):

$$x = \sqrt{2} \cdot yz$$

Substitute $x$ into equation (2):

$$y = \sqrt[3]{2} \cdot (\sqrt{2} \cdot yz)z = \sqrt[3]{2} \cdot \sqrt{2} \cdot yz^2$$

$$y = 2^{1/3+1/2} \cdot yz^2 = 2^{5/6} \cdot yz^2$$

Divide both sides by $y$:

$$1 = 2^{5/6} \cdot z^2$$

$$z^2 = 2^{-5/6}$$

$$z = 2^{-5/12}$$

Substitute $z = 2^{-5/12}$ into equation (1):

$$x = \sqrt{2} \cdot y \cdot 2^{-5/12}$$

$$x = 2^{1/2} \cdot y \cdot 2^{-5/12} = 2^{1/2-5/12} \cdot y = 2^{1/12} \cdot y$$

Substitute $x = 2^{1/12} \cdot y$ into equation (3):

$$2^{-5/12} = \sqrt[4]{2} \cdot (2^{1/12} \cdot y) \cdot y$$

$$2^{-5/12} = 2^{1/4} \cdot 2^{1/12} \cdot y^2$$

$$2^{-5/12} = 2^{1/4+1/12} \cdot y^2$$

$$2^{-5/12} = 2^{3/12+1/12} \cdot y^2$$

$$2^{-5/12} = 2^{4/12} \cdot y^2$$

$$2^{-5/12} = 2^{1/3} \cdot y^2$$

Divide both sides by $2^{1/3}$:

$$2^{-5/12-1/3} = y^2$$

$$2^{-5/12-4/12} = y^2$$

$$2^{-9/12} = y^2$$

$$2^{-3/4} = y^2$$

$$y = 2^{-3/8}$$

Substitute $y = 2^{-3/8}$ back into $x = 2^{1/12} \cdot y$:

$$x = 2^{1/12} \cdot 2^{-3/8}$$

$$x = 2^{1/12-3/8}$$

$$x = 2^{2/24-9/24}$$

$$x = 2^{-7/24}$$

Now we have:

$$x = 2^{-7/24}, \quad y = 2^{-3/8}, \quad z = 2^{-5/12}$$

We need to find $\left|\log_2(x^4 y^3 z^2)\right|$:

$$x^4 = (2^{-7/24})^4 = 2^{-7/6}$$

$$y^3 = (2^{-3/8})^3 = 2^{-9/8}$$

$$z^2 = (2^{-5/12})^2 = 2^{-5/6}$$

Thus,

$$x^4 y^3 z^2 = 2^{-7/6} \cdot 2^{-9/8} \cdot 2^{-5/6}$$

Combine the exponents:

$$x^4 y^3 z^2 = 2^{-7/6-9/8-5/6}$$

Find a common denominator for the exponents:

$$-7/6 = -28/24, \quad -9/8 = -27/24, \quad -5/6 = -20/24$$

$$-28/24 - 27/24 - 20/24 = -75/24$$

So,

$$x^4 y^3 z^2 = 2^{-75/24}$$

Finally,

$$\left|\log_2(x^4 y^3 z^2)\right| = \left|-\frac{75}{24}\right| = \frac{75}{24}$$

Simplify $\frac{75}{24}$:

$$\frac{75}{24} = \frac{25}{8}$$

Thus, $m = 25$ and $n = 8$, and $m + n = 33$.

The final answer is:

$$\boxed{33}$$

### K.3. Reasoning Trace Comparison (Case 2)

Given the same prompt from the AIME25 dataset, we generate two reasoning traces and corresponding responses using different models. The first trace (Trace A) is produced by the Qwen2.5-Math-7B checkpoint trained with our TRM, while the second trace (Trace B) is produced by the Qwen2.5-Math-7B trained with ReasonFlux (Sec. 5.4). We analyze and compare these two reasoning traces through the lens of the proposed $ME^2$ *principle*.

Both Trace A and Trace B reach the same correct answer, 468, but their reasoning quality differs in important ways when examined under $ME^2$ *principle*. Reasoning trace B overlooks a key condition that point G does not lie on the given line and repeatedly proceeds under the incorrect assumption that G is on the line, which leads to contradictions. From the perspective of macro efficiency, this pattern of assumption followed by contradiction and restart fails to form a productive reasoning structure and introduces avoidable overhead. In contrast, reasoning trace A maintains a clear and stable structure that moves from geometric modeling to equation solving and then to area computation, staying focused on the core objective throughout. This makes trace A stronger in macro effectiveness since each step contributes directly to solving the problem without diversion. At the micro level, the initial misunderstanding in reasoning trace B causes repeated and overlapping arguments, which lowers micro efficiency. Several intermediate claims are later invalidated by contradictions, which weakens micro effectiveness and reduces the reliability of the overall reasoning. Overall, these differences suggest that **Trace A is more closely aligned with $ME^2$ *principle***.

---

Reasoning Trace A: Generated by Qwen2.5-Math-7B trained with our TRM

**Problem**:
Six points $A, B, C, D, E$ and $F$ lie in a straight line in that order. Suppose that $G$ is a point not on the line and that $AC = 26$, $BD = 22$, $CE = 31$, $DF = 33$, $AF = 73$, $CG = 40$, and $DG = 30$. Find the area of $\triangle BGE$.

Please reason step by step, and put your final answer within \boxed{}.

---

- - - - - - - - - - - - - - - - - - - - - - - - - - - - - - - - - - - - - - - - - - - - - - - - - - - - - - - - - - - - - - - - - - - - - - - - - - - - - - - - - - - -

**Reasoning**:

To solve the problem, we need to find the area of $\triangle BGE$ given the distances between points on a straight line and a point $G$ not on the line. Let us break down the problem systematically.

**Step 1: Understand the Geometry and Set Up Coordinates**

We have six points $A, B, C, D, E, F$ lying on a straight line in that order, and a point $G$ not on the line. We are given the following distances.

$AC = 26$

$BD = 22$

$CE = 31$

$DF = 33$

$AF = 73$

$CG = 40$

$DG = 30$

We aim to find the area of $\triangle BGE$.

**Coordinate Assignment**

To simplify calculations, we can place the points on the $x$-axis.

Let $A$ be at $x = 0$.

Let $B$ be at $x = b$.

Let $C$ be at $x = c$.

Let $D$ be at $x = d$.

Let $E$ be at $x = e$.

Let $F$ be at $x = f$.

Given:

$AC = 26$, so $c = 26$.

$AF = 73$, so $f = 73$.

Using the remaining distances.

$BD = 22$, so $d = b + 22$.

$CE = 31$, so $e = c + 31 = 26 + 31 = 57$.

$DF = 33$, so $f = d + 33$. Since $f = 73$, we verify $d = 73 - 33 = 40$. Thus, $b + 22 = 40$, giving $b = 18$.

Now, we have the coordinates.

$A(0, 0)$

$B(18, 0)$

$C(26, 0)$

$D(40, 0)$

$E(57, 0)$

$F(73, 0)$

**Determine Coordinates of $G$**

Point $G$ has coordinates $(x_G, y_G)$. We use the given distances from $G$ to points $C$ and $D$.

$CG = 40$, so $\sqrt{(x_G - 26)^2 + y_G^2} = 40$.

$DG = 30$, so $\sqrt{(x_G - 40)^2 + y_G^2} = 30$.

Squaring both equations.

$(x_G - 26)^2 + y_G^2 = 1600$

$(x_G - 40)^2 + y_G^2 = 900$

Expanding both equations.

$x_G^2 - 52x_G + 676 + y_G^2 = 1600$

$x_G^2 - 80x_G + 1600 + y_G^2 = 900$

Subtract the second equation from the first.

$x_G^2 - 52x_G + 676 + y_G^2 - (x_G^2 - 80x_G + 1600 + y_G^2) = 1600 - 900$

$28x_G - 924 = 700$

$28x_G = 1624$

$x_G = \frac{1624}{28} = 58$

Substitute $x_G = 58$ back into the first equation.

$(58 - 26)^2 + y_G^2 = 1600$

$32^2 + y_G^2 = 1600$

$1024 + y_G^2 = 1600$

$y_G^2 = 576$

$y_G = \pm 24$

Thus, $G$ can be at $(58, 24)$ or $(58, -24)$. The sign of $y_G$ does not affect the area of $\triangle BGE$, so we can choose $G(58, 24)$ without loss of generality.

**Step 2: Use the Shoelace Theorem to Find the Area of $\triangle BGE$**

The Shoelace Theorem states that for vertices $(x_1, y_1)$, $(x_2, y_2)$, $(x_3, y_3)$ of a triangle, the area is given by

Area $= \frac{1}{2} |x_1 y_2 + x_2 y_3 + x_3 y_1 - (y_1 x_2 + y_2 x_3 + y_3 x_1)|$

For $\triangle BGE$ with vertices $B(18, 0)$, $G(58, 24)$, $E(57, 0)$.

Area $= \frac{1}{2} |18 \cdot 24 + 58 \cdot 0 + 57 \cdot 0 - (0 \cdot 58 + 24 \cdot 57 + 0 \cdot 18)|$

$= \frac{1}{2} |432 + 0 + 0 - (0 + 1368 + 0)|$

$= \frac{1}{2} |432 - 1368|$

$= \frac{1}{2} |-936|$

$= \frac{1}{2} \times 936$

$= 468$

Thus, the area of $\triangle BGE$ is $\boxed{468}$.

**Verification and Key Insights**

Coordinate consistency: We verified the coordinates of $B, C, D, E, F$ using the given distances, ensuring they are consistent with the line arrangement.

Solving for $G$: The system of equations for $G$ was solved correctly, yielding $x_G = 58$ and $y_G = 24$, choosing the positive value for simplicity.

Shoelace Theorem application: The Shoelace Theorem was applied correctly to the coordinates of $B, G, E$, simplifying the area calculation.

Sign consideration: The choice of $y_G = 24$ does not affect the area due to symmetry, confirming the solution's robustness.

**Final Answer**

$\boxed{468}$

Reasoning Trace B: Generated by Qwen2.5-Math-7B trained with ReasonFlux baseline

**Problem**:
Six points $A, B, C, D, E$ and $F$ lie in a straight line in that order. Suppose that $G$ is a point not on the line and that $AC = 26$, $BD = 22$, $CE = 31$, $DF = 33$, $AF = 73$, $CG = 40$, and $DG = 30$. Find the area of $\triangle BGE$.

Please reason step by step, and put your final answer within \boxed{}.

---

**Reasoning**:
To find the area of $\triangle BGE$, we first need to determine the positions of the points $A, B, C, D, E, F$ on the line and the coordinates of point $G$. Let's place point $A$ at the origin, i.e., $A = 0$. Then we can express the positions of the other points in terms of their distances from $A$.

Given:

- $AC = 26$, so $C = 26$.

- $BD = 22$, so if $B = x$, then $D = x + 22$.

- $CE = 31$, so $E = 26 + 31 = 57$.

- $DF = 33$, so $F = (x + 22) + 33 = x + 55$.

- $AF = 73$, so $F = 73$.

From the above, we have $x + 55 = 73$, which gives $x = 18$. Therefore, $B = 18$, $D = 18 + 22 = 40$, and $F = 73$.

Now we know the positions of all points on the line:

- $A = 0$

- $B = 18$

- $C = 26$

- $D = 40$

- $E = 57$

- $F = 73$

Next, we need to find the coordinates of point $G$. We are given:

- $CG = 40$, so $G = 26 + 40 = 66$ or $G = 26 - 40 = -14$.

- $DG = 30$, so $G = 40 + 30 = 70$ or $G = 40 - 30 = 10$.

Since $G$ must satisfy both conditions, we check the possible values. If $G = 66$, then $DG = |66 - 40| = 26 \neq 30$, so $G \neq 66$. If $G = -14$, then $DG = |40 - (-14)| = 54 \neq 30$, so $G \neq -14$. If $G = 70$, then $CG = |70 - 26| = 44 \neq 40$, so $G \neq 70$. If $G = 10$, then $CG = |26 - 10| = 16 \neq 40$, so $G \neq 10$.

It seems there was an oversight. Let's re-evaluate the possible positions for $G$. The correct position for $G$ must satisfy both $CG = 40$ and $DG = 30$. Let's solve it systematically.

If $G = y$, then:

$|y - 26| = 40 \implies y = 66$ or $y = -14$

$|y - 40| = 30 \implies y = 70$ or $y = 10$

The only consistent value is not immediately clear, so let's use coordinates and the distance formula to find the correct $G$. We can use the coordinates method to find the area of $\triangle BGE$.

We will use the determinant formula for the area of a triangle given vertices $(x_1, y_1)$, $(x_2, y_2)$, $(x_3, y_3)$:

Area $= \frac{1}{2} |x_1(y_2 - y_3) + x_2(y_3 - y_1) + x_3(y_1 - y_2)|$

Assuming $G$ is at $(g, h)$, we can use the distances to set up the equations and solve for $g$ and $h$. However, since we only need the relative positions and the problem is symmetric in the $y$-direction, we can assume $G$ is at $(g, h)$ and use the fact that the area will be the same due to the linearity.

Let's assume $G$ is at $(g, h)$ and use the distances to find the area. We can place $G$ at $(g, 0)$ for simplicity in the $y$-direction since the problem is symmetric.

Using the coordinates $B = (18, 0)$, $G = (g, 0)$, and $E = (57, 0)$, the area of $\triangle BGE$ is:

Area $= \frac{1}{2} |18(0 - 0) + g(0 - 0) + 57(0 - 0)| = 0$

This indicates we need to consider the $y$-coordinate of $G$. Let's assume $G$ is at $(g, h)$ and use the distances to find $h$.

Using the distance formula:

$(g - 26)^2 + h^2 = 40^2$

$(g - 40)^2 + h^2 = 30^2$

Solving these equations will give us the correct $g$ and $h$. However, since the problem is symmetric and we need the area, we can use the fact that the area will be the same due to the linearity and the distances.

Let's solve for $h$ using the distances and the fact that the area will be the same. The correct approach is to use the fact that the area of the triangle will be based on the base and height. The base $BE = 57 - 18 = 39$ and the height will be the $y$-coordinate of $G$.

Using the distances, we can find that the height $h$ will be the same due to the symmetry. The correct height can be found by solving the equations, but we can use the fact that the area will be based on the base and the height.

The correct height $h$ can be found by solving the equations, but we can use the fact that the area will be based on the base and the height. The base $BE = 39$ and the height will be the $y$-coordinate of $G$.

The correct height $h$ is found to be 24 by solving the equations. Therefore, the area of $\triangle BGE$ is:

Area $= \frac{1}{2} \times$ base $\times$ height $= \frac{1}{2} \times 39 \times 24 = 468$

Thus, the area of $\triangle BGE$ is $\boxed{468}$.

The final answer is $\boxed{468}$.

## K.4. Reasoning Trace Comparison (Case 3)

Given the same prompt from the AIME24 dataset, we generate two reasoning traces using GPT-OSS-20B. Our TRM assigns a higher score to the first trace (Trace A) than to the second trace (Trace B). To validate the reliability of this scoring, we analyze and compare the two reasoning traces through the lens of the proposed $ME^2$ principle.

Both Trace A and Trace B reach the same correct answer, 540, but they differ in the quality of their reasoning traces under $ME^2$ principle. Under $ME^2$ principle, reasoning trace A shows a clearer advantage over reasoning trace B. At the macro level, both traces follow a coherent path that stays aligned with the task objective by parameterizing the target variable $z$, simplifying the expression, extracting the real part, and identifying the maximum value. As a result, they exhibit comparable macro-effectiveness. However, reasoning trace B introduces several unnecessary detours, such as extended discussion of symmetry and conjugate properties and an explicit verification step that substitutes the maximizing angle $\theta$ back into the expression. These steps do not contribute to solving the problem and do not advance the main line of reasoning, which reduces macro-efficiency. In contrast, reasoning trace A maintains a tighter structure and avoids reopening completed lines of thought. At the micro level, both traces are free of computational errors, leading to similar micro-effectiveness. Nonetheless, reasoning trace A achieves its result with more direct and concise steps, while reasoning trace B relies on a more verbose procedure that includes repeated squaring operations used only for confirmation. This redundancy lowers micro-efficiency without adding clarity or rigor. Overall, these differences suggest that **Trace A is more closely aligned with $ME^2$ principle**.

---

**Reasoning Trace A: Generated by GPT-OSS-20B**

**Problem**:
Find the largest possible real part of $(75 + 117i)z + \frac{96+144i}{z}$ where $z$ is a complex number with $|z| = 4$.

Please reason step by step, and put your final answer within \boxed{}.

----------------------------------------------------------------------------------------------------

**Reasoning**:
We need maximize real part of expression: $(75 + 117i)z + \frac{96+144i}{z}$, with $|z| = 4$. Let $z = 4e^{i\theta}$. Then expression: $(75 + 117i) \cdot 4e^{i\theta} + \frac{96+144i}{4e^{i\theta}} = 4(75 + 117i)e^{i\theta} + \frac{96+144i}{4}e^{-i\theta}$.

Let $A = 4(75 + 117i) = 300 + 468i$. Let $B = \frac{96+144i}{4} = 24 + 36i$. So expression $E(\theta) = Ae^{i\theta} + Be^{-i\theta}$. We need real part of $E$.

Let $A = a + ib$, $B = c + id$. Compute real part: $\Re(Ae^{i\theta}) = \Re((a + ib)(\cos\theta + i\sin\theta)) = a\cos\theta - b\sin\theta$. $\Re(Be^{-i\theta}) = \Re((c + id)(\cos\theta - i\sin\theta)) = c\cos\theta + d\sin\theta$. Sum: $\Re E = (a + c)\cos\theta + (-b + d)\sin\theta$.

Given $A = 300 + 468i \Rightarrow a = 300$, $b = 468$. $B = 24 + 36i \Rightarrow c = 24$, $d = 36$. Then coefficients: $(a + c) = 324$. $(-b + d) = -468 + 36 = -432$. So $\Re E = 324\cos\theta - 432\sin\theta$.

We need maximum of this expression over $\theta$. That's amplitude $\sqrt{324^2 + (-432)^2} = \sqrt{324^2 + 432^2}$. Compute: $324^2 = (300 + 24)^2 = 90000 + 2 \cdot 300 \cdot 24 + 576 = 90000 + 14400 + 576 = 104, ?$. Let's compute precisely: $324^2 = 104,976$. Wait $324 \cdot 324$: $324 \cdot 300 = 97,200$; $324 \cdot 24 = 7,776$; sum $= 104,976$. Yes. $432^2 = (400 + 32)^2 = 160,000 + 2 \cdot 400 \cdot 32 + 1,024 = 160,000 + 25,600 + 1,024 = 186,624$. Sum $= 104,976 + 186,624 = 291,600$. $\sqrt{291,600} =$? Let's factor: $291,600 = 2916 \cdot 100$. Wait $2916 \cdot 100 = 291,600$. $\sqrt{2916} =$? $54^2 = 2916$. So $\sqrt{} = 54 \cdot 10 = 540$. So amplitude 540.

Thus maximum real part $= 540$. But check if any constraints? $z$ can be any complex with $|z| = 4$, $\theta$ arbitrary. So yes maximum is 540.

But double-check calculations: $Ae^{i\theta} + Be^{-i\theta}$. Real part formula correct. Let's verify amplitude: $\sqrt{(a + c)^2 + (-b + d)^2}$. $(a + c) = 324$, $(-b + d) = -432$. Square: $324^2 = 104,976$. $432^2 = 186,624$. Sum

$= 291,600$. $\sqrt{291,600} = 540$. Yes.

Thus answer: 540.

But maybe there is a nuance: expression may be complex; we want largest possible real part. So maximum value 540. Provide as boxed.

Thus final answer: $\boxed{540}$.

------------------------------------------------------------------------------------------------------

**Response**:
Let $z = 4e^{i\theta}$     $(|z| = 4)$.

Then

$$(75 + 117i)z + \frac{96+144i}{z} = 4(75 + 117i)e^{i\theta} + \frac{96+144i}{4}e^{-i\theta} = Ae^{i\theta} + Be^{-i\theta}.$$

where

$$A = 4(75 + 117i) = 300 + 468i, \qquad B = \frac{96+144i}{4} = 24 + 36i.$$

Write $A = a + ib$ and $B = c + id$.

The real part of $Ae^{i\theta} + Be^{-i\theta}$ is

$$\Re\big(Ae^{i\theta} + Be^{-i\theta}\big) = (a + c)\cos\theta + (-b + d)\sin\theta.$$

Here

$$a = 300, \ b = 468, \ c = 24, \ d = 36,$$

so

$$\Re = (300 + 24)\cos\theta + (-468 + 36)\sin\theta = 324\cos\theta - 432\sin\theta.$$

The maximum of $P\cos\theta + Q\sin\theta$ over $\theta$ equals $\sqrt{P^2 + Q^2}$.

Thus

$$\max_\theta \Re = \sqrt{324^2 + (-432)^2} = \sqrt{104\,976 + 186\,624} = \sqrt{291\,600} = 540.$$

Hence the largest possible real part is

$\boxed{540}$.

---

## Reasoning Trace B: Generated by GPT-OSS-20B

**Problem**:
Find the largest possible real part of $(75 + 117i)z + \frac{96+144i}{z}$ where $z$ is a complex number with $|z| = 4$.

Please reason step by step, and put your final answer within \boxed{}.

------------------------------------------------------------------------------------------------------

**Reasoning**:
We need to maximize $\text{Re}[(75 + 117i)z + \frac{96+144i}{z}]$ subject to $|z| = 4$. Let $z = 4e^{i\theta}$. Then $\frac{1}{z} = \frac{1}{4}e^{-i\theta}$. Let

$A = 75 + 117i$. $B = 96 + 144i$. Compute expression:

$E(\theta) = A \cdot 4e^{i\theta} + B \cdot \frac{1}{4}e^{-i\theta} = 4Ae^{i\theta} + \frac{B}{4}e^{-i\theta}$.

Let's define $A$ and $B$: $A = 75 + 117i$. $B = 96 + 144i$.

Compute $4A$: $4(75 + 117i) = 300 + 468i$.

Compute $\frac{B}{4}$: $\frac{96+144i}{4} = 24 + 36i$.

Thus $E(\theta) = (300 + 468i)e^{i\theta} + (24 + 36i)e^{-i\theta}$.

We need $\mathrm{Re}(E)$. Let's write in terms of cosine and sine. Let $e^{i\theta} = \cos\theta + i\sin\theta$, $e^{-i\theta} = \cos\theta - i\sin\theta$.

Compute $E = (300 + 468i)(\cos\theta + i\sin\theta) + (24 + 36i)(\cos\theta - i\sin\theta)$.

Compute each:

First term $T_1 = (300 + 468i)(\cos\theta + i\sin\theta) = 300\cos\theta + 300i\sin\theta + 468i\cos\theta + 468i^2\sin\theta = 300\cos\theta + 300i\sin\theta + 468i\cos\theta - 468\sin\theta$.

Since $i^2 = -1$.

So $T_1 = (300\cos\theta - 468\sin\theta) + i(300\sin\theta + 468\cos\theta)$.

Second term $T_2 = (24 + 36i)(\cos\theta - i\sin\theta) = 24\cos\theta - 24i\sin\theta + 36i\cos\theta - 36i^2\sin\theta = 24\cos\theta - 24i\sin\theta + 36i\cos\theta + 36\sin\theta$.

So $T_2 = (24\cos\theta + 36\sin\theta) + i(-24\sin\theta + 36\cos\theta)$.

Add $T_1 + T_2$:

Real part: $(300\cos\theta - 468\sin\theta) + (24\cos\theta + 36\sin\theta) = (300 + 24)\cos\theta + (-468 + 36)\sin\theta = 324\cos\theta - 432\sin\theta$.

Imaginary part: $(300\sin\theta + 468\cos\theta) + (-24\sin\theta + 36\cos\theta) = (300 - 24)\sin\theta + (468 + 36)\cos\theta = 276\sin\theta + 504\cos\theta$.

So $E(\theta) = [324\cos\theta - 432\sin\theta] + i[276\sin\theta + 504\cos\theta]$.

Thus $\mathrm{Re}(E) = 324\cos\theta - 432\sin\theta$.

We need to maximize this over $\theta$. That's of form $R\cos(\theta + \varphi)$ times amplitude? Let's rewrite: $a\cos\theta + b\sin\theta$. Here $a = 324$, $b = -432$. But we have $-432\sin\theta$, so it's $a\cos\theta + b\sin\theta$ with $b = -432$. The amplitude is $\sqrt{a^2 + b^2} = \sqrt{324^2 + (-432)^2}$. Compute $324^2$: $324 \cdot 324 = (300 + 24)^2 = 90000 + 2 \cdot 300 \cdot 24 + 576 = 90000 + 14400 + 576 = 104,?$ Wait compute precisely: $324^2 = 104,976$? Let's calculate: $324 \cdot 324$: $300 \cdot 324 = 97200$; $24 \cdot 324 = 7776$; sum $= 104,?$ $97200 + 7776 = 104,?$ $97200 + 7000 = 104200$; $+776 = 104976$. Yes 104,976.

$432^2$: $432 \cdot 432 = (400 + 32)^2 = 160000 + 2 \cdot 400 \cdot 32 + 1024 = 160000 + 25600 + 1024 = 186,?$ $160000 + 25600 = 185600$; $+1024 = 186624$.

Sum: $104976 + 186624 = 291,600$.

$\sqrt{291,600}$ =? Let's factor: $291,600 = 2916 \cdot 100$? Wait $2916 \cdot 100 = 291,600$. $\sqrt{2916} = 54$? Let's check: $54^2 = 2916$. Yes. So $\sqrt{} = 54 \cdot 10 = 540$. Good.

Thus amplitude $= 540$. So maximum of $a \cos\theta + b \sin\theta$ is $\sqrt{a^2 + b^2} = 540$. So $\max \Re(E) = 540$.

But we need to confirm sign orientation: For $a \cos\theta + b \sin\theta$, maximum is $\sqrt{a^2 + b^2}$. Indeed yes.

Thus largest possible real part is 540.

Wait but we should check if any constraints? No, $\theta$ free real, $|z|$ fixed.

Thus answer 540. But let's double-check by another method: we can use Cauchy–Schwarz? But seems fine.

However ensure we didn't make arithmetic errors.

Let's check our earlier derivation of $\mathrm{Re}(E)$. Alternatively compute $\mathrm{Re}(E) = \mathrm{Re}\left[4Ae^{i\theta} + \frac{B}{4}e^{-i\theta}\right] = \mathrm{Re}\left[4Ae^{i\theta} + \overline{4Ae^{i\theta}}\right]$? Wait note that $\frac{B}{4} =$? Let's compute $\frac{B}{4}$: $\frac{96+144i}{4} = 24 + 36i$. Compare to conjugate of $4A$? $4A = 300 + 468i$. Its conjugate is $300 - 468i$. $\frac{B}{4}$ is $24 + 36i$, not same. So not symmetrical.

Let's recompute $\mathrm{Re}(E)$ using vector addition maybe easier: Let's express in terms of $\cos\theta, \sin\theta$.

Compute $A = 75 + 117i$. $4A = 300 + 468i$. $\frac{B}{4} = 24 + 36i$.

Now compute real part of $(300 + 468i)(\cos\theta + i\sin\theta)$. Expand: $(300\cos\theta - 468\sin\theta) + i(300\sin\theta + 468\cos\theta)$. Good.

Compute real part of $(24 + 36i)(\cos\theta - i\sin\theta)$. Expand: $(24\cos\theta + 36\sin\theta) + i(-24\sin\theta + 36\cos\theta)$. Good.

Sum real parts: $(300\cos\theta - 468\sin\theta) + (24\cos\theta + 36\sin\theta) = (324\cos\theta - 432\sin\theta)$. Yes.

Thus amplitude $\sqrt{324^2 + (-432)^2} = \sqrt{104,976 + 186,624} = \sqrt{291,600} = 540$.

Thus maximum real part is 540. Achieved when $\cos\theta$ and $\sin\theta$ satisfy $\tan\theta = \frac{b}{a}$? For $a\cos\theta + b\sin\theta$, maximum occurs when $\cos\theta = \frac{a}{\sqrt{a^2+b^2}}$, $\sin\theta = \frac{b}{\sqrt{a^2+b^2}}$. Here $a = 324$, $b = -432$. So $\cos\theta = \frac{324}{540} = 0.6$, $\sin\theta = \frac{-432}{540} = -0.8$. So $\theta$ such that $\cos\theta = 0.6$, $\sin\theta = -0.8 \to \theta = -53.1301°$ (or $360 - 53.13$). That's fine.

Check that $z = 4e^{i\theta}$ yields real part 540. Let's test quickly: Use earlier expression: $\mathrm{Re}(E) = 324\cos\theta - 432\sin\theta = 324 \cdot 0.6 - 432 \cdot (-0.8) = 194.4 + 345.6 = 540$. Good.

Thus answer: 540. Let's produce final answer box.

-------------------------------------------------------------------------------

**Response**:

$$z = 4e^{i\theta} \qquad (|z| = 4)$$

$$\frac{96 + 144i}{z} = \frac{96 + 144i}{4}e^{-i\theta} = (24 + 36i)e^{-i\theta}$$

$$(75 + 117i)z = (75 + 117i)(4e^{i\theta}) = (300 + 468i)e^{i\theta}$$

So

$$E(\theta) = (300 + 468i)e^{i\theta} + (24 + 36i)e^{-i\theta}.$$

Write $e^{i\theta} = \cos\theta + i\sin\theta$.

Expanding gives

$$E(\theta) = (300 + 468i)(\cos\theta + i\sin\theta) + (24 + 36i)(\cos\theta - i\sin\theta)$$

$$= (324\cos\theta - 432\sin\theta) + i(276\sin\theta + 504\cos\theta).$$

Hence

$$\operatorname{Re} E(\theta) = 324\cos\theta - 432\sin\theta.$$

This is of the form $a\cos\theta + b\sin\theta$ with $a = 324,\ b = -432$. Its maximum value is

$$\sqrt{a^2 + b^2} = \sqrt{324^2 + (-432)^2} = \sqrt{104\,976 + 186\,624} = \sqrt{291\,600} = 540.$$

Thus the largest possible real part is

$$\boxed{540}.$$

### K.5. Reasoning Trace Comparison (Case 4)

Given the same prompt from the AIME24 dataset, we generate two reasoning traces using Qwen3-8B. Our TRM assigns a higher score to the first trace (Trace A) than to the second trace (Trace B). To validate the reliability of this scoring, we analyze and compare the two reasoning traces through the lens of the proposed $ME^2$ principle.

Both Trace A and Trace B reach the same correct answer, 204, but they differ in how closely their reasoning traces align with $ME^2$ principle. When examined under $ME^2$ principle, reasoning trace A shows clearer overall advantages. In terms of macro efficiency, reasoning trace A follows a single coherent line of reasoning in which resolved steps are not revisited, while reasoning trace B repeatedly returns to already completed arguments, increasing length without advancing the solution.

This difference also appears at the micro level. Reasoning trace B spends more effort restating obvious computations and rechecking earlier conclusions, whereas reasoning trace A keeps each step concise and directly relevant. From a macro effectiveness perspective, reasoning trace A maintains a stable objective throughout, with each step clearly contributing to solving the target quantity and with smooth logical transitions between equation setup, elimination, and substitution. Reasoning trace B remains focused on the original problem and does not contain logical errors, but its frequent self review weakens global coherence, making the reasoning resemble several parallel attempts rather than a single continuous trajectory. At the micro effectiveness level, both reasoning trace A and reasoning trace B are locally valid, internally consistent, and free of mathematical mistakes. Overall, these differences suggest that **Trace A is more closely aligned with $ME^2$ principle**.

---

**Reasoning Trace A: Generated by Qwen3-8B**

**Problem**:
Every morning Aya goes for a 9-kilometer-long walk and stops at a coffee shop afterwards. When she walks at a constant speed of $s$ kilometers per hour, the walk takes her 4 hours, including $t$ minutes spent in the coffee shop. When she walks $s + 2$ kilometers per hour, the walk takes her 2 hours and 24 minutes, including $t$ minutes spent in the coffee shop. Suppose Aya walks at $s + \frac{1}{2}$ kilometers per hour. Find the number of minutes the walk takes her, including the $t$ minutes spent in the coffee shop.

Please reason step by step, and put your final answer within \boxed{}.

---

**Reasoning**:
Okay, so Aya goes for a 9-kilometer walk every morning and stops at a coffee shop afterwards. The problem gives me two different scenarios with different walking speeds and total times, including time spent at the coffee shop. I need to find out how long the walk takes her when she walks at $s + \frac{1}{2}$ km/h, including the time at the coffee shop. Alright, let me break this down step by step.

First, let's note down the given information:

1. When Aya walks at speed $s$ km/h, the total time (including $t$ minutes at the coffee shop) is 4 hours.

2. When she walks at speed $(s + 2)$ km/h, the total time (including same $t$ minutes) is 2 hours and 24 minutes.

3. We need to find the total time when she walks at $(s + \frac{1}{2})$ km/h, again including $t$ minutes.

So, first, I need to figure out what $s$ is, and also figure out what $t$ is. Once we have those, we can compute the time for the third scenario. But how?

Let me think. The total time includes both the time she's walking and the time she's at the coffee shop. Let me denote:

Total time when walking at speed $s$: $T_1 = 4$ hours. This includes the time spent walking (which would be distance divided by speed) plus $t$ minutes at the coffee shop. Similarly for the second case, total time $T_2 = 2$ hours 24 minutes, which is 2.4 hours (since 24 minutes is 0.4 hours). Again, that includes walking time and $t$ minutes.

Therefore, if I convert all times to hours, maybe I can set up equations.

First, convert $T_2$ to hours: 2 hours 24 minutes $= 2 + \frac{24}{60} = 2 + 0.4 = 2.4$ hours.

So, for first case:

Walking time at speed $s$: distance / speed $= \frac{9}{s}$ hours. Then adding $t$ minutes (which is $\frac{t}{60}$ hours) gives total time $T_1 = 4$ hours. Therefore:

$\frac{9}{s} + \frac{t}{60} = 4$.

Similarly for second case:

Walking time at speed $s + 2$ is $\frac{9}{s+2}$ hours, adding $\frac{t}{60}$ hours gives total time $T_2 = 2.4$ hours. Therefore:

$\frac{9}{s+2} + \frac{t}{60} = 2.4.$

So now, I have two equations with two variables: $s$ and $t$. Let me write them again:

Equation 1: $\frac{9}{s} + \frac{t}{60} = 4.$

Equation 2: $\frac{9}{s+2} + \frac{t}{60} = 2.4.$

So if I subtract Equation 2 from Equation 1, I can eliminate $\frac{t}{60}$:

$\left( \frac{9}{s} - \frac{9}{s+2} \right) + \left( \frac{t}{60} - \frac{t}{60} \right) = 4 - 2.4,$

which gives

$\frac{9}{s} - \frac{9}{s+2} = 1.6.$

Therefore:

Let me compute $\frac{9}{s} - \frac{9}{s+2} = 1.6.$

Compute left side:

$\frac{9(s+2)-9s}{s(s+2)} = \frac{9s+18-9s}{s(s+2)} = \frac{18}{s(s+2)}.$

Therefore:

$\frac{18}{s(s+2)} = 1.6.$

So:

$s(s + 2) = \frac{18}{1.6} = 11.25.$

So:

$s^2 + 2s - 11.25 = 0.$

Let me solve this quadratic equation for $s$.

Quadratic equation: $s^2 + 2s - 11.25 = 0.$

Using quadratic formula:

$s = \frac{-2 \pm \sqrt{(2)^2 + 4 \cdot 1 \cdot 11.25}}{2 \cdot 1} = \frac{-2 \pm \sqrt{4+45}}{2} = \frac{-2 \pm \sqrt{49}}{2} = \frac{-2 \pm 7}{2}.$

Since speed can't be negative, take positive solution:

$s = \frac{-2+7}{2} = \frac{5}{2} = 2.5$ km/h.

So $s$ is 2.5 km/h. Then, we can find $t$ from Equation 1 or 2. Let's use Equation 1:

Equation 1: $\frac{9}{s} + \frac{t}{60} = 4$.

Substituting $s = 2.5$:

$\frac{9}{2.5} + \frac{t}{60} = 4$.

Compute $9 \div 2.5$:

Well, $2.5 \times 3 = 7.5$, so $9 - 7.5 = 1.5$, so $3 + \frac{1.5}{2.5} = 3 + 0.6 = 3.6$? Wait:

Wait 9 divided by 2.5:

Multiply numerator and denominator by 2: $\frac{18}{5} = 3.6$. Yes, correct. Therefore $3.6 + \frac{t}{60} = 4 \Rightarrow \frac{t}{60} = 0.4 \Rightarrow t = 0.4 \times 60 = 24$ minutes. Therefore $t$ is 24 minutes. So that's consistent with both equations? Let me check with equation 2:

Equation 2: $\frac{9}{s+2} + \frac{t}{60} = \frac{9}{2.5+2} + \frac{24}{60} = \frac{9}{4.5} + 0.4 = 2 + 0.4 = 2.4$ hours. Which is correct. Perfect. Therefore our values are correct.

So now, we need to find the total time when she walks at $s + \frac{1}{2}$ km/h, which is $2.5 + 0.5 = 3$ km/h.

So walking speed is now 3 km/h. Therefore, walking time is $\frac{9}{3} = 3$ hours. Then adding $t$ minutes (which is 24 minutes) gives total time: 3 hours $+$ 24 minutes. Convert that into minutes? Or into hours?

But the problem says "Find the number of minutes the walk takes her, including the $t$ minutes spent in the coffee shop."

Therefore, convert 3 hours into minutes: $3 \times 60 = 180$ minutes. Then add 24 minutes, so total time is $180 + 24 = 204$ minutes. Therefore answer is 204 minutes.

Wait, that seems straightforward, but let me check once again.

Original equations:

$s$ was found as 2.5 km/h, $t$ was found as 24 minutes. So when she walks at $s + \frac{1}{2} = 3$ km/h, walking time is $\frac{9}{3} = 3$ hours $= 180$ minutes. Adding $t$ minutes (24) gives total of 204 minutes, yes. Therefore answer is $\boxed{204}$.

But let me check once again if there's something missing here. For example, maybe I misread the problem? Let me check again:

"When she walks at a constant speed of $s$ kilometers per hour, the walk takes her 4 hours, including $t$ minutes spent in the coffee shop."

So total time (walking $+$ coffee) is 4 hours. Which is why I set up equations accordingly. Then same for other case. So solving gives us $t = 24$ minutes. Therefore when she walks at $s + \frac{1}{2}$, time is walking time $+ t$ minutes. Which is 3 hours $+$ 24 minutes $= 204$ minutes. Seems correct.

Therefore answer is 204. Therefore boxed answer $\boxed{204}$ is correct.

**Final Answer**

204

---

**Response**:
To determine the number of minutes the entire walk (including the coffee shop visit time $t$) would last in the third case, we carefully examine the relationship among the distance, speed, time, and fixed waiting time provided across three different conditions for the number Aya walks in the morning.

Let us derive from this situation by analyzing the expressions:

—

Step 1: Represent Time and Known Quantities

Let:

$s = $ Walking speed in km/h

$t = $ Fixed time at coffee shop in minutes

From the walk distance being **always the same** and assuming:

Walk Length $= D = 9$ km

She walks the entire route (excluding coffee shop) at speed $s$, hence:

$T_{\text{walk}} = \frac{9}{s}$

So the actual walking time is included within a schedule time $T_{\text{schedule}}$, where:

$T_{\text{schedule}} = $ walking time in hours $+ \frac{t}{60} = \frac{D}{v} + \frac{t}{60}$

From the data:

1) Walking speed $s = s$: schedule time $T_1 = 4$ h

$\Rightarrow \frac{9}{s} + \frac{t}{60} = 4$

2) Speed becomes $s + 2$: schedule time becomes 2 h and 24 min $= 2 + \frac{24}{60} = 2.4$

So now:

$\Rightarrow \frac{9}{s+2} + \frac{t}{60} = 2.4$

—

Subtracting $(1)$ from the result of subtracted steps:

From:

$\left( \frac{9}{s} - \frac{9}{s+2} \right) = 1.6$

After calculation (algebra details omitted but consistent), solve the difference to eliminate $t$:

That expression gives:

$\frac{9}{s} - \frac{9}{s+2} = \frac{18}{s(s+2)} = 1.6 \Rightarrow s(s+2) = \frac{18}{1.6} = 11.25$

This implies the resulting expression for:

$s^2 + 2s = 11.25 \Rightarrow s^2 + 2s - 11.25 = 0$

Now solving using the quadratic formula:

$s = \frac{-2 \pm \sqrt{4+45}}{2} = \frac{-2 \pm \sqrt{49}}{2} \Rightarrow s = \frac{5}{2} = \boxed{2.5}$

Then:

$t = \left(4 - \frac{9}{2.5}\right) \cdot 60 = (4 - 3.6) \cdot 60 = 0.4 \cdot 60 = \boxed{24}$

—

Step 2: When Walking Speed is $s + \frac{1}{2} = 3$ km/h

Walking time becomes:

$\frac{9}{3} = 3$ h

Adding fixed coffee time:

Coffee $= t = 24$ minutes

Thus:

$3$ h $= 180$ min

$\Rightarrow$ Total time including coffee time is $180 + 24 = \boxed{204}$

—

**Final Result**

$\boxed{204}$

---

**Problem**:
Every morning Aya goes for a 9-kilometer-long walk and stops at a coffee shop afterwards. When she walks at a constant speed of $s$ kilometers per hour, the walk takes her $4$ hours, including $t$ minutes spent in the coffee shop. When she walks $s + 2$ kilometers per hour, the walk takes her $2$ hours and $24$ minutes, including $t$ minutes spent in the coffee shop. Suppose Aya walks at $s + \frac{1}{2}$ kilometers per hour. Find the number of minutes the walk takes her, including the $t$ minutes spent in the coffee shop.

Please reason step by step, and put your final answer within \boxed{}.

--------------------------------------------------------------------------------

**Reasoning**:
Okay, so Aya goes for a 9-kilometer walk every morning and stops at a coffee shop afterwards. The problem

gives me two scenarios where she walks at different speeds and tells me the total time taken, including the time spent at the coffee shop. I need to figure out how long the walk takes her when she walks at a speed of $s + \frac{1}{2}$ km/h.

First, let me parse the problem again to make sure I understand all the details.

In the first scenario, she walks at a constant speed of $s$ km/h, and the total time for the walk, including $t$ minutes at the coffee shop, is 4 hours. In the second scenario, she walks at $s + 2$ km/h, and the total time is 2 hours and 24 minutes, again including $t$ minutes at the coffee shop. Then, I need to find the total time when she walks at $s + \frac{1}{2}$ km/h, including the same $t$ minutes at the coffee shop.

So, the key here is that the total time includes both the time spent walking and the time spent at the coffee shop. Therefore, if I can figure out the time she spends walking in each scenario, I can subtract the coffee shop time $t$ from the total time to get the walking time, and then use that to solve for $s$ and $t$. Then, once I have $s$ and $t$, I can compute the total time for the third scenario.

Let me write down the given information mathematically.

First scenario:

- Speed $= s$ km/h

- Total time including coffee shop $= 4$ hours

- Therefore, walking time $= 4$ hours $-t$ minutes. But wait, the units here are inconsistent. The total time is given in hours, but the coffee shop time is in minutes. I need to convert them to the same unit.

Let me convert $t$ minutes to hours. Since there are 60 minutes in an hour, $t$ minutes is $\frac{t}{60}$ hours. Therefore, walking time in the first scenario is 4 hours $-\frac{t}{60}$ hours.

Similarly, the distance is 9 km. So, walking time is also equal to distance divided by speed. So:

Walking time $= \frac{9}{s}$ hours.

Therefore, we have the equation:

$\frac{9}{s} = 4 - \frac{t}{60}$.

Similarly, for the second scenario:

Speed $= s + 2$ km/h

Total time including coffee shop $= 2$ hours 24 minutes. Let me convert that to hours. 2 hours is 2 hours, 24 minutes is $\frac{24}{60}$ hours $= 0.4$ hours. So total time is 2.4 hours. Therefore, walking time is 2.4 hours $-\frac{t}{60}$ hours.

Again, walking time is also equal to distance divided by speed:

$\frac{9}{s+2} = 2.4 - \frac{t}{60}$.

So now we have two equations:

1) $\frac{9}{s} = 4 - \frac{t}{60}$

2) $\frac{9}{s+2} = 2.4 - \frac{t}{60}$

So, two equations with two variables: $s$ and $t$. We can solve for $s$ and $t$.

Let me write them again:

Equation 1: $\frac{9}{s} = 4 - \frac{t}{60}$

Equation 2: $\frac{9}{s+2} = 2.4 - \frac{t}{60}$

So, if I can express $t$ from one equation and substitute into the other, I can solve for $s$.

Let me rearrange equation 1 to solve for $t$.

From equation 1:

$\frac{t}{60} = 4 - \frac{9}{s}$

Multiply both sides by 60:

$t = 60\left(4 - \frac{9}{s}\right) = 240 - \frac{540}{s}$

Similarly, from equation 2:

$\frac{t}{60} = 2.4 - \frac{9}{s+2}$

Multiply both sides by 60:

$t = 60\left(2.4 - \frac{9}{s+2}\right) = 144 - \frac{540}{s+2}$

Therefore, since both expressions equal $t$, we can set them equal to each other:

$240 - \frac{540}{s} = 144 - \frac{540}{s+2}$

Let me write that equation:

$240 - \frac{540}{s} = 144 - \frac{540}{s+2}$

Let me subtract $144$ from both sides:

$240 - 144 - \frac{540}{s} = -\frac{540}{s+2}$

Which simplifies to:

$96 - \frac{540}{s} = -\frac{540}{s+2}$

Let me rearrange the equation:

$96 = \frac{540}{s} - \frac{540}{s+2}$

Factor out $540$:

$96 = 540\left(\frac{1}{s} - \frac{1}{s+2}\right)$

Compute the difference in the parentheses:

$$\frac{1}{s} - \frac{1}{s+2} = \frac{(s+2)-s}{s(s+2)} = \frac{2}{s(s+2)}$$

Therefore:

$$96 = 540 \cdot \frac{2}{s(s+2)}$$

So:

$$96 = \frac{540 \cdot 2}{s(s+2)}$$

Compute $540 \cdot 2 = 1080$

Therefore:

$$96 = \frac{1080}{s(s+2)}$$

Multiply both sides by $s(s+2)$:

$$96 \cdot s(s+2) = 1080$$

Divide both sides by 12 to simplify:

$$8 \cdot s(s+2) = 90$$

Wait, 96 divided by 12 is 8, 1080 divided by 12 is 90. Let me check:

Yes, $\frac{96}{12} = 8$, $\frac{1080}{12} = 90$. So:

$$8s(s+2) = 90$$

Divide both sides by 2:

$$4s(s+2) = 45$$

So:

$$4s^2 + 8s - 45 = 0$$

So, quadratic equation: $4s^2 + 8s - 45 = 0$

Let me solve this quadratic equation for $s$.

Quadratic formula: $s = \frac{-b \pm \sqrt{b^2 - 4ac}}{2a}$

Here, $a = 4$, $b = 8$, $c = -45$

Discriminant $D = b^2 - 4ac = 64 - 4 \cdot 4 \cdot (-45) = 64 + 720 = 784$

$\sqrt{784} = 28$

Therefore:

$$s = \frac{-8 \pm 28}{2 \cdot 4} = \frac{-8 \pm 28}{8}$$

So two solutions:

$s = \frac{-8+28}{8} = \frac{20}{8} = \frac{5}{2} = 2.5$ km/h

$s = \frac{-8-28}{8} = \frac{-36}{8} = -4.5$ km/h

Since speed cannot be negative, we discard the negative solution. Therefore, $s = 2.5$ km/h.

Now that we have $s$, we can find $t$ from one of the earlier equations. Let me use equation 1:

$t = 240 - \frac{540}{s}$

$s$ is 2.5, so:

$t = 240 - \frac{540}{2.5}$

Compute $\frac{540}{2.5}$:

Well, 540 divided by 2.5. Since 2.5 is $\frac{5}{2}$, dividing by $\frac{5}{2}$ is multiplying by $\frac{2}{5}$. So $540\left(\frac{2}{5}\right) = 540(0.4) = 216$.

Therefore, $t = 240 - 216 = 24$ minutes.

So $t$ is 24 minutes.

Let me check this with equation 2 to make sure.

From equation 2: $t = 144 - \frac{540}{s+2}$

$s + 2 = 2.5 + 2 = 4.5$ km/h

So $\frac{540}{4.5} = ?$

Well, $4.5 \cdot 100 = 450$, so $\frac{540}{4.5} = 120$. Because $4.5 \cdot 120 = 540$.

Therefore, $t = 144 - 120 = 24$ minutes. Perfect, same result. So $t = 24$ minutes.

So now, we know that $s = 2.5$ km/h and $t = 24$ minutes.

Now, the problem asks: Suppose Aya walks at $s + \frac{1}{2}$ km/h. Find the number of minutes the walk takes her, including the $t$ minutes spent in the coffee shop.

So, first, compute her speed: $s + \frac{1}{2} = 2.5 + 0.5 = 3.0$ km/h.

So, she walks at 3.0 km/h. Now, we need to find the total time, which is walking time $+t$ minutes.

First, compute walking time. Walking time $= \frac{\text{distance}}{\text{speed}} = \frac{9 \text{ km}}{3.0 \text{ km/h}} = 3$ hours.

Therefore, total time is 3 hours $+t$ minutes. But $t$ is 24 minutes, so total time is 3 hours $+24$ minutes.

But the question asks for the number of minutes the walk takes her, including the $t$ minutes. So we need to convert 3 hours into minutes and add 24 minutes.

3 hours $= 180$ minutes. So total time is $180 + 24 = 204$ minutes.

Wait, but let me check again. Wait, the total time is walking time (in hours) plus $t$ minutes. But when we calculated walking time, we had to convert it to hours? Wait, no. Wait, let me think.

Wait, in the previous scenarios, the total time was given as hours, but the coffee shop time was in minutes. So when we calculated, we converted $t$ minutes to hours. But here, if we compute walking time as $\frac{9 \text{ km}}{3 \text{ km/h}} = 3$ hours, and then add $t$ minutes (24 minutes). So total time is 3 hours $+24$ minutes. To express this in minutes, we have $3 \cdot 60 + 24 = 180 + 24 = 204$ minutes. Therefore, the answer is 204 minutes.

But let me check if there is another way to think about this. Alternatively, maybe I need to check if my calculation of walking time is correct. Let me verify.

Alternatively, maybe I should compute the total time in hours, then convert to minutes. Let me see:

Walking time is 3 hours. Then total time is 3 hours $+24$ minutes. To convert total time to minutes: 3 hours is 180 minutes, plus 24 minutes is 204 minutes. So yes, same result.

But let me check if my value of $s$ is correct. Because $s$ was 2.5 km/h. Let me check the first scenario.

First scenario: walking speed 2.5 km/h. Time to walk 9 km is $\frac{9}{2.5} = 3.6$ hours. Which is 3 hours and 36 minutes. Then total time is 4 hours, which includes $t$ minutes. Therefore, 3.6 hours $+t$ minutes $= 4$ hours. Therefore, $t$ minutes is 4 hours $-3.6$ hours $= 0.4$ hours $= 24$ minutes. Which matches. Similarly, second scenario: speed is $s + 2 = 4.5$ km/h. Time to walk is $\frac{9}{4.5} = 2$ hours. Total time is 2 hours and 24 minutes, which is 2.4 hours. Therefore, 2 hours $+t$ minutes $= 2.4$ hours. Therefore, $t$ minutes is 0.4 hours $= 24$ minutes. So that checks out. Therefore, $t$ is indeed 24 minutes.

Therefore, when she walks at 3 km/h, walking time is $\frac{9}{3} = 3$ hours. Total time is 3 hours $+24$ minutes $= 3$ hours 24 minutes $= 204$ minutes. Therefore, the answer is 204 minutes.

But let me check again if I didn't make any miscalculations. Let me recheck the quadratic equation part.

Earlier, we had:

From the two equations, we arrived at $4s^2 + 8s - 45 = 0$. Let me check that again.

Original equations:

After simplifying, we had:

$96 = \frac{1080}{s(s+2)}$

Therefore:

$s(s + 2) = \frac{1080}{96}$

Compute $\frac{1080}{96}$. Let me compute that. Divide numerator and denominator by 12: $\frac{90}{8} = 11.25$. Wait, but earlier I had $96 \cdot s(s + 2) = 1080$, so $s(s + 2) = \frac{1080}{96} = 11.25$. Then, expanding:

$s^2 + 2s = 11.25$

Therefore, $s^2 + 2s - 11.25 = 0$

But earlier, I had $4s^2 + 8s - 45 = 0$. Let me check if multiplying by 4 gives that.

Multiply $s^2 + 2s - 11.25 = 0$ by 4: $4s^2 + 8s - 45 = 0$. Yes, that's correct. So that's correct. There-fore, solving that quadratic gives $s = \frac{-8\pm\sqrt{64+180}}{8}$? Wait, no. Wait, discriminant $D = b^2 - 4ac$. For equation $4s^2 + 8s - 45 = 0$, $D = 64 - 4 \cdot 4 \cdot (-45) = 64 + 720 = 784$. Which is $\sqrt{784} = 28$. Then $s = \frac{-8\pm28}{8}$. Which gives $\frac{20}{8} = 2.5$ or $\frac{-36}{8} = -4.5$. So that is correct.

So, $s = 2.5$ is correct. Therefore, calculations are correct. Then, the rest follows.

Therefore, the answer is 204 minutes.

But let me check once again the final step. When she walks at $s + \frac{1}{2} = 2.5 + 0.5 = 3.0$ km/h. Walking time is $\frac{9}{3} = 3$ hours. Total time is walking time + coffee shop time. Coffee shop time is 24 minutes. So total time is 3 hours +24 minutes.

But the problem says "the number of minutes the walk takes her, including the $t$ minutes spent in the cof-fee shop." So, converting 3 hours to minutes is 180 minutes, plus 24 minutes is 204 minutes. So 204 minutes. Therefore, the answer is 204.

Alternatively, maybe there is a trick here? Let me check again.

Wait, is the coffee shop time $t$ minutes, which is 24 minutes. So when she walks at 3 km/h, the time spent walking is 3 hours, so total time is 3 hours +24 minutes. But maybe the question is trying to trick me into thinking that the coffee shop time is included in the total time, but in the previous scenarios, the total time was given as including $t$ minutes. So in the first scenario, the total time was 4 hours, which included $t$ minutes. Therefore, when they ask for the total time including $t$ minutes, it's just walking time $+t$ minutes. So yes, 3 hours +24 minutes. Which is 204 minutes.

Alternatively, maybe I need to check if there is a different interpretation. For example, is the coffee shop time part of the total time? The problem says: "the walk takes her 4 hours, including $t$ minutes spent in the coffee shop." So the total time for the walk (which includes walking and coffee shop) is 4 hours. Therefore, when they say "the number of minutes the walk takes her, including the $t$ minutes spent in the coffee shop," it's just the same as the total time. So in the first case, total time is 4 hours, which is 240 minutes. But in the second case, total time is 2 hours and 24 minutes, which is 144 minutes. Wait, but when we calculated $t$, we found that $t$ is 24 minutes. So in the first scenario, total time is 4 hours = 240 minutes, which is equal to walking time plus $t$ minutes. Walking time was 3.6 hours = 216 minutes. $216 + 24 = 240$. Correct. Similarly, second scenario: walking time is 2 hours = 120 minutes, plus 24 minutes is 144 minutes, which is 2 hours 24 minutes. Correct. Therefore, in the third scenario, walking time is 3 hours = 180 minutes, plus 24 minutes is 204 minutes. Therefore, the answer is 204 minutes.

Therefore, I think that is correct. So I think the answer is 204 minutes.

But just to make sure, let me check if there's another way someone might approach this problem and per-haps make a mistake. For example, someone might think that the coffee shop time is part of the walking time? But the problem says "the walk takes her 4 hours, including $t$ minutes spent in the coffee shop." So the walk is the 9 km walk, and then she stops at the coffee shop afterwards. Therefore, the total time is the time spent walking plus the time spent in the coffee shop. Therefore, the coffee shop time is separate. So when they say "the walk takes her 4 hours, including $t$ minutes," it's the total time from start to finish, including the coffee shop. Therefore, the answer is 204 minutes.

Alternatively, maybe I need to check if the problem is in another way. For example, maybe the coffee shop time is part of the walk? But the problem says she stops at the coffee shop afterwards. So the walk is the 9 km walk, and then she stops at the coffee shop. So the total time is walking time plus coffee shop time. So yes, the way I did it is correct.

Therefore, I think the answer is 204 minutes.

**Final Answer**

204

---------------------------------------------------------------------------------------

**Response**:
To determine how long Aya's walk takes when she walks at a speed of $s + \frac{1}{2}$ kilometers per hour, we begin by analyzing the given information and setting up equations based on the relationship between distance, speed, and time.

—

Step 1: Understand the Problem

Aya walks 9 kilometers every morning. The total time for the walk, including $t$ minutes spent at the coffee shop, is given for two different walking speeds:

First scenario:

Speed $= s$ km/h

Total time $= 4$ hours $= 240$ minutes

Walking time $= 240$ minutes $-t$ minutes

Second scenario:

Speed $= s + 2$ km/h

Total time $= 2$ hours 24 minutes $= 144$ minutes

Walking time $= 144$ minutes $-t$ minutes

—

Step 2: Express Walking Time in Terms of Speed and Distance

The walking time can also be calculated using the formula:

Walking time (in hours) $= \frac{9}{\text{speed}}$

Convert walking time to minutes by multiplying by 60.

So, the equations become:

$\frac{9}{s} \times 60 = 240 - t$

$\frac{9}{s+2} \times 60 = 144 - t$

—

Step 3: Solve the System of Equations

From equation (1):

$\frac{540}{s} = 240 - t \Rightarrow t = 240 - \frac{540}{s}$

From equation (2):

$\frac{540}{s+2} = 144 - t \Rightarrow t = 144 - \frac{540}{s+2}$

Set the two expressions for $t$ equal:

$240 - \frac{540}{s} = 144 - \frac{540}{s+2}$

Simplify:

$96 = \frac{540}{s} - \frac{540}{s+2}$

$96 = 540\left(\frac{1}{s} - \frac{1}{s+2}\right)$

$96 = 540 \cdot \frac{2}{s(s+2)} \Rightarrow 96 = \frac{1080}{s(s+2)}$

$s(s+2) = \frac{1080}{96} = 11.25$

Now solve the quadratic:

$s^2 + 2s - 11.25 = 0$

Use the quadratic formula:

$s = \frac{-2 \pm \sqrt{4+45}}{2} = \frac{-2 \pm \sqrt{49}}{2} = \frac{-2 \pm 7}{2}$

$s = \frac{5}{2} = 2.5$ (only valid solution, as speed can't be negative)

—

Step 4: Find $t$

$t = 240 - \frac{540}{2.5} = 240 - 216 = 24$ minutes

—

Step 5: Calculate Total Time at $s + \frac{1}{2} = 3$ km/h

Walking time $= \frac{9}{3} = 3$ hours $= 180$ minutes

Coffee shop time $= 24$ minutes

Total time $= 180 + 24 = 204$ minutes

—

**Final Answer**

$\boxed{204}$

