# OpenReview forum: "Characterizing, Evaluating, and Optimizing Complex Reasoning"
_ICML.cc/2026/Conference — ICML 2026 spotlight_

### Official Review · Reviewer_CLDy · 2026-02-28

**Soundness:** 3
**Presentation:** 4
**Significance:** 3
**Originality:** 4
**Overall Recommendation:** 5
**Confidence:** 4

**Summary:**

This paper addresses the challenges of defining and evaluating high-quality reasoning, as well as how to utilize these evaluation signals for reasoning optimization. It introduces the ME2 principle to evaluate reasoning quality, which simultaneously considers global structural organization and local step properties. Based on the ME2 principle, the paper models reasoning traces as Directed Acyclic Graphs (DAGs) and develops a DAG-based pairwise evaluation method to capture complex reasoning structures. Subsequently, the authors construct the TRM-Preference dataset and train a Thinking Reward Model (TRM) to evaluate reasoning quality. Experimental results demonstrate that thinking rewards serve as an effective optimization signal, and selecting better reasoning trajectories leads to superior final outcomes.

**Compliance With Llm Reviewing Policy:**

Affirmed.

**Final Justification:**

I recommend Accepting the paper, as the authors' rebuttal effectively resolved the primary concerns regarding scalability and evaluation integrity. New results on large-scale models (GPT-OSS-120B and Qwen3-235B) confirm that the TRM remains effective with state-of-the-art generators, while technical clarifications on the DAG construction process ensure that the evaluation remains grounded in the original reasoning without "fixing" logical errors. Overall, the ME² principle and DAG-based modeling provide a sound and significant contribution to characterizing and optimizing complex reasoning.

**Key Questions For Authors:**

1. Why choose to use an LLM to select parent nodes instead of following a sequential order? Could this affect the evaluation of reasoning quality? For instance, if a policy model generates a disjointed or incoherent reasoning trace, might the LLM fix the logic while constructing the DAG, resulting in an artificially high-quality representation?
2. Could you provide a experimental comparison with other existing methods for characterizing high-quality reasoning?
3. Can the generalization of the pairwise preference evaluation be demonstrated on other datasets?
4. The models used for test-time scaling are small; could you provide results or insights on performance when scaled to much larger models?

**Limitations:**

For example, the generalization and effectiveness of the framework on significantly larger scale models.

**Strengths And Weaknesses:**

## Soundness
### Strength
The technical roadmap is sound, and the claims are well-supported. The progression from proposing the ME2 principle and constructing reasoning traces as DAGs, to using ME2 for pairwise comparisons to derive thinking rewards, is logical. The TRM trained with these thinking rewards proves effective in both test-time scaling and Reinforcement Learning (RL) training processes.

### Weakness
The paper lacks a experimental comparison with other methods for characterizing high-quality reasoning, and there is no ablation study comparing DAGs with other reasoning structure modeling methods.

---

## Presentation
### Strength
The paper is well-written with a clear structure. The figures and tables are easy to understand.

### Weakness
While the paper introduces many novel components, the central focus becomes somewhat diluted as a result.

---

## Significance
### Strength
The paper successfully addresses how to evaluate reasoning quality and apply the resulting signals to RL training. Integrating both global structure and local properties to assess reasoning quality, while decoupling it from answer correctness, sounds both novel and well-justified.

### Weakness
1) The experimental scope is somewhat limited; pairwise preference evaluation is only shown on the validation set, with no evidence of generalization across other datasets.
2) The model scale used for test-time scaling is relatively small. It remains unclear whether the approach remains equally effective on much larger models (e.g., GPT-OSS-120B).

---

## Originality
### Strength
The paper proposes a new principle for evaluating reasoning and a novel dataset for training reward models. The distinctions from related literature are clearly articulated.

---

> ### Author Rebuttal · Authors · 2026-03-30
>
> Thanks for your valuable comments. We will incorporate them to improve the paper.
>
> > Q1. Why choose to use an LLM to select parent nodes instead of following a sequential order? Could this affect the evaluation of reasoning quality? For instance, if a policy model generates a disjointed or incoherent reasoning trace, might the LLM fix the logic while constructing the DAG, resulting in an artificially high-quality representation?
> >
> 1. **We do not assume a purely sequential chain because complex reasoning is often non-linear (Sec. 1, Sec. 3.1).** A fixed adjacent-step order would miss branching/merging and can make backtracking or branch switching look artificially coherent. The DAG is introduced to capture such dependencies and enable a faithful evaluation of reasoning quality.
> 2. **Our pipeline does not “fix the logic”.** The LLM only selects parent nodes among existing steps. It does not rewrite the trace or add missing logic. After compression, each super-node still maps to a semantically continuous span of the original trace, and micro abstraction is performed on the dominant path serialized from the original nodes. Thus, evaluation remains grounded in the original reasoning rather than repaired by the evaluator.
> 3. **We also reduce construction noise via robust evaluation** **(Sec. 4.3)**. We use original/reversed orderings with repeated judgments, and retain only consistent non-tied labels.
>
> > Q2. Could you provide an experimental comparison with other existing methods for characterizing high-quality reasoning?
> >
> 1. **Prior work is heterogeneous and not directly comparable.** As discussed in Sec. 1/2 and App. C.1, existing methods usually capture only partial aspects of reasoning quality (e.g., step correctness or verbosity), while structure-based methods often rely on predefined markers or complex, non-automated pipelines, limiting scalability and reproducibility. In contrast, our ME² principle provides a unified characterization of arbitrary free-form reasoning.
> 2. **For the closest operational baseline, we already provide direct comparison with ReasonFlux-PRM [1].** As shown in Tab. 1, Sec. 5, App. C.1, TRM achieves higher accuracy and consistent gains in TTS/RL and reasoning-quality analysis.
>
> [1] ReasonFlux-PRM: Trajectory-Aware PRMs for Long Chain-of-Thought Reasoning in LLMs
>
> > Q3. Can the generalization of the pairwise preference evaluation be demonstrated on other datasets?
> >
>
> We believe the main challenge here is the limited availability of other directly comparable pairwise preference datasets.
>
> 1. **Existing preference datasets are not directly comparable to our setting.** We focus on the reasoning structure that models exhibit on complex reasoning tasks in classic domains such as math and STEM, with labels targeting reasoning quality under ME² rather than generic response preference. Existing preference datasets (e.g., [1,2]) are instead mainly RLHF-oriented (e.g. helpful/honest/harmless).
> 2. **Our preference data is fairly broad in scope.** It is built from 64K WebInstruct-verified prompts spanning diverse domains (Sec. 5.1), with candidate traces generated by multiple reasoning model families (Qwen3, DeepSeek-Distill, GPT-OSS), so it is not tied to a narrow data/model distribution.
>
> [1] Training a Helpful and Harmless Assistant with Reinforcement Learning from Human Feedback
>
> [2] UltraFeedback: Boosting Language Models with High-quality Feedback
>
> > Q4. The models used for test-time scaling are small. Could you provide results or insights on performance when scaled to much larger models (e.g., GPT-OSS-120B)?
> >
>
> Thanks for raising this point. We agree that the main text covers only relatively small generators. We therefore add AIME25 results on much larger models (GPT-OSS-120B and Qwen3-235B-A22B) under the same protocol as Sec. 5.3.
>
> GPT-OSS-120B:
>
> |N|1|2|4|8|16|
> |-|-|-|-|-|-|
> |Qwen-PRM|80.0|80.7|82.7|83.3|83.3|
> |ReasonFlux-PRM|80.0|82.0|82.0|84.0|84.7|
> |TRM (ours)|80.0|81.3|83.3|84.7|86.7|
>
> Qwen3-235B-A22B:
>
> |N|1|2|4|8|16|
> |-|-|-|-|-|-|
> |Qwen-PRM|71.3|72.0|74.0|75.3|76.7|
> |ReasonFlux-PRM|71.3|72.7|74.0|76.0|76.0|
> |TRM (ours)|71.3|72.7|74.7|76.0|78.7|
>
> TRM outperforms both baselines, consistent with Fig. 4. The gains are smaller than on smaller generators, as expected given the stronger base model performance. This suggests **TRM is not limited to small-model TTS and remains effective at larger scale**. We will add these results in the revision.
>
> > Q5. While the paper introduces many novel components, the central focus becomes somewhat diluted as a result.
> >
>
> These components all serve a single goal reflected in our title, `Characterizing, Evaluating, and Optimizing Complex Reasoning`. ME² characterizes reasoning quality, DAG-based evaluation evaluates it, and TRM/TTS/RL optimize it. We will make this central thread more explicit in revision.

---

> > ### Author Rebuttal · Reviewer_CLDy · 2026-04-01
> >
> > I thank the authors for their comprehensive and convincing rebuttal. In particular, the inclusion of experimental results on large-scale models (GPT-OSS-120B and Qwen3-235B) directly addresses my concerns regarding scalability and confirms that the proposed TRM remains effective even with stronger base generators.
> >
> > Furthermore, the authors' clarification on the DAG construction process effectively alleviates my worry about potential "logic-fixing" by the evaluator LLM. The explanation that the LLM only selects parent nodes without rewriting the trace ensures the evaluation remains grounded in the original reasoning.
> >
> > While the comparison with non-DAG reasoning structures is still primarily supported by theoretical arguments, the overall empirical evidence and the performance gains over ReasonFlux-PRM are sufficient to support the paper's core claims. I maintain my positive assessment and recommend Accept.

---

> > > ### Author Response · Authors · 2026-04-04
> > >
> > > We appreciate your follow-up and are glad our clarifications and additional evidence helped address your concerns.

---

### Official Review · Reviewer_oPQG · 2026-03-05

**Soundness:** 3
**Presentation:** 4
**Significance:** 3
**Originality:** 3
**Overall Recommendation:** 5
**Confidence:** 4

**Summary:**

This paper introduces a novel method to evaluate and conpare the quality of reasoning traces. The work proposes $ME^2$ principle to characterize reasoning quality and designed a DAG-based pairwise evaluation method. Based on the method, a dataset called TRM-Preference is constructed to train a Thinking Reward Model, which provides guidance for test-time scaling and RL training.

**Compliance With Llm Reviewing Policy:**

Affirmed.

**Final Justification:**

The paper is technically solid, well presented and the author's rebuttal addresses my  concern, so I will maintain my recommendation of acceptance.

**Key Questions For Authors:**

1. During construction of the dataset, do you observe position bias using your comparison method?
2. The dataset only contains traces with final answers that are verified as correct. I'm curious about if the wrong samples can also be utilized, as intuitively it may encode information about model's failure mode.

**Limitations:**

yes

**Strengths And Weaknesses:**

## Strengths:
1. The presentation of the paper is good: The paper structure is well organized and very easy to follow.
2. The proposed reasoning quality evaluation method is principled and makes sense.
3. The empirical results are solid, supporting the claim of paper and showing the effectiveness of proposed evaluation method in improving test-time scaling and reasoning quality.

## Weaknesses:
1. The proposed comparison method seems expensive, since multiple LLM calls are needed in DAG construction and comparison. A detailed analysis of average cost for each pair would be beneficial, for example, number of API calles and cost in $ for each pair, or computation time if running locally.
2. The average of quality constructed DAG is less clear from several case studies.

---

> ### Author Rebuttal · Authors · 2026-03-30
>
> Thank you for your helpful feedback. We will revise the paper accordingly.
>
> > Q1. The proposed comparison method seems expensive, since multiple LLM calls are needed in DAG construction and comparison. A detailed analysis of average cost for each pair would be beneficial, for example, number of API calls and cost in ＄ for each pair, or computation time if running locally.
> >
>
> We agree that the proposed comparison pipeline is non-trivial in cost, and a detailed breakdown would be helpful.
>
> 1. **App. E explicitly states that our simplified attachment pool reduces the candidate set to O(d+b), substantially reducing DAG-construction cost.**
> 2. **The cost is competitive relative to prior work: ∼＄5k total, i.e., ∼＄0.05/pair over 103k retained pairs.** We build DAGs for ∼180k traces, ∼30 LLM calls/trace and ∼1k tokens/call on average (∼5.4B tokens). Pairwise comparison adds ∼130k comparisons: 4 dims × 4 orders × ∼500 tokens, plus aggregation at ∼5k tokens with the same 4 orders, totaling ∼3.6B tokens. Together with all other steps, the total remains ＜12B DeepSeek-V3.2 tokens. Based on our estimates from reported settings, Qwen-PRM [1] uses ∼30B tokens, while ReasonFlux-PRM [2] would require ＞3B GPT-4o tokens (∼＄30k).
> 3. **This cost is incurred once in a one-time offline construction pipeline.** The resulting TRM can then be reused across subsequent RL and TTS, so the cost is amortized over multiple downstream uses.
>
> We will add a more detailed cost breakdown in the revision to make things clearer.
>
> [1] The Lessons of Developing Process Reward Models in Mathematical Reasoning
>
> [2] ReasonFlux-PRM: Trajectory-Aware PRMs for Long Chain-of-Thought Reasoning in LLMs
>
> > Q2. The average of quality constructed DAG is less clear from several case studies.
> >
> 1. **App. K provides qualitative evidence that the DAGs are meaningful.** The case studies show that the inferred DAGs largely follow the generator’s reasoning flow, including progression, branching, and merging.
> 2. **This is also supported at scale.** The same pipeline yields consistent gains in TTS/RL (Sec. 5) and reasoning-quality analyses (Sec. 6), suggesting the extracted DAGs are useful beyond selected examples.
> 3. **We additionally run a reliability evaluation.** On 400 sampled DAGs, we ask both human experts and Gemini 3.1 Pro (thinking mode, IMO gold-medal level) to judge whether the DAG reflects the underlying structure. The positive rates are high: **95.3%** (Gemini) and **96.0%** (human), supporting that the constructed DAGs are reliable on average.
>
> Thus, the average quality is better supported by aggregate evidence than by a few examples alone.
>
> > Q3. During construction of the dataset, do you observe position bias using your comparison method?
> >
>
> We acknowledge that raw pairwise comparison may exhibit position bias. However, **Sec. 4.3 & App. J explicitly addresses this** via a robust protocol: each pair is evaluated under both original and reversed orderings, repeated multiple times, and **only consistent, non-tied labels are retained**. Thus, position-sensitive pairs are filtered out, and the constructed dataset is largely free of position-bias artifacts.
>
> > Q4. The dataset only contains traces with final answers that are verified as correct. I'm curious about if the wrong samples can also be utilized, as intuitively it may encode information about model's failure mode.
> >
> 1. As clarified in App. C.1/App. F, we intentionally keep only verifier-correct traces to decouple reasoning quality from answer correctness. **Mixing in incorrect traces would make TRM behave more like an answer judge, partly duplicating answer-based supervision, rather than cleanly modeling reasoning quality.**
> 2. **This orthogonality is central to our design.** In RLVR, correctness is already enforced by the verifier, while TRM shapes the quality of reasoning among correct solutions. Injecting correctness signals into TRM would weaken this separation.
> 3. That said, we agree that incorrect traces may still be useful for failure-mode analysis or auxiliary objectives. We view this as complementary future work.

---

> > ### Author Rebuttal · Reviewer_oPQG · 2026-04-02
> >
> > Thank you for the rebuttal! My concerns are addressed and I will maintain my recommendation of acceptance.

---

> > > ### Author Response · Authors · 2026-04-04
> > >
> > > We appreciate your follow-up and are glad our clarifications and additional evidence helped address your concerns.

---

### Official Review · Reviewer_XGKw · 2026-03-08

**Soundness:** 3
**Presentation:** 3
**Significance:** 2
**Originality:** 2
**Overall Recommendation:** 4
**Confidence:** 4

**Summary:**

The authors propose the $ME^2$ principle (Macro/Micro x Efficiency/Effectiveness) and a Directed Acyclic Graph (DAG) abstraction to characterize and optimize the reasoning traces of Large Reasoning Models. Based on this framework, they construct the TRM-Preference dataset using an LLM-as-a-judge and train a Thinking Reward Model (TRM). This TRM is subsequently integrated into reinforcement learning via GRPO to provide dense, process-oriented supervision, effectively decoupling reasoning quality from the correctness of the final outcome.

**Compliance With Llm Reviewing Policy:**

Affirmed.

**Final Justification:**

After reading the rebuttal, I am more positive on this paper than I was initially. I still think the overall pipeline is fairly heavy, but the authors did a good job answering the concerns that mattered most to me. In particular, the added complexity discussion and cost estimates make the scalability story much more credible, and the new agreement results with human experts and strong LLMs go a long way toward addressing my concern that the DAGs and preference data might mostly reflect the biases of a single judge model.

I also found the clarification about automatic prefix mining helpful, because it makes the method look less brittle than I had assumed from the original submission. More broadly, I still like the framing of ME^2 and the DAG view of reasoning traces, and the paper presents a fairly complete story from trace modeling to preference construction to reward learning and downstream use. So while I still see this as an engineering-heavy paper, the rebuttal changed my assessment in a meaningful way and made me notably more supportive of the final recommendation.

**Key Questions For Authors:**

1. The DAG construction requires invoking the evaluator model for each individual step to determine its parent nodes, resulting in an $O(N)$ inference cost per trace. While this graph extraction is performed offline to train the TRM, this massive computational overhead makes scaling the preference dataset generation (e.g., scaling from 103K to millions of pairs for robust data synthesis) practically prohibitive. Given this bottleneck, how do you justify the scalability of this data pipeline? Furthermore, the prompt explicitly instructs the evaluator that "Merging is rare" and to "prefer continue." Doesn't this artificially force the model to output linear structures, thereby invalidating the claim that the DAG captures genuine, emergent reasoning topologies?

2. As detailed in Appendix D, the step partitioning heavily relies on hard-coded lexical prefixes specific to certain models (e.g., "therefore" for DeepSeek, "need" for GPT). Have you evaluated how this rule-based heuristic generalizes to other base models that exhibit entirely different vocabulary distributions and linguistic habits?

3. The entire 103K preference dataset was generated by DeepSeek-V3.2 without any human-in-the-loop validation. How do you ensure that the TRM is learning an objective standard for high-quality reasoning, rather than merely overfitting to the stylistic and formatting biases of this specific judge model?

**Limitations:**

The authors have not adequately addressed two major limitations: the questionable authenticity of the extracted DAG structures and the extremely poor return on investment (ROI) of their overall pipeline.

Standard LLMs inherently generate reasoning traces through a linear, auto-regressive accumulation of tokens. Building a DAG after the fact with a separate judge model does not reveal a genuine cognitive topology. This fundamental issue is only worsened by the fact that the judge is heavily constrained by subjective prompt instructions. Instead of capturing an emergent reasoning structure, the pipeline essentially forces a naturally sequential generative process into an artificial maze. The resulting graph is more of a formatting artifact produced by the evaluator rather than an accurate representation of the generator's actual thought process.

Additionally, the paper glosses over the severe imbalance between its immense methodological complexity and the marginal empirical gains it achieves. The proposed framework requires heavy and unscalable engineering tricks. These include rule-based step partitioning, candidate pool truncation, and $O(N)$ sequential inference steps per trace. Yet, this massive system complexity yields a surprisingly low ROI. For example, applying the TRM via GRPO to Qwen2.5-Math-7B only provides a 2.0% absolute improvement over a naive rule-based verifier. The authors need to critically acknowledge this unfavorable cost-benefit ratio and discuss whether such a convoluted and computationally expensive approach is practically justified for large-scale RL or real-world deployment.

**Strengths And Weaknesses:**

**Strengths:**
1. The formulation of the $ME^2$ principle is highly intuitive and provides a clean, comprehensive taxonomy for diagnosing reasoning trace quality.
2. Transitioning from traditional linear or tree-based trace analysis to a DAG-based topology is a meaningful structural upgrade, as it naturally accommodates "merging" behaviors common in multi-step mathematical derivations.
3. The paper presents a remarkably complete engineering pipeline: conceptual definition, graph extraction, dataset construction, reward modeling, and RL fine-tuning, which demonstrates a strong commitment to establishing a full-stack optimization framework.

**Weaknesses:**
1. The DAG construction process relies on computationally expensive, sequential LLM inference steps and heavily constrained heuristics, which undermines its scalability and structural authenticity.
2. Given the immense computational complexity of the pipeline, the ultimate downstream performance improvements are surprisingly marginal.

---

> ### Author Rebuttal · Authors · 2026-03-30
>
> Thanks for your time and suggestions. We will incorporate them to improve our work.
>
> > Q1. The DAG construction relies on computationally expensive, sequential LLM inference steps (O(N) cost). How do you justify the scalability of this pipeline? (`Weakness 1` & `Key Question 1`)
> The downstream performance improvements are marginal. (`Weakness 2` & `Limitation 2`)
> >
>
> We believe there is a misunderstanding here, and the ROI of our pipeline is favorable:
>
> 1. App. E explicitly states that our simplified attachment pool **reduces the candidate set from O(n) to O(d+b), not O(n)**, substantially lowering DAG-construction cost.
> 2. **The cost is competitive with prior work.** For ∼180k traces, DAG construction uses ∼30 calls/trace at ∼1k tokens/call, i.e. ∼5.4B tokens, and ＜12B DeepSeek-V3.2 tokens overall (∼$5k). Based on our estimates from reported settings, Qwen-PRM [1] uses ∼30B tokens, while ReasonFlux-PRM [2] would require ＞3B GPT-4o tokens (∼＄30k). We will clarify this in revision.
> 3. **This is a one-time offline cost**. The resulting TRM is then reused across RL and TTS, so it is amortized over downstream uses.
> 4. The return is not marginal. In TTS, TRM yields clear gains (e.g., +11% on GPT-OSS-20B/AIME24 vs. +9% for Qwen-PRM). In RL, it outperforms both the verifier and ReasonFlux by **+2–4% across models/benchmarks, which is practically meaningful given RL gains of only ∼5–12% in this setting.**
>
> [1] The Lessons of Developing Process Reward Models in Mathematical Reasoning
>
> [2] ReasonFlux-PRM: Trajectory-Aware PRMs for Long Chain-of-Thought Reasoning in LLMs
>
> > Q2. The prompt instructs the evaluator that "Merging is rare" and to "prefer continue." Doesn’t this bias the evaluator toward linear structures, undermining the claim that the DAG captures genuine, emergent reasoning topology? (`Key Question 1`)
> Is the DAG merely an evaluator-induced formatting artifact, rather than an accurate representation of the generator’s actual reasoning process? (`Limitation 1`)
> >
> 1. **There is a misreading of “Merging is rare” & “prefer continue”**. In App. J.1 Fig. 17, the instruction is `"Merging is rare and should only be used when there is clear evidence of combining distinct prior lines of reasoning"` & `"While in doubt, prefer continue"` , a much narrower, precision-oriented prompt.
> 2. **This is meant to avoid false-positive merges, not to artificially impose linear structures.** Additionally, on 100 raw traces sampled before DAG construction, we find ∼1.4 potential merges per trace, vs. ∼7 branchings and ∼24 continue edges, suggesting merges are intrinsically rare.
> 3. **DAGs are empirically meaningful.** App. K case studies show that they align with the reasoning flow, and the gains in Fig. 5 further suggest they are useful rather than superficial formatting artifacts.
> 4. **We additionally run a reliability evaluation.** On 400 sampled DAGs, Gemini 3.1 Pro (IMO gold-medal level) and human experts judge whether the DAG reflects the underlying structure. The positive rates are high: **95.3%** (Gemini) and **96.0%** (human).
>
> > Q3. Appendix D relies on hard-coded lexical prefixes to certain models. How does this heuristic generalize to other models with different vocabulary and linguistic styles? (`Key Question 2`)
> >
> 1. **Our partitioner is not a brittle hard-coded parser for a single model.** We first split by `\n\n`, then automatically refine boundaries using high-frequency prefixes mined from that model’s traces, capturing both shared cues and family-specific styles (App. D).
> 2. **The procedure is automatic and transferable.** For a new model, we simply re-estimate prefix frequencies and select frequent prefixes, allowing the procedure to adapt to different vocabulary distributions automatically.
> 3. **We already apply it across three distinct families (Qwen3, DeepSeek-Distill, GPT-OSS)**, whose prefix distributions differ substantially (App. D), suggesting it is not tied to a single vocabulary style.
>
> > Q4. Since the 103K preference data is labeled by DeepSeek-V3.2 without human validation, how do you know TRM learns reasoning quality rather than overfitting to this judge’s stylistic biases? (`Key Question 3`)
> >
>
> Thanks for your suggestions. We agree that a reliability validation is necessary. **Therefore, we sample 400 pairs and ask human experts and LLMs (Gemini 3.1 Pro, GPT-5.4, Claude Opus 4.6) to independently label each pair with win/tie/loss**, using the same robust protocol as Sec. 4.3 and reporting non-tie accuracy as in Sec. 5.2.
>
> |Evaluator|Accuracy|
> |-|-|
> |Gemini 3.1 pro|86.3%|
> |GPT-5.4|93.0%|
> |Claude Opus 4.6|89.8%|
> |Human|90.0%|
>
> These high agreement rates suggest the preference data is not merely overfitting stylistic biases of DeepSeek-V3.2. Instead, the learned preferences largely align with ME². The consistent TTS/RL gains and reasoning-quality analysis (Sec. 5/6) further support that TRM learns a transferable quality signal. We will add this in revision.

---

> > ### Author Rebuttal · Reviewer_XGKw · 2026-04-01
> >
> > I have read the authors' rebuttal and appreciate the clarifications provided. The additional details regarding the complexity reduction and the corresponding cost estimates adequately address my questions about the pipeline's scalability. Furthermore, the newly provided agreement data helps mitigate my initial concerns regarding potential evaluator bias and the authenticity of the extracted reasoning structures. The explanation of the automated prefix mining process also clarifies the method's generalizability across different model families. While the overall engineering framework remains somewhat heavy, the authors have reasonably justified the return on investment given the reported performance gains. Therefore, my primary concerns have been resolved, and I will adjust my evaluation accordingly.

---

> > > ### Author Response · Authors · 2026-04-04
> > >
> > > We appreciate your follow-up and are glad our clarifications and additional evidence helped address your concerns.

---

### Decision · Program_Chairs · 2026-04-30

**Decision:**

Accept (spotlight)

**Comment:**

The paper proposes to evaluate the reasoning process itself by representing it as a DAG. It builds a contrastive dataset to evaluate the quality of reasoning traces and build a reward model for it, which can then be used to RL the model. Basically the paper turns reasoning quality into something measurable. The framework is useful for understanding, evaluation and optimization, and thus recommend acceptance.